# FedEBA+: Towards Fair and Effective Federated Learning via Entropy-Based Model

**Zhichao Wang** [* 1]  **Lin Wang** [* 1]  **Ye Shi** [2]  **Sai Praneeth Karimireddy** [3]  **Xiaoying Tang** [† 1 4 5]

## Abstract

Federated Learning (FL) often suffers from sacrificing global model accuracy when improving client-level fairness due to data heterogeneity, which often leads to inconsistent performance of the globally trained models, resulting in unfair outcomes among users. Existing fair FL algorithms face a bottleneck: they either sacrifice global model accuracy to promote fairness or fall short of achieving optimal fairness.

In this paper, we propose a novel framework that effectively improves fairness while preserving global accuracy by integrating information-theoretic principles with model alignment. Specifically, we leverage the Maximum Entropy Principle to derive an analytic, closed-form solution for fair aggregation weights, ensuring significant fairness enhancements. We further employ a stepwise model alignment strategy that synchronizes gradient directions across heterogeneous clients, effectively mitigating the drift induced by local updates.

Theoretical analysis proves that our method guarantees convergence even in non-convex settings. Importantly, we push the theoretical frontier of federated fairness by extending performance variance analysis to generalized regression, providing broader guarantees. Extensive experiments on five datasets demonstrate that our approach consistently outperforms state-of-the-art methods, achieving superior fairness without sacrificing global accuracy. Our code is available at https://github.com/T-Lab-CUHKSZ/FedEBA-Plus.

---
*Equal contribution  [1]The Chinese University of HongKong, Shenzhen [2]ShanghaiTech University [3]University of Southern California [4]Shenzhen Future Network of Intelligence Institute (FNii-Shenzhen) [5]Guangdong Provincial Key Laboratory of Future Networks of Intelligence, CUHK(SZ). Correspondence to: Xiaoying Tang <tangxiaoying@cuhk.edu.cn>.

*Proceedings of the 43rd International Conference on Machine Learning*, Seoul, South Korea. PMLR 306, 2026. Copyright 2026 by the author(s).

## 1 Introduction

Federated Learning (FL) is a distributed learning paradigm that allows clients to collaborate with a central server to train a global model (McMahan et al., 2017). Clients process data locally and only periodically transmit model updates, enabling collaborative learning without transferring sensitive data. A major challenge in FL stems from data heterogeneity, which causes the global model's performance to vary substantially across clients, leading directly to the critical issue of performance unfairness (Shi et al., 2023). Achieving fairness, where the global model's performance is uniformly distributed among all clients, is vital to prevent performance discrimination, client disengagement, and ethical concerns (Caton & Haas, 2020).

Significant progress has been made in addressing these disparities (Li et al., 2019a; Zhao & Joshi, 2022; Pan et al., 2023), though existing solutions typically face one of two challenges. On one hand, several approaches prioritize fairness but often come at the cost of global model accuracy (Li et al., 2019a; Mohri et al., 2019; Zhang et al., 2023). On the other hand, methods that implicitly model the optimization objective may yield sub-optimal fairness results, as they often lack explicit mechanisms to constrain performance variance (fairness) among diverse clients (Lin et al., 2022; Li et al., 2021).

To bridge this gap, we draw inspiration from an efficient mechanism rooted in information theory. Information-theoretic approaches, particularly those utilizing the Maximum Entropy Principle, are widely recognized and frequently employed in machine learning and optimization to address fairness and resource balancing challenges (Singh & Vishnoi, 2014; Johansson & Sternad, 2005). These models are successful because they inherently seek the most uniform distribution subject to constraints. We ground our approach in the fundamental principle that high-entropy weight distributions inherently enforce uniformity. This principle serves as a natural mechanism to mitigate the performance variance (thus promoting fairness) among clients in FL.

However, a common misconception pervades the direct use of this tool, leading to a frequent mistake: equating stan-

dard entropy maximization directly with a viable fairness solution. In FL, fairness requires equitable performance across diverse clients with heterogeneous data (Shi et al., 2023; Donahue & Kleinberg, 2021), not just uniform resource distribution. To address this, FedEBA+ formulates entropy over aggregation distribution, constraining the distance between aggregated and fairness-aware objectives (see Section 4.1), leading to an aggregation distribution proportional to loss. Compared with typical aggregation methods, like FedAvg (McMahan et al., 2017) and q-FFL (Li et al., 2019a), FedEBA+ ensures more uniform client accuracy (please refer to Appendix J.1 for a toy-case example). The maximum entropy model efficiently provides an analytic solution at each computation step, making the optimization problem computationally efficient without requiring cyclic updates.

**Our major contributions can be summarized as below:**

- We propose a novel fair FL algorithm, FedEBA+, involving a well-designed objective function capturing both the global model performance and the entropy-based fair aggregation, aimed at enhancing fairness without sacrificing the overall accuracy of the FL global model. We leverage the Maximum Entropy Principle to derive an analytic, closed-form solution for fair aggregation weights, ensuring significant fairness enhancements with minimal computational overhead. We further employ the alignment strategy that synchronizes gradient directions across heterogeneous clients, effectively mitigating the drift induced by local updates

- To alleviate the communication burdens, we further present a practical algorithm Prac-FedEBA+, achieving competitive performance with communication costs comparable to FedAvg.

- Theoretically, we provide the convergence guarantee for FedEBA+ under a nonconvex setting. Importantly, we push the theoretical frontier of federated fairness by extending performance variance analysis to generalized regression, providing broader guarantees.

- Extensive evaluations across five datasets and four neural architectures demonstrate that our approach consistently outperforms state-of-the-art federated fairness algorithms in both fairness and global accuracy. Furthermore, experiments confirm that our methods exhibit significant robustness to label noise and privacy protection.

## 2  Related Work

There have been encouraging efforts to address fairness in Federated Learning, including function-based approaches like q-FFL (Li et al., 2019a) and AFL (Deng et al., 2020), gradient-based methods such as FedFV (Wang et al., 2021) and MGDA (Hu et al., 2022; Pan et al., 2023), and person-

alized methods (Li et al., 2021; Lin et al., 2022). Despite their success in promoting fairness, a key remaining frontier in this field is the simultaneous explicit optimization of fairness and global performance. Currently, explicitly addressing the dual objective of minimizing performance variance without compromising overall model accuracy presents an ongoing research opportunity. To this end, we propose a novel optimization algorithm to enhance fairness among clients while maintaining the global accuracy. One closely related work is AAggFF (Hahn et al., 2024). It proposes an aggregation strategy based on sequential decision-making, which shares a key principle with ours: assigning higher weights to under-performing clients can enhance fairness in FL. This provides further empirical support for entropy-based methods, as both approaches share a conceptually similar formulation. In contrast, our method is theoretically grounded in a constrained maximum entropy optimization model. Moreover, beyond aggregation, we introduce a stepwise alignment update, which improves client-level fairness while maintaining the global accuracy. These distinctions position our approach as a novel and principled extension of the paradigm.

Due to page limits, a more comprehensive discussion of the related work and fairness concepts can be found in Appendix B and Appendix C.

## 3  Preliminaries and Metrics

**Notations.** Let $N$ be the number of clients and $|S_t| = m$ be the number of selected clients for round $t$. We denote $K$ as the number of local update steps and $T$ as the total number of communication rounds. We use $F_i(x)$ and $f(x)$ to represent the local and global loss of client $i$ with model $x$, respectively. Specifically, $x_{t,k}^i$ and $g_{t,k}^i = \nabla F_i(x_{t,k}^i, \xi_{t,k}^i)$ represent the model parameter and local gradient of the $k$-th local step in the $i$-th worker after the $t$-th communication, respectively. $x$ is the global model and $x_t$ is the global model at round $t$. The global model update is denoted as $\Delta_t = x_{t+1} - x_t$, while the local model update is represented as $\Delta_t^i = x_{t,K}^i - x_{t,0}^i$.

**Problem Formulation.** The typical FL objective can be formulated as follows:

$$\min_x f(x) = \sum_{i=1}^N p_i F_i(x),  \tag{1}$$

where $F_i(x) = \mathbb{E}_{\xi_i \sim D_i} F_i(x, \xi_i)$ is the local objective function of client $i$ over data distribution $D_i$, $\xi_i$ means the sampled data of client $i$ and $p_i$ represents the aggregation weight of client $i$.

In this paper, our goal is to enhance fairness while maintaining the accuracy of the global model. This motivates us to establish the following optimization objective as our *final*

*objective*:

$$x^* = \arg\min_x f(x) = \arg\min_x \left\{ \sum_{i=1}^{N} p_i F_i(x) + \beta \Phi(x) \right\}, \tag{2}$$

where $x^*$ is the optimal model parameters, $F_i(x)$ is the local loss on client $i$, and $f(x)$ represents the global model's loss, aimed at optimizing the global model's performance. $\beta > 0$ is the penalty coefficient of the fairness regularization, while $\Phi(x)$ is the regularization term that aims to improve fairness. Thus, optimizing this objective entails enhancing the global model's fairness without sacrificing accuracy. Our method shares a high-level functional similarity with the innovative AAggFF approach (Hahn et al., 2024) in that both emphasize underperforming clients through aggregation weights. The underlying mechanisms and scope of application are fundamentally distinct. Specifically, AAggFF employs a sequential decision-making process to adjust client weights via online learning. In sharp contrast, our approach is rigorously grounded in constrained maximum entropy optimization to derive these aggregation weights. Furthermore, we introduce a novel step-wise model alignment approach, which significantly enhances fairness without compromising global test accuracy.

We explicitly formulate $\Phi(x)$ in Section 4.2, building on the fair aggregation optimization in Section 4.1, and rewrite the optimization problem (2) to (7).

**Metrics.** This paper aims to *1) promote fairness* in FL while *2) maintain the global model's performance*. Typically, the global model's performance is evaluated based on its accuracy or loss. Regarding the fairness metric, we adhere to the definition proposed by (Li et al., 2019a), which employs the variance of clients' performance as the fairness metric:

**Definition 3.1** (Fairness via variance). *A model $x_A$ is fairer than $x_B$ if the test performance distribution of $x_A$ across the network with $N$ clients is more uniform than that of $x_B$, i.e. $\mathrm{var}\{F_i(x_A)\}_{i\in[N]} < \mathrm{var}\{F_i(x_B)\}_{i\in[N]}$, where $F_i(\cdot)$ denotes the test loss of client $i \in [N]$ and $\mathrm{var}\{F_i(x)\} = \frac{1}{N}\sum_{i=1}^{N}\left[F_i(x) - \frac{1}{N}\sum_{i=1}^{N}F_i(x)\right]^2$ denotes the variance.*

Ensuring the global model's accuracy is the fundamental goal of FL. However, fairness-targeted algorithms may compromise high-performing clients to mitigate variance (Shi et al., 2023). Our evaluation of fairness algorithms extends beyond global accuracy, considering the accuracy of the best $5\%$ and worst $5\%$ clients. This analysis, also viewed as a form of robustness in some studies (Yu et al., 2023; Li et al., 2021), provides insights into potential compromises.

---

**Algorithm 1** FedEBA+

1: **Input:** Number of selected clients per round $m$, global learning rate $\eta$, local learning rate $\eta_L$, number of local update steps $K$, total training rounds $T$.
2: **Output:** Final model parameter $x_T$.
3: **Initialize:** model $x_0$.
4: **for** round $t = 1, \ldots, T$ **do**
5:     Server selects a set of clients $|S_t|$ and broadcast model $x_t$;
6:     Server collects selected clients' loss $\mathbf{L} = [F_1(x_t), \ldots, F_m(x_t)]$;
7:     Server receives $\nabla F_i(x_t)$, calculates the fair gradient and broadcasts it to clients: $\tilde{g}^t = \sum_{i\in S_t} \frac{\exp[F_i(x_t)/\tau]}{\sum_{j\in S_t}\exp[F_j(x_t)/\tau]}\nabla F_i(x_t)$;
8:     **for** Client $i \in S_t$ in parallel **do**
9:         **for** $k = 0, \cdots, K-1$ **do**
10:             $h_{t,k}^i \leftarrow (1-\alpha)\nabla F_i(x_{t,k}^i; \xi_i) + \alpha\tilde{g}^t$;
11:         **end for**
12:         $\Delta_t^i = x_{t,K}^i - x_{t,0}^i = -\eta_L\sum_{k=0}^{K-1}h_{t,k}^i$;
13:     **end for**
14:     Aggregation: $\Delta_t = \sum_{i\in S_t}p_i\Delta_t^i$, where $p_i = \frac{\exp[F_i(x_{t,K}^i)/\tau]}{\sum_{j\in S_t}\exp[F_j(x_{t,K}^j)/\tau]}$;
15:     Server update: $x_{t+1} = x_t + \eta\Delta_t$;
16: **end for**

---

## 4 FedEBA+: An Effective Fair Algorithm

In this section, we present FedEBA+ by explicitly connecting its optimization objective with the algorithmic design. Our method is built from a constrained maximum entropy formulation over client aggregation probabilities. Solving this formulation yields an entropy-based aggregation rule that assigns larger weights to underperforming clients, thereby improving fairness (Sec. 4.1). Based on this formulation, we further combine the standard FL objective with the fairness constraint term, as shown in Eq. (2). The resulting objective gives rise to the step-wise alignment update when optimized with respect to the model parameter, providing a principled way to guide local updates toward the fairness-aware direction (Sec. 4.2). The complete procedure is summarized in Algorithm 1.

### 4.1 Entropy-Based Aggregation for Fairness: EBA

Inspired by the success of Shannon entropy in promoting fairness (Jaynes, 1957), which ensures unbiased probability distribution by maximizing neutrality towards unobserved information and eliminating inherent bias (Hubbard et al., 1990; Sampat & Zavala, 2019), we formulate the following optimization problem with designed constraints on FL aggregation:

$$\max_{p_i, \forall i\in[N]} \mathbb{H}(p_i) := -\sum_{i=1}^{N} p_i\log(p_i),$$
$$s.t. \quad \sum_{i=1}^{N}p_i = 1, \; p_i \geq 0, \; \sum_{i=1}^{N}p_iF_i(x^i) = \tilde{f}(x). \tag{3}$$

$\mathbb{H}(p_i)$ denotes the entropy of aggregation probability $p_i$, and $\tilde{f}(x)$ signifies the fairness-aware loss, representing the global model's performance under an ideal training setting. Conceptually, $\tilde{f}(x)$ represents the target objective that the aggregated model should approach when client performance is made fair. In practical FL, however, the full $\tilde{f}(x)$ and its gradient are not directly available because only a subset of clients $S_t$ participates in each communication round. Therefore, we construct a principled approximation of its gradient from the available clients in round $t$ by applying the same entropy-based weighting to their losses and gradients. This approximation yields the fairness-aware gradient as:

$$\nabla \tilde{f}(x_{t,k}^i) = \tilde{g}^t = \sum\nolimits_{i \in S_t} p_i \nabla F_i(x_{t,k}), \qquad (4)$$

where $p_i = \exp[F_i(x_t)/\tau] / \sum_{j \in S_t} \exp[F_j(x_t)/\tau]$, $\tilde{g}^t$ represents the fair gradient of the selected clients, obtained using the global model's performance on these clients without severe local shift (i.e., one local update step). In particular, for each local epoch $k$, we use the same fair gradient, independent of $k$.

The classical entropy model reduces prior distribution knowledge and avoids bias from subjective influences. Compared to the existing entropy model of fairness (Johansson & Sternad, 2005), we first incorporate the FL constraints $\sum_{i=1}^{N} p_i F_i(x^i) = \tilde{f}(x)$ to force aggregation into the fair regularization region, specifically improving fairness. Maximizing constrained entropy implies greater fairness, as shown in the toy example in Appendix J.1.

**Proposition 4.1.** *By solving the constrained maximum entropy problem, we propose an aggregation strategy called **EBA** to enhance fairness in FL, expressed as follows:*

$$p_i = \frac{\exp[F_i(x^i)/\tau]}{\sum_{j=1}^{N} \exp[F_j(x^j)/\tau]}, \qquad (5)$$

*where $\tau > 0$ is the temperature, and the derivation of $\tau$ is related to $\tilde{f}(x)$.*

Details for deriving the above proposition and the proof of the uniqueness of the solution for the constrained maximum entropy model are provided in Appendix D.1 and L, respectively. To theoretically prevent $\min p_i$ from dropping below a minimal positive value $\epsilon_p$, which could destabilize the algorithm and invalidate the unbounded analysis, the temperature parameter $\tau$ must be dynamically clamped. This clamping is dependent on the loss range ($\Delta F = \max F(x) - \min F(x)$) across all participating clients in the current communication round. Specifically, we set the lower bound for $\tau$ as:$\tau_{\min} = \frac{F_{\max} - F_{\min}}{\ln\left(\frac{1}{m\epsilon_p}\right)}$. Crucially, this constraint serves only as a technical requirement for analytical stability and requires no practical tuning. (For the detailed derivation, please refer to the Appendix I)

Proposition 4.1 shows that assigning higher aggregation weights to underperforming clients directs the aggregated

global model's focus toward these users, enhancing their performance and reducing the gap with top performers, ultimately promoting fairness, as shown in the experiments in Table 14. It is worth noting that the aggregation probability can be solved in closed form, relying solely on the loss of the local model, making it computationally efficient.

When taking into account the prior distribution of aggregation probability $p_i$, which is typically expressed as the relative data ratio $q_i = v_i / \sum_{i \in S_t} v_i$ where $v_i$ is the number of data in client $i$, the expression of fair aggregation probability becomes $p_i = \frac{q_i \exp[F_i(x)/\tau]}{\sum_{j=1}^{N} q_j \exp[F_j(x)/\tau]}$. Without loss of generality, we utilize Eq. (5) to represent entropy-based aggregation in this paper. The derivations for fair aggregation probability expression without prior distribution are given in Appendix D.1. Robust variants of the aggregation algorithm are shown in Appendix E.

### 4.2 Final Objective Formulation and Step-Wise Alignment Update

Recall the *final objective* (2) to develop an objective function that simultaneously improves fairness and global model performance. Based on the proposed maximum entropy model, we define

$$\Phi = - \left[ \sum_{i=1}^{N} p_i \log p_i + \lambda_0 \left( \sum_{i=1}^{N} p_i - 1 \right) \right. \\ \left. + \frac{1}{\tau} \left( \tilde{f}(x) - \sum_{i=1}^{N} p_i F_i(x) \right) \right] \qquad (6)$$

where $N$ is the total number of clients. Maximizing $\Phi$ with respect to $p_i$ ensures the same fair aggregation result as Proposition 4.1. Thus, we develop *final objective* as below:

$$\min_x \max_{p_i} L(x, p_i) := \sum_{i=1}^{N} p_i F_i(x) - \beta \left[ \sum_{i=1}^{N} p_i \log p_i \right. \\ \left. + \lambda_0 \left( \sum_{i=1}^{N} p_i - 1 \right) + \frac{1}{\tau} \left( \tilde{f}(x) - \sum_{i=1}^{N} p_i F_i(x) \right) \right], \qquad (7)$$

Eq. (7) can be viewed as the standard weighted FL objective regularized by the constrained entropy term $\Phi$. This formulation directly motivates our second core design: step-wise alignment.

**Step-Wise Alignment.** For the model variable $x$, differentiating Eq. (7) gives

$$\frac{\partial L(x, p_i)}{\partial x} = \left( 1 + \frac{\beta}{\tau} \right) \sum_{i=1}^{N} p_i \nabla F_i(x) - \frac{\beta}{\tau} \nabla \tilde{f}(x). \qquad (8)$$

By reparameterizing the coefficient as the alignment strength $\alpha$, the update direction can be written as

$$\frac{\partial L(x, p_i)}{\partial x} = (1 - \alpha) \sum_{i=1}^{N} p_i \nabla F_i(x) + \alpha \nabla \tilde{f}(x). \qquad (9)$$

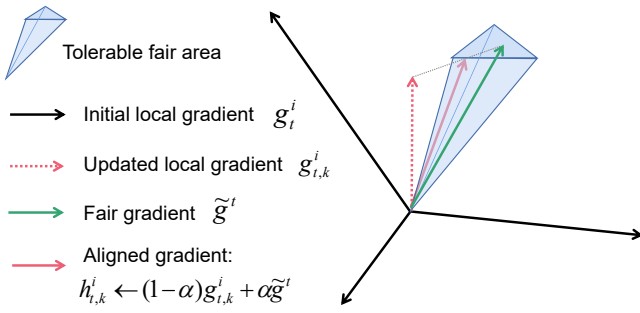

*Figure 1.* **Gradient Alignment.** Gradient alignment ensures that each local step's gradient stays on track and does not deviate too far from the fair direction. It achieves this by constraining the aligned gradient, denoted by $h_{t,k}^i$, to fall within the tolerable fair area. The gradient $g_t^i$ represents the gradient of global model for each client in round $t$, while $\tilde{g}^t$ denotes the fairness-aware gradient for model $x_t$. The gradient $g_{t,k}^i = \nabla F_i(x_{t,k}^i; \xi_i)$ is the gradient of client $i$ at round $t$ and local update step $k$.

This expression reveals the role of step-wise alignment: FedEBA+ combines the entropy-weighted FL gradient with a fairness-aware gradient $\nabla \tilde{f}(x)$.

Based on the model update direction in Eq. (9), FedEBA+ applies step-wise alignment to guide each local update toward a fairness-aware direction. Specifically, the server first receives $F_i(x_t)$ and $\nabla F_i(x_t)$ from selected clients, and computes the entropy-weighted fair gradient

$$\tilde{g}^t = \sum\nolimits_{i \in S_t} p_i \nabla F_i(x_t), \qquad p_i = \frac{\exp[F_i(x_t)/\tau]}{\sum_{j \in S_t} \exp[F_j(x_t)/\tau]}. \tag{10}$$

Here, $\tilde{g}^t$ represents the fairness-aware direction estimated from the selected clients at the global model $x_t$, before local drift occurs. During local training, each client aligns its stochastic gradient with this fair direction:

$$h_{t,k}^i \leftarrow (1 - \alpha)\nabla F_i(x_{t,k}^i; \xi_i) + \alpha \tilde{g}^t. \tag{11}$$

This step-wise alignment prevents local updates from drifting too far away from the fair direction across multiple local steps, eliminating the potential severe conflicts that may arise during gradient updates, while ensuring that the global test accuracy is not compromised. The procedure is depicted in Algorithm 1, Steps 8–15.

### 4.3 Practical Gradient Alignment to Reduce Communication Cost

Note that in the above discussion, the server needs to obtain the one-step update to calculate the aligned gradient $\tilde{g}^t$ and sends it back to clients for local update. Considering the communication burden of FL, we propose a practical version of the gradient alignment method:

**Proposition 4.2.** *For approximating the aligned gradient and overcoming the communication overhead issue, we use the average of multiple local updates to approximate the*

*one-step gradient. Then, the fair gradient is approximated by:*

$$\tilde{g}^t = \sum\nolimits_{i \in S_t} \frac{\exp[F_i(x_t)/\tau]}{\sum_{j \in S_t} \exp[F_j(x_t)/\tau]} \frac{1}{K} \sum\nolimits_{k=0}^{K-1} \nabla F_i(x_{t,k}^i; \xi_i). \tag{12}$$

*In this way, the client only needs to communicate the model once to the server, **the same as FedAvg**. The complete practical algorithm, named Prac-FedEBA+, is presented in Algorithm 3.*

## 5 Analysis of Convergence and Fairness

In this section, we analyze the convergence and fairness property of FedEBA+.

### 5.1 Convergence Analysis of FedEBA+

To facilitate the theoretical analysis, we adopt common assumptions for nonconvex federated learning: L-smoothness, unbiased local gradient estimators, and bounded gradient dissimilarity. See Appendix H for assumptions' details.

**Theorem 5.1.** *Under Assumption 1–3, and let constant local and global learning rate $\eta_L$ and $\eta$ be chosen such that $\eta_L < \min(1/(8LK), C)$, where $C$ is obtained from the condition that $\frac{1}{2} - 10L^2 \frac{1}{N} \sum_{i=1}^N K^2 \eta_L^2 (A^2 + 1)(\chi_{p\|w}^2 A^2 + 1) > C > 0$, and $\eta \leq 1/(\eta_L L)$. In particular, let $\eta_L = \mathcal{O}\left(\frac{1}{\sqrt{T}KL}\right)$ and $\eta = \mathcal{O}\left(\sqrt{NK}\right)$, the convergence rate of Algorithm 1 (FedEBA+) with $\alpha = 0$ is:*

$$\min_{t \in [T]} \mathbb{E} \|\nabla f(\boldsymbol{x}_t)\|^2 \leq \mathcal{O}\left(\frac{(f^0 - f^*) + N/2 \sum_i w_i^2 \sigma_L^2}{\sqrt{NKT}}\right)$$
$$+ \mathcal{O}\left(\frac{5(\sigma_L^2 + 4K\sigma_G^2) + 40K(A^2 + 1)\chi_{\boldsymbol{w}\|\boldsymbol{p}}^2 \sigma_G^2}{2KT}\right). \tag{13}$$

Here, $A \geq 0$ is a constant defined in Assumption 3, and $w$ is the prior aggregation distribution detailed in Lemma I.1. The proof details of Theorem 5.1 are provided in Appendix I.

**Remark 5.2.** *According to the property of unified probability, we know $\frac{1}{N} \leq \sum_{i=1}^N w_i^2 \leq 1$, where the right inequality comes from $\sum_i w_i^2 \leq \sum_i w_i$ and the left inequality comes from Cauchy-Schwarz inequality. Therefore, the worst case of the convergence rate will be $\mathcal{O}(\frac{\sqrt{N}}{\sqrt{KT}} + \frac{1}{T})$.*

**Remark 5.3.** *When $\alpha \neq 0$, the convergence rate of FedEBA+ is:* $\min_{t \in [T]} \mathbb{E} \|\nabla f(\boldsymbol{x}_t)\|^2 \leq \mathcal{O}(\frac{(1-\alpha)^2 \sum_i w_i^2 \sqrt{m}\sigma_L^2 + \alpha^2 \sqrt{K}\rho^2}{\sqrt{KT}} + \frac{1}{T})$, where $\sigma_L \sim \rho$ by Assumption 4, thus a larger $\alpha$ indicating a tighter convergence upper bound than only using reweight aggregation with $\alpha = 0$. $K$ represents the local epoch times (in each communication round) and $m$ represents the client numbers, usually client numbers are larger than the local epoch in*

the cross-device FL. In addition, when $w_i = \frac{1}{N}$, i.e., uniform aggregation, the rate is $\mathcal{O}\big(\frac{(1-\alpha)^2\sigma_L^2 + \alpha^2\sqrt{K/N}\rho^2}{\sqrt{NKT}} + \frac{1}{T}\big)$. When $\sqrt{K/N} << 1$, using the proposed alignment update results in a faster convergence rate than FedAvg. The proof details are provided in Appendix I.2.

## 5.2 Fairness Analysis of FedEBA+

**Variance Analysis.** We analyze the cross-client test-loss variance of FedEBA+ under both the generalized linear regression model and the strongly convex model.

**Theorem 5.4.** *Under the two settings specified below, FedEBA+ exhibits no larger cross-client test-loss variance than FedAvg:*

*(1) For the generalized regression model, following the setup in (Li et al., 2020a), the loss is formulated as $f(\mathbf{x};\xi) = T(\xi)^\top \mathbf{x} - A(\xi)$, where $T(\xi)$ represents the generalized regression coefficient and $A(\xi)$ denotes the Gaussian noise term. We then derive the test-loss variance of FedEBA+ and compare it with FedAvg:*

$$\mathrm{Var}_{i\in[N]}\left[F_i^{test}(\boldsymbol{x}_{EBA+})\right] = \frac{\tilde{b}^2}{4}\mathrm{Var}_{i\in[N]}\left[\|\tilde{\mathbf{x}}^\star - \mathbf{x}_i^\star\|_2^2\right],$$

$$\mathrm{Var}_{i\in[N]}\left[F_i^{test}(\boldsymbol{x}_{EBA+})\right] \le \mathrm{Var}_{i\in[N]}\left[F_i^{test}(\boldsymbol{x}_{Avg})\right].$$

$$(14)$$

*where $\mathbf{x}_i^\star$ represents the true parameter of client $i$, $\tilde{\mathbf{x}}^\star = \sum_{i=1}^N p_i \mathbf{x}_i^\star$ denotes the weighted population parameter induced by FedEBA+, and $\tilde{b}$ is a constant that approximates $b_i$ in $\boldsymbol{\Xi}_i^\top \boldsymbol{\Xi}_i = nb_i\mathbf{I}_d$, where $\boldsymbol{\Xi}_i = [T(\xi_{i,1}),\dots,T(\xi_{i,n})]$. The data heterogeneity is reflected in the heterogeneity of $\mathbf{x}_i^\star$.*

*(2) For the strongly convex setting, we assume each client's loss to be smooth and strongly convex, following the setting in (Chu et al., 2023). Assuming the existence of an outlier, we derive the test-loss variance of FedEBA+ and compare it with FedAvg:*

$$\mathrm{Var}_{i\in[N]}\left[F_i^{test}(\boldsymbol{x}_{EBA+})\right] = \frac{1}{N}\sum_{i=1}^N \tilde{L}_i^2 - \left(\frac{1}{N}\sum_{i=1}^N \tilde{L}_i\right)^2,$$

$$\mathrm{Var}_{i\in[N]}\left[F_i^{test}(\boldsymbol{x}_{EBA+})\right] \le \mathrm{Var}_{i\in[N]}\left[F_i^{test}(\boldsymbol{x}_{Avg})\right],$$

$$(15)$$

*where $\tilde{L}_i := F_i^{test}(\boldsymbol{x}_{EBA+})$ is the test loss of FedEBA+ on client $i$, distinct from the training loss $F_i(\mathbf{x})$.*

Details regarding the linear regression setting, the smooth and strongly convex assumptions, and the detailed derivations are presented in Appendix J.2 and Appendix J.3.

In addition to analyzing cross-client test-loss variance in federated learning, we demonstrate that the entropy-based aggregation rule of FedEBA+ satisfies Pareto-optimality and uniqueness as per Property 1 of (Sampat & Zavala, 2019). This supports the fairness effectiveness of our aggregation strategy, with further details provided in Appendix K and Appendix L.

# 6 Numerical Results

## 6.1 Experimental Setup

**Datasets and Models.** We test the performance of FedEBA+ on five public datasets: MNIST (LeCun et al., 1998), Fashion MNIST (Xiao et al., 2017), CIFAR-10, CIFAR-100 (Krizhevsky et al., 2009), and Tiny-ImageNet (Deng et al., 2009).

As for the model, we use the MLP model (Rumelhart et al., 1986) with 2 hidden layers on MNIST and Fashion-MNIST, and a CNN model (LeCun et al., 1998) with 2 convolution layers on CIFAR-10, ResNet-18 (He et al., 2016) on CIFAR-100, and MobileNet-v2 (Sandler et al., 2018) on Tiny-ImageNet.

**Federated Data Partitioning.** We use two methods to split the datasets into non-iid datasets [1]:

- **2 Shards for Each Client**. Following the setting of (Wang et al., 2021), where 100 clients participate in the federated system, and according to the labels, we divide Fashion-MNIST, CIFAR-10, and MNIST (please refer to Appendix N for results on MNIST) into 200 shards separately, and each user randomly picks up 2 shards for local training.

- **Dirichlet Partition**. We leverage Latent Dirichlet Allocation (LDA) to control the distribution drift with the Dirichlet parameter $\alpha$. We utilized the Dirichlet distribution ($\alpha = 0.1$) to partition CIFAR-100 and Tiny-ImageNet-200 datasets and report the corresponding results in the main paper. A comprehensive analysis of results using different $\alpha$ values is provided in Appendix N.

**Metrics and Baselines.** We use **variance**, worst 5% accuracy, and best 5% accuracy as performance metrics for fairness evaluation, and **global accuracy** to evaluate the global model's performance. We compare FedEBA+ with FedAvg, FedSGD (McMahan et al., 2016), and fair FL algorithms, including AFL (Mohri et al., 2019), q-FFL (Li et al., 2019a), FedMGDA+(Hu et al., 2022), PropFair (Zhang et al., 2023), TERM (Li et al., 2020a), FOCUS (Chu et al., 2023), Ditto (Li et al., 2021), AAggFF (Hahn et al., 2024) and lp-proj (Lin et al., 2022). Additional implementation details, such as number of clients, number of clients selected per communication rounds, models and hyperparameters, are available in Appendix M.

---

[1]We select the most common representative and challenging extreme Non-iid scenarios

*Table 1.* **Performance of algorithms on FashionMNIST and CIFAR-10.** We highlight the best and the second-best results by using **bold font** and blue text. We report the accuracy of the global model, variance fairness, $C_V$, worst 5%, and best 5% accuracy, where $C_V = $ Global Acc./Var. and a larger value indicates a better accuracy-fairness trade-off. The data is divided into 100 clients, with 10 clients sampled in each round. All experiments are running over 2000 rounds for a single local epoch with $K = 10$ update steps, local batch size $= 50$, and learning rate $\eta = 0.1$. The reported results are averaged over 5 runs with different random seeds.

| Algorithm | FashionMNIST | | | | | CIFAR-10 | | | | |
|---|---|---|---|---|---|---|---|---|---|---|
| | Global Acc. ↑ | Var. ↓ | $C_V$ ↑ | Worst 5% ↑ | Best 5% ↑ | Global Acc. ↑ | Var. ↓ | $C_V$ ↑ | Worst 5% ↑ | Best 5% ↑ |
| FedAvg | 86.49 ±0.09 | 62.44 ±4.55 | 1.39 | 71.27 ±1.14 | 95.84 ±0.35 | 67.79 ±0.35 | 103.83 ±10.46 | 0.65 | 45.00 ±2.83 | 85.13 ±0.82 |
| q-FFL | 86.57 ±0.19 | 54.91 ±2.82 | 1.58 | 70.88 ±0.98 | 95.06 ±0.17 | 68.76 ±0.22 | 97.81 ±2.18 | 0.70 | 48.33 ±0.84 | 84.51 ±1.33 |
| FedMGDA+ | 84.64 ±0.25 | 57.89 ±6.21 | 1.46 | **73.49 ±1.17** | 93.22 ±0.20 | 65.19 ±0.87 | 89.78 ±5.87 | 0.73 | 48.84 ±1.12 | 81.94 ±0.67 |
| Ditto | 86.37 ±0.13 | 55.56 ±5.43 | 1.55 | 69.20 ±0.37 | 95.79 ±0.38 | 60.11 ±4.41 | 85.99 ±7.13 | 0.70 | 42.20 ±2.20 | 77.90 ±4.90 |
| AFL | 84.14 ±0.18 | 90.76 ±6.13 | 0.93 | 60.11 ±0.69 | 96.00 ±0.09 | 65.60 ±0.14 | 87.67 ±2.39 | 0.75 | 46.01 ±0.40 | 82.30 ±0.12 |
| PropFair | 85.51 ±0.28 | 75.27 ±5.38 | 1.14 | 63.60 ±0.53 | 97.60 ±0.19 | 65.79 ±0.53 | 79.67 ±5.71 | 0.83 | 49.88 ±0.93 | 82.40 ±0.40 |
| TERM | 84.31 ±0.38 | 73.46 ±2.06 | 1.15 | 68.23 ±0.10 | 94.16 ±0.16 | 65.41 ±0.37 | 91.99 ±2.69 | 0.71 | 49.08 ±0.66 | 81.98 ±0.19 |
| FOCUS | 86.24 ±0.18 | 61.15 ±1.17 | 1.41 | 68.15 ±0.25 | **98.50 ±0.10** | 59.60 ±1.52 | 455.14 ±11.19 | 0.13 | 9.54 ±0.18 | **87.72 ±0.12** |
| lp-proj | 86.21 ±0.02 | 56.71 ±2.25 | 1.52 | 68.47 ±0.37 | 97.86 ±0.52 | 68.86 ±0.51 | 78.65 ±7.01 | 0.88 | 49.53 ±1.11 | 83.33 ±1.23 |
| Rank-Core-Fed | 85.54 ±0.33 | 58.19 ±2.83 | 1.47 | 67.80 ±0.55 | 96.60 ±0.40 | 67.15 ±1.12 | 87.02 ±2.46 | 0.77 | 45.41 ±0.62 | 85.82 ±0.20 |
| AAggFF | 86.83 ±0.33 | 58.19 ±2.83 | 1.49 | 69.44 ±0.73 | 97.67 ±0.32 | 68.07 ±0.09 | 87.21 ±5.04 | 0.78 | 47.4 ±0.53 | 84.98 ±0.19 |
| Prac-FedEBA+ | 86.62 ±0.07 | 46.41 ±0.88 | 1.87 | 71.40 ±0.15 | 96.10 ±0.46 | 69.83 ±0.34 | 74.16 ±1.66 | 0.94 | 52.40 ±0.50 | 84.10 ±0.39 |
| FedEBA+ | **87.50 ±0.19** | **43.41 ±4.34** | **2.02** | 72.07 ±1.47 | 95.91 ±0.19 | **72.75 ±0.25** | **68.71 ±4.39** | **1.06** | **55.80 ±1.28** | 86.93 ±0.52 |

## 6.2   Main Results

**Fairness Performance.**   As shown in Table 1 and Table 2, FedEBA+ significantly reduces performance variance (thus promoting fairness). The variance improvement is **11.5%** on FashionMNIST and nearly **10%** on CIFAR-10 compared to the best-performing baseline. One may notice the AFL's significant improvement in fairness on CIFAR-100. This is due to the fact that the optimization of the worst client affected the performance of the global model, leading even the clients who could have performed better to be affected as well.

**Global Accuracy.**   Table 1 and Table 2 demonstrate that many baselines improve fairness (i.e., reduce variance) at the expense of global model accuracy, showing either lower global accuracy or limited improvement compared to FedAvg. And Figure 2(a) clearly shows FedEBA+'s superiority in both fairness and global accuracy. It consistently maintains excellent global test accuracy (e.g., nearly 4% on CIFAR-10 compared with the second-best baseline) while significantly enhancing fairness. Figure 2(b) demonstrates that our algorithm can converge to a better global accuracy.

Perhaps the most striking thing is that, with the same communication cost as FedAvg, Prac-FedEBA+ surpasses other baselines in terms of fairness and still maintains a decent global test accuracy, which achieves the second-best result on three datasets.

**The Trade-Off between Fairness and Accuracy.**   To further evaluate whether fairness improvement is achieved without sacrificing global accuracy, we introduce an additional trade-off metric, denoted as $C_V = \frac{Accuracy}{Variance}$, where a larger value indicates a better balance between high global accuracy and low cross-client performance variance. This metric complements the original evaluation metrics by jointly reflecting the accuracy-fairness trade-off.

As shown in Tables 1, 2, and 3, FedEBA+ consistently achieves the best $C_V$ across FashionMNIST, CIFAR-10, CIFAR-100, and Tiny-ImageNet under standard settings, demonstrating that the fairness gains of FedEBA+ do not come at the cost of global accuracy. The practical variant Prac-FedEBA+ also obtains the second-best $C_V$ on Fashion-MNIST and CIFAR-10 in Table 1, and on Tiny-ImageNet in Table 2, indicating that the communication-efficient approximation preserves a strong accuracy-fairness trade-off. Under severe label corruption, FedEBA+ again achieves the highest $C_V$ in both the vanilla and LSR-enhanced groups on FashionMNIST and CIFAR-10, further confirming its robustness in maintaining global performance while improving fairness.

**Ablation Study.**

- **All the components of FedEBA+ are necessary**. In Table 14 of Appendix N, we conduct the ablation study on FedEBA+, showing that each step of FedEBA+ is beneficial. Even the aggregation alone improves global performance and fairness.

- **FedEBA+ is stable to the hyperparameters**. Figure 3(a) indicates that increasing $\alpha$ improves fairness but may slightly decrease accuracy. Figure 3(b) demonstrates that decreasing $\tau$ enhances fairness, with $\tau > 1$ generally leading to better global accuracy. Overall, the selection of hyperparameters does not cause the

*Table 2.* **Performance of algorithms on CIFAR-100 and Tiny-ImageNet.** We report the accuracy of the global model, variance fairness, $C_V$, worst 5%, and best 5% accuracy, where $C_V = $ Global Acc./Var. and a larger value indicates a better accuracy-fairness trade-off. The local batch size is 128.

| Algorithm | CIFAR-100 | | | | | Tiny-ImageNet | | | | |
|---|---|---|---|---|---|---|---|---|---|---|
| | Global Acc. ↑ | Var. ↓ | $C_V$ ↑ | Worst 5% ↑ | Best 5% ↑ | Global Acc. ↑ | Var. ↓ | $C_V$ ↑ | Worst 5% ↑ | Best 5% ↑ |
| FedAvg | 30.94 ±0.04 | 297.22 ±2.76 | 0.10 | 0.20 ±0.00 | 65.90 ±1.48 | 61.99 ±0.17 | 19.62 ±1.12 | 3.16 | 53.60 ±0.06 | **71.18 ±0.13** |
| q-FFL | 24.97 ±0.46 | 211.41 ±6.11 | 0.12 | 0.00 ±0.00 | 45.04 ±0.53 | 62.42 ±0.46 | 15.44 ±1.89 | 4.04 | 54.13 ±0.11 | 70.01 ±0.09 |
| AFL | 20.84 ±0.43 | **128.14 ±4.53** | 0.16 | **4.03 ±0.14** | 50.83 ±0.30 | 62.09 ±0.53 | 16.47 ±0.88 | 3.77 | 54.65 ±0.64 | 68.83 ±1.30 |
| FedFV | 31.23 ±0.04 | 306.25 ±0.70 | 0.10 | 0.20 ±0.00 | 66.05 ±0.11 | 62.13 ±0.08 | 15.69 ±0.58 | 3.96 | 53.92 ±0.30 | 69.60 ±0.31 |
| FedMGDA+ | 31.34 ±0.12 | 275.89 ±9.63 | 0.11 | 0.74 ±0.12 | 65.21 ±1.15 | 62.33 ±0.26 | 17.49 ±0.31 | 3.56 | 53.77 ±0.16 | 70.04 ±0.30 |
| TERM | 28.98 ±0.45 | 295.50 ±4.47 | 0.10 | 0.37 ±0.02 | 63.85 ±0.40 | 61.29 ±0.37 | 19.36 ±0.94 | 3.17 | 52.92 ±0.65 | 69.82 ±0.44 |
| AAggFF | 31.05 ±0.04 | 285.95 ±9.81 | 0.11 | 0.83 ±0.07 | 66.10 ±0.36 | 62.16 ±0.22 | 16.33 ±1.31 | 3.81 | 54.35 ±0.30 | 69.97 ±0.42 |
| Prac-FedEBA+ | 31.95 ±0.12 | 231.95 ±2.74 | 0.14 | 1.05 ±0.25 | 67.20 ±0.03 | 63.43 ±0.56 | 15.13 ±0.48 | 4.19 | 54.38 ±0.67 | 70.15 ±0.33 |
| FedEBA+ | **31.98 ±0.30** | 189.06 ±4.40 | **0.17** | 1.12 ±0.05 | **67.94 ±0.54** | **63.75 ±0.09** | **13.89 ±0.72** | **4.59** | **55.64 ±0.18** | 70.93 ±0.22 |

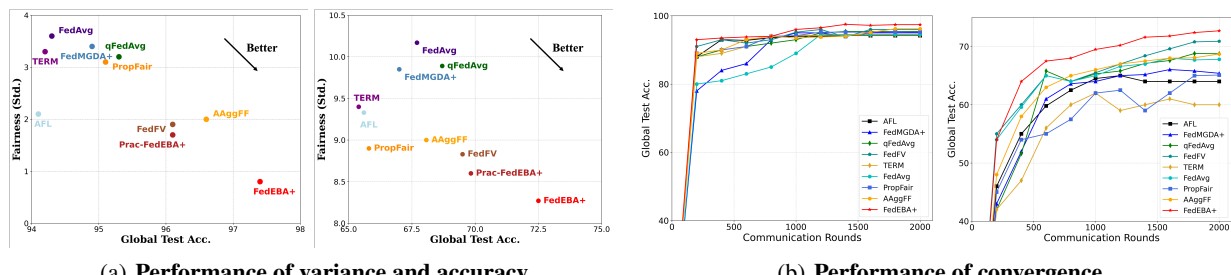

(a) **Performance of variance and accuracy**          (b) **Performance of convergence**

*Figure 2.* Performance of algorithms on (a) left: variance and accuracy on MNIST, (a) right: variance and accuracy on CIFAR-10, (b) left: convergence on MNIST, (b) right: convergence on CIFAR-10.

effectiveness of the algorithm to collapse, and it is relatively stable.

To further reduce the need for manual tuning of $\tau$, we also consider dynamic annealing schedules over communication rounds. Starting from an initial temperature $\tau_0$, $\tau$ is gradually decayed so that training first explores a smoother aggregation landscape and then progressively enforces stricter fairness. We evaluate three schedules:

$$\text{Linear: } \tau_t = \frac{\tau_0}{1+t},$$
$$\text{Concave: } \tau_t = \frac{\tau_0}{(1+t)^{1/2}},$$
$$\text{Convex: } \tau_t = \frac{\tau_0}{(1+t)^3}.$$

As shown in Fig. 5 in the Appendix, these annealing schedules reduce the need for careful temperature tuning and consistently yield strong fairness performance.

**Robustness and Privacy Evaluation.** To evaluate the robustness of FedEBA+ under severe label corruption, we further consider a noisy-label setting, following (Jiang et al., 2022; Fang & Ye, 2022), where local labels are corrupted by symmetric flipping with noise ratio $\epsilon = 0.5$. All other

hyperparameters are kept consistent with the main experimental settings. We also combine several methods with Local Self-Regularization (LSR) (Jiang et al., 2022) to examine whether FedEBA+ remains compatible with existing noise-robust training strategies.

The results in Table 3 show that FedEBA+ remains robust under corrupted inputs. Without LSR, FedEBA+ achieves the highest global accuracy and the lowest variance on both FashionMNIST and CIFAR-10. With LSR, FedEBA+ further improves global accuracy while maintaining the best fairness. These results suggest that FedEBA+ is robust to severe label corruption and can be effectively integrated with noise-robust regularization techniques to further enhance performance under corrupted inputs.

Figure 4 further indicates that FedEBA+ is compatible with differential privacy methods without significant performance degradation. Additional details are provided in Appendix N.

**Additional Results.** Due to page limits, we demonstrate the superiority of FedEBA+ via more experimental results in Appendix N, including:

- Our approach is orthogonal to advanced optimization methods such as momentum (Table 10) and VARP

*Table 3.* **Corrupted inputs evaluation on FashionMNIST and CIFAR-10.** We evaluate the robustness of different algorithms under severe label corruption, where local noisy labels follow symmetric flipping with noise ratio $\epsilon = 0.5$. We report the accuracy of the global model, variance fairness, $C_V$, worst 5%, and best 5% accuracy, where $C_V = \text{Global Acc.}/\text{Var.}$ and a larger value indicates a better accuracy-fairness trade-off. We highlight the best and the second-best results within each group by using **bold font** and blue text.

| Algorithm | FashionMNIST | | | | | CIFAR-10 | | | | |
|---|---|---|---|---|---|---|---|---|---|---|
| | Global Acc. ↑ | Var. ↓ | $C_V$ ↑ | Worst 5% ↑ | Best 5% ↑ | Global Acc. ↑ | Var. ↓ | $C_V$ ↑ | Worst 5% ↑ | Best 5% ↑ |
| FedAvg | 80.59 ±0.42 | 57.34 ±2.98 | 1.41 | 65.40 ±0.43 | 94.87 ±0.25 | 33.45 ±0.89 | 38.03 ±2.30 | 0.88 | 21.67 ±0.96 | 46.27 ±1.65 |
| q-FFL | 79.85 ±0.31 | 68.00 ±4.34 | 1.17 | 64.13 ±0.75 | 95.47 ±0.19 | 30.83 ±0.76 | 44.46 ±2.76 | 0.69 | 17.21 ±1.03 | 44.33 ±0.19 |
| AFL | 80.34 ±0.35 | 57.35 ±6.06 | 1.40 | 65.60 ±2.01 | 95.00 ±0.91 | 32.64 ±0.33 | 35.58 ±3.17 | 0.92 | 20.47 ±0.82 | 44.80 ±1.61 |
| FedFV | 63.08 ±0.88 | 88.95 ±3.06 | 0.71 | 46.13 ±0.77 | 83.13 ±1.52 | 34.28 ±0.39 | 41.07 ±0.77 | 0.83 | 21.13 ±0.90 | 46.60 ±0.33 |
| FOCUS | 80.79 ±0.27 | 58.61 ±3.61 | 1.38 | 64.40 ±1.85 | 94.80 ±0.62 | 26.81 ±1.22 | 44.04 ±0.68 | 0.61 | 6.84 ±1.58 | 56.69 ±1.22 |
| AAggFF | 81.44 ±0.63 | 55.34 ±3.83 | 1.47 | 66.15 ±0.83 | 96.06 ±0.75 | 34.65 ±0.51 | 35.22 ±2.12 | 0.98 | 22.59 ±1.10 | 46.88 ±0.87 |
| FedEBA+ | 82.03 ±0.42 | 49.23 ±7.21 | 1.67 | 67.67 ±1.06 | 95.27 ±0.81 | 35.04 ±0.21 | 34.60 ±3.69 | 1.01 | 23.07 ±1.24 | 47.80 ±1.23 |
| FedAvg (+ LSR) | 84.36 ±0.07 | 57.80 ±5.71 | 1.46 | 69.20 ±0.75 | 96.87 ±0.34 | 58.90 ±0.42 | 80.80 ±8.73 | 0.73 | 40.80 ±0.75 | 76.93 ±1.24 |
| q-FFL (+ LSR) | 84.23 ±0.08 | 63.69 ±1.62 | 1.32 | 64.73 ±0.09 | 96.87 ±0.41 | 58.91 ±0.75 | 86.32 ±10.20 | 0.68 | 41.33 ±0.90 | 77.60 ±2.73 |
| AAggFF (+ LSR) | 85.13 ±0.32 | 56.85 ±2.59 | 1.50 | 66.39 ±0.60 | 96.56 ±0.40 | 60.60 ±0.39 | 75.49 ±1.25 | 0.80 | 42.75 ±0.98 | 76.16 ±2.36 |
| FedEBA+ (+ LSR) | 85.30 ±0.12 | 54.10 ±4.13 | 1.58 | 67.93 ±0.62 | 96.80 ±0.28 | 61.21 ±0.88 | 64.73 ±0.97 | 0.95 | 43.40 ±1.72 | 75.53 ±2.05 |

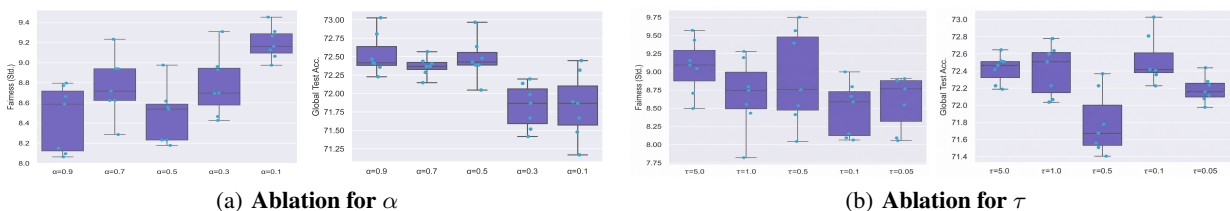

(a) **Ablation for $\alpha$**         (b) **Ablation for $\tau$**

*Figure 3.* Ablation study for hyperparameters

(Table 11), enabling seamless integration.

- The superiority of our method in terms of the additional fairness indicators (e.g., cosine similarity and entropy metrics (Table 16).

- Experimental analysis and discussion on the Dirichlet parameter (non-iid-ness), and annealing strategies of $\tau$ (Table 12, Figure 6, and Figure 5, respectively).

- Scalability of FedEBA+ (different network structure depths and widths) in Table 17 and 18.

- The improvement rates achieved by FedEBA+ can be higher when increasing the local update step size $K$. Specifically, as shown in Table 9, the improvement rates achieved by FedEBA+ over FedAvg are 33%, 33.4%, and 34.7% when the local step size $K$ is set to 1, 5, and 10, respectively. This increasing trend unequivocally demonstrates that higher $K$ values benefit FedEBA+ disproportionately more than FedAvg in reducing variance (enhancing fairness).

- Our method can be adapted to different FL tasks, which is shown in the experiments on the non-vision dataset(Table 19). We evaluate the fairness algorithms on the language modeling task Reddit (Caldas et al., 2018), and the results show that our algorithm can

achieve the best fairness and competitive global performance across various types of tasks.

## 7  Conclusions, Limitations and Future Work

This paper introduces FedEBA+, a framework that improves client-level fairness while maintaining strong global accuracy, incorporating an entropy-based aggregation method and adaptive alignment strategy. Theoretical analysis confirms its convergence in non-convex settings, and empirical results show it outperforms existing methods in client-level fairness and global accuracy.

Future work will also investigate extending FedEBA+ to model-heterogeneous FL. While the current gradient alignment step assumes homogeneous architectures, the entropy-based aggregation weights are architecture-agnostic since they depend only on scalar client losses. A promising extension is to replace parameter-space alignment with logit-space alignment via federated knowledge distillation, where clients align their predictions to a fair global logit target aggregated on a public reference set.

## Acknowledgments

This work is supported in part by the Guangdong Basic and Applied Basic Research Foundation under Grant No. 2025A1515012968, in part by the Shenzhen Science and Technology Program under Grant No. JCYJ20240813113502004, in part by the National Natural Science Foundation of China under Grant No. 62001412, in part by Shenzhen Stability Science Program 2023, in part by the Guangdong Provincial Key Laboratory of Future Networks of Intelligence (Grant No. 2022B1212010001), and in part by the Shenzhen Key Lab of Crowd Intelligence Empowered Low-Carbon Energy Network (Grant No. ZDSYS20220606100601002).

## Impact Statement

This paper introduces FedEBA+, a framework that improves client-level fairness while maintaining strong global performance in Federated Learning. By integrating the Maximum Entropy Principle with gradient alignment, the research ensures equitable algorithmic benefits across diverse users without sacrificing model accuracy. This approach mitigates performance discrimination against minority data distributions and enhances the inclusivity and trustworthiness of decentralized AI in critical sectors like healthcare and finance, providing a principled foundation for more ethical and pervasive distributed intelligence.

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

# Contents of Appendix

## A  Remark Motivation and Contribution

### A.1  Motivation: Re-examining Fairness and Entropy in Federated Learning

A central challenge in Federated Learning (FL) stems from the inherent data heterogeneity across clients. This heterogeneity often leads to significant performance disparities, where a global model may perform well for some clients but poorly for others, raising critical fairness concerns. While numerous methods have been proposed to address this, many rely on heuristics or introduce a trade-off between fairness and overall model accuracy.

Furthermore, a common misconception pervades the use of information-theoretic tools like entropy to promote fairness. A frequent mistake is to equate entropy directly with fairness. For instance, standard entropy maximization, subject only to the constraint that aggregation weights sum to one ($\max H(p)$ s.t. $\sum p_i = 1$), invariably yields uniform weights ($p_i = 1/N$). In heterogeneous settings, this is the aggregation scheme of FedAvg, which is precisely the source of the unfairness problem.

This motivates us to ask a foundational question: What is the proper role of entropy in achieving fairness? We argue that **entropy itself is not fairness; rather, constrained entropy maximization is a principled tool to achieve fairness with minimal bias.** Our motivation is to clarify this crucial distinction and leverage it to build a novel FL framework that ensures equity without compromising performance.

### A.2 Contribution: FedEBA+, A Framework for Principled Fairness via Constrained Entropy

To address the aforementioned challenges, we propose FedEBA+, the **first entropy-based method in Federated Learning explicitly designed for fairness.** Our core contributions are as follows:

1. **A Principled Fairness Constraint:** Unlike unconstrained approaches, FedEBA+ introduces an explicit fairness constraint into the optimization process. This constraint formally encodes the fairness objective—for instance, requiring the expected client performance to match an ideal, equitable target ($\sum p_i f(x_i) = f_{\text{ideal}}$). This ensures that the resulting aggregation is guaranteed to satisfy the desired fairness criterion.

2. **The Least Biased Fair Solution:** Among all possible weight distributions that satisfy the fairness constraint, FedEBA+ selects the one with the maximum entropy. According to the principle of maximum entropy, this choice represents the **least biased solution**, as it makes the fewest assumptions beyond the established constraint. In practice, this means our method adjusts client weights (e.g., by upweighting underperformers) *only as needed* to meet the fairness target, thus avoiding unnecessary or excessive intervention.

3. **Transforming Fairness into a Principled Optimization Problem:** Our framework transforms the abstract goal of fairness into a concrete, solvable, constrained optimization problem. This principled approach avoids reliance on ad-hoc heuristics, making the enforcement of fairness more robust and generalizable. As demonstrated in our experiments, this method successfully reduces performance variance among clients by 37% while maintaining high model accuracy, effectively breaking the fairness-performance trade-off.

In summary, our work establishes a principled connection between entropy and fairness. We clarify that fairness is not achieved by simply maximizing entropy, but by using entropy as a tool to enforce equity with minimal assumptions. By transforming fairness goals into an equitable aggregation scheme, FedEBA+ offers a robust and effective solution to one of the most pressing challenges in Federated Learning.

## B An Expanded Version of The Related Work

**Fairness-Aware Federated Learning.** Various fairness concepts have been proposed in FL, including performance fairness (Li et al., 2019a; 2021; Wang et al., 2021; Zhao & Joshi, 2022; Kanaparthy et al., 2022; Huang et al., 2022), group fairness (Du et al., 2021; Ray Chaudhury et al., 2022), selection fairness (Zhou et al., 2021), and contribution fairness (Cong et al., 2020), among others (Shi et al., 2023; Wu et al., 2022; Chen et al., 2023). These concepts address specific aspects and stakeholder interests, making direct comparisons inappropriate. This paper specifically focuses on performance fairness, the most commonly used metric in FL, which serves client interests while improving model performance. We list and compare the commonly used fairness metrics of FL in the next section, i.e., Section C.

Some works propose objective function-based approaches to enhance performance fairness for FL. In (Li et al., 2019a), q-FFL uses $\alpha$-fair allocation for balancing fairness and efficiency, but specific $\alpha$ choices may introduce bias. In contrast, FedEBA+ employs maximum entropy aggregation to accommodate diverse preferences. Additionally, FedEBA+ introduces a novel fair FL objective with dual-variable optimization, enhancing global model performance and variance. Besides, (Deng et al., 2020) achieves fairness by defining a min-max optimization problem in FL. In the gradient-based approach, FedFV (Wang et al., 2021) mitigates gradient conflicts among FL clients to promote fairness, but it consumes much computational and storage resources. Efforts have been made to connect fairness and personalized FL to enhance robustness (Li et al., 2021; Lin et al., 2022), different from our goal of learning a valid global model to guarantee fairness.

FOCUS (Chu et al., 2023) introduces the *Fairness via Agent-Awareness* (FAA) metric, quantifying the maximum discrepancy in excess loss across agents. Utilizing an Expectation Maximization (EM) algorithm, FOCUS achieves soft clustering of clients. However, it involves communication between all clients and the server, with each client requiring all cluster models, resulting in elevated communication and computation costs. Although addressing FAA is not our primary focus, we illustrate that FedEBA+ remains effective and outperforms FOCUS in both variance and FAA in our experimental setting, as detailed in Table 1 and Table 15. Notably, our method operates without imposing data distribution or model class assumptions, distinguishing it from existing work (Chu et al., 2023) that relies on the distance disparity of local loss and ideal loss as a fairness measure. The use of variance in performance fairness naturally aligns with the goal of ensuring uniform performance across clients. Compared to TERM(Li et al., 2020a), which also employs exponential weighting to address underperforming clients, FedEBA+ further improves overall performance while explicitly promoting fairness through a bi-level optimization framework. FedSRCVaR and FedMinMax (Papadaki et al., 2022; 2024), which target group fairness, develop effective algorithms based on the Lagrange dual of the minimax formulation—highlighting the broader effectiveness of bi-level optimization in fairness-aware learning. In contrast, FedEBA+ focuses on client-level fairness and utilizes a closed-form inner-loop solution that significantly reduces computational overhead. Similarly, Scaff-PD(Yu et al., 2024) adopts a min-max formulation for fairness improvement. Finally, unlike FLRA (Reisizadeh et al., 2020), which addresses model heterogeneity via affine shift correction, FedEBA+ introduces a lightweight, fairness-aware mechanism without introducing additional parameter overhead. Recently, reweighting methods encourage a uniform performance by up-reweighting the importance of underperforming clients (Zhao & Joshi, 2022; Mollanejad et al., 2024). However, these methods enhance fairness at the expense of the performance of the global model (Kanaparthy et al., 2022; Huang et al., 2022). In contrast, we propose FedEBA+ as a solution that significantly promotes fairness while improving the global model performance. Notably, FedEBA+ is orthogonal to existing optimization methods like momentum (Karimireddy et al., 2020a) and VARP (Jhunjhunwala et al., 2022), allowing seamless integration, as shown in Table 10 and Table 11.

Recently, several federated learning studies have explored a diverse range of fairness objectives, such as Proportionality (Chaudhury et al., 2024; Ray Chaudhury et al., 2022), Disparity (Hamman & Dutta), Stability (Gao et al.), and fairness in vertical FL (Fan et al.; Qi et al., 2022). (Chaudhury et al., 2024) provides explainable proportional fairness guarantees to the agents in general settings in which the error rates of the agents are proportional to the size of their local data, and (Ray Chaudhury et al., 2022) proposes a core-stability as fairness metric that is more resilient to noisy data from certain clients. The used fairness is sensitive to data, while ours focuses on performance fairness for clients, regarding the data distribution, thus the objective is different. (Hamman & Dutta) offers an information-theoretic perspective on group fairness trade-offs in federated learning, utilizing partial information decomposition to identify unfairness. (Gao et al.) mainly focuses on establishing a theoretical bound for showing the influence of clients' altruistic behaviors and the configuration of the friend-relationship network on the achievable egalitarian fairness. These works aim to establish the theoretical bound for analyzing the fairness and trade-offs, from an information perspective and game theory, instead of providing a fair algorithm. (Fan et al.; Qi et al., 2022) discuss fairness in vertical FL by learning fair and unified representations, where feature fields are decentralized across different platforms. In contrast, our work focuses on horizontal FL and compares our results with state-of-the-art horizontal FL fairness algorithms.

**Aggregation in Federated Optimization.** FL employs aggregation algorithms to combine decentralized data for training a global model (Kairouz et al., 2019). Approaches include federated averaging (FedAvg) (McMahan et al., 2017), robust federated weighted averaging (Pillutla et al., 2019; Laguel et al., 2021; Pillutla et al., 2023), importance aggregation (Wang et al., 2022), and federated dropout (Zheng et al., 2022). However, these algorithms can be sensitive to the number and quality of participating clients, causing fairness issues (Li et al., 2019b; Balakrishnan et al., 2021; Shi et al., 2023). To the best of our knowledge, we are the first to analyze the aggregation from the view of entropy. Unlike heuristics that assign weights proportional to client loss (Zhao & Joshi, 2022; Kanaparthy et al., 2022), our method has physical meanings, i.e., the aggregation probability ensures that known constraints are as certain as possible while retaining maximum uncertainty for unknowns. By selecting the maximum entropy solution with constraints, we actually choose the solution that fits our information with the least deviation (Jaynes, 1957), thus achieving fairness. AAggFF (Hahn et al., 2024) proposes an aggregation strategy based on sequential decision-making, which shares a key principle with ours: assigning higher weights to underperforming clients can enhance fairness in federated learning. This provides further empirical support for entropy-based methods, as both approaches share a conceptually similar formulation. In contrast, our method is theoretically grounded in a constrained maximum entropy optimization model. Moreover, beyond aggregation, we introduce an alignment update (Sections 4.2–4.3), which simultaneously improves client-level fairness and global model performance. These distinctions position our approach as a novel and principled extension of the paradigm, offering greater interpretability and

broader applicability in heterogeneous FL settings.

Our proposed aggregation method differs from existing approaches in several key aspects. First, the aggregation formulation is novel, with probabilities $p_i = e^{\frac{F_i(x)/\tau}{Z}}$ proportional to the exponential of client loss and regulated by a controllable parameter $\tau$. Unlike heuristic methods that assign weights directly proportional to client loss $p_i \propto F_i(x)$ (Mollanejad et al., 2024; Zhao & Joshi, 2022; Kanaparthy et al., 2022), our approach is derived from a constrained optimization framework. Second, the objective is fundamentally different. Existing entropy-based aggregation methods (Huang et al., 2022; Herath et al., 2024) and softmax-based reweighting approaches (Zhao & Joshi, 2022; Kanaparthy et al., 2022) aim to enhance model accuracy without addressing fairness, whereas our approach focuses explicitly on improving fairness. Third, our method introduces a novel constrained entropy model, the first of its kind in the FL fairness community, which prioritizes underperforming clients to achieve weighted fair aggregation. Furthermore, our approach offers practical advantages, such as its exponent form and control parameter $\tau$, which effectively mitigate extreme unfairness and allow flexibility in recovering existing aggregation methods like FedAvg, AFL, and q-FFL. Empirically, our entropy-based aggregation (FedEBA+ with $\alpha = 0$ ) outperforms state-of-the-art methods like q-FFL and TERM, achieving superior results in both fairness and accuracy.

**FL others.** In addition to fairness algorithms, FL faces other challenges such as privacy preservation (Wang et al., 2023; Zhou et al., 2023; Chen et al., 2023) and communication efficiency (Chai et al., 2023; Almanifi et al., 2023; Paragliola & Coronato, 2022). Given the widespread adoption of FL, our primary focus in this work is on designing a high-performance fairness algorithm. Nonetheless, we acknowledge the significance of other aspects in FL, such as privacy preservation. Hence, we provide experimental results demonstrating the compatibility of our algorithm with existing privacy protection methods and its robustness to external noise scenarios.

## C   Discussion of fairness metrics

In this section, we summarize the commonly used definitions of fairness metrics and comment on their advantages and disadvantages.

Euclidean Distance and person correlation coefficient are usually used for contribution fairness, and risk difference and Jain's fairness Index are usually used for group fairness, which is a different target from performance fairness in this paper. In particular, cosine similarity and entropy play roles similar to variance, used to measure the performance distribution among clients. The more uniform the distribution, the smaller the variance and the more similar to vector $1$. The larger the entropy of the normalized performance, the more similar to vector $1$. Thus, for performance fairness, we only need one of them. We use variance, which is the most widely used metric in related works.

The detailed discussion of each metric is shown below:

- **Variance**, applied in accuracy parity and performance fairness scenarios, is valued for its simplicity and straightforward implementation, focusing on a common performance metric. However, it has a limitation as it only measures relative fairness, making it sensitive to outliers (Zafar et al., 2017; Li et al., 2019a; 2021; Hu et al., 2022; Shi et al., 2023).

- **Cosine similarity**, sharing applications with variance, is known for its similarity to variance and the ease with which it captures linear relationships (Li et al., 2019a). Nevertheless, it falls short when it comes to capturing magnitude differences and is sensitive to zero vectors (Selbst et al., 2019; Hardt et al., 2016).

- Also utilized in scenarios akin to variance, **entropy** offers simplicity but has dependencies on normalization and sensitivity to the number of clients involved in the computation, making it less robust in certain situations (Li et al., 2019a; Selbst et al., 2019; Hardt et al., 2016).

- Applied in contribution fairness, **Euclidean distance** provides a straightforward interpretation and is sensitive to magnitude differences. However, it lacks consideration for the direction of the differences, limiting its overall effectiveness.

- In contribution fairness scenarios, the **Pearson correlation coefficient** is appreciated for its scale invariance and ability to capture linear relationships (Jia et al., 2019). Yet, it may be sensitive to outliers and may not accurately capture magnitude differences, assuming a linear relationship between the data variables (Wang et al., 2019).

- Commonly used in group fairness contexts, **risk difference** is sensitive to group disparities and offers interpretability (Du et al., 2021). However, it lacks normalization, which can impact its effectiveness in certain scenarios (Dwork et al., 2012).

- **Jain's Fairness Index** finds application in various fairness aspects, including group fairness, selection fairness, performance fairness, and contribution fairness. It boasts normalization across groups and flexibility in handling various

metrics. Nevertheless, it is sensitive to metric choice and introduces complexity in interpretability (Chiu, 1984; Liu et al., 2022).

## D  Entropy Analysis

### D.1  Derivation of Proposition 4.1

In this section, we derive the maximum entropy distribution for the aggregation strategy employed in FedEBA+.

The choice of an exponential formula treatment for the loss function, represented as $p_i \propto e^{F_i(x)/\tau}$, is motivated by our adherence to a maximum entropy distribution. This approach is favored over alternatives such as $p_i \propto F_i(x)$ because our aggregation strategy is designed to achieve maximum entropy.

Maximizing entropy minimizes the incorporation of prior information into the distribution, ensuring that the selected probability distribution is free from subjective influences and biases (Bian et al., 2021; Sampat & Zavala, 2019). Simultaneously, this aligns with the tendency of many physical systems to evolve towards configurations with maximal entropy over time (Jaynes, 1957).

In the following we will give a derivation to show that $p_i \propto e^{F_i(x_i)/\tau}$ is indeed the maximum entropy distribution for FL. The derivation below is closely following (Jaynes, 1957) for statistical mechanics. Suppose the loss function of the user corresponding to the aggregation probability $p_i$ is $F_i(x_i)$. We would like to maximize the entropy $\mathbb{H}(p_i) = -\sum_{i=1}^{m} p_i \log p_i$, subject to FL constrains that $\sum_{i=1}^{m} p_i = 1, p_i \geq 0, \sum_i p_i F_i(x_i) = \tilde{f}(x)$, which means we constrain the reweighted clients' performance to be close to ideal model's performance, such as ideal global model performance or the ideal fair performance.

*Proof.*

$$L\left(p, \lambda_0; \frac{1}{\tau}\right) := -\left[\sum_{i=1}^{N} p_i \log p_i + \lambda_0 \left(\sum_{i=1}^{N} p_i - 1\right) + \frac{1}{\tau}\left(\mu - \sum_{i=1}^{N} p_i F_i(x_i)\right)\right], \tag{16}$$

where $\mu = \tilde{f}(x)$.

By setting

$$\frac{\partial L\left(p, \lambda_0; \frac{1}{\tau}\right)}{\partial p_i} = -\left[\log p_i + 1 + \lambda_0 - \frac{1}{\tau}F_i(x_i)\right] = 0, \tag{17}$$

we get:

$$p_i = \exp\left[-\left(\lambda_0 + 1 - \frac{1}{\tau}F_i(x_i)\right)\right]. \tag{18}$$

According to $\sum_i p_i = 1$, we have:

$$\lambda_0 + 1 = \log \sum_{i=1}^{N} \exp\left(\frac{1}{\tau}F_i(x_i)\right) =: \log Z, \tag{19}$$

which is the log-partition function.

Thus, we reach the exponential form of $p_i$ as:

$$p_i = \frac{\exp\left[F_i(x_i)/\tau\right]}{\sum_{j=1}^{N} \exp(F_j(x_j)/\tau)}. \tag{20}$$

$\square$

When taking into account the prior distribution of aggregation probability (Li et al., 2020b; Balakrishnan et al., 2021), which is typically expressed as $q_i = n_i/\sum_{i \in S_t} n_i$, the original entropy formula can be extended to include the prior distribution as

follows:

$$H(p_i) = \sum_{i=1}^{m} p_i \log(\frac{q_i}{p_i}) \,. \tag{21}$$

Thus, the solution of the original problem under this prior distribution becomes:

$$p_i = \frac{q_i \exp[F_i(x_i)/\tau]}{\sum_{j=1}^{N} q_j \exp[F_j(x_i)/\tau]} \,. \tag{22}$$

*Proof.*

$$L\left(p, \lambda_0; \frac{1}{\tau}\right) := -\sum_{i=1}^{N} p_i \log \frac{q_i}{p_i} + \lambda_0 \left(\sum_{i=1}^{N} p_i - 1\right) + \frac{1}{\tau}\left(\mu - \sum_{i=1}^{N} p_i F_i(x_i)\right) \,. \tag{23}$$

Following similar derivation steps, let

$$\frac{\partial L\left(p, \lambda_0; \frac{1}{\tau}\right)}{\partial p_i} = -\log(q_i) + \log(p_i) + 1 + \lambda_0 - \frac{1}{\tau}F_i(x_i) = 0 \,, \tag{24}$$

we get:

$$p_i = \exp\left[-\left(\lambda_0 + 1 - \log(q_i) - \frac{1}{\tau}F_i(x_i)\right)\right] \,. \tag{25}$$

According to $\sum_i p_i = 1$, we have:

$$\sum_i p_i = \sum_i \exp\left[-\left(\lambda_0 + 1 - \log(q_i) - \frac{1}{\tau}F_i(x_i)\right)\right] = 1 \,. \tag{26}$$

Therefore, we get:

$$\lambda_0 + 1 = \log \sum_{i=1}^{N} q_i \exp\left(\frac{1}{\tau}F_i(x)\right) =: \log(Z) \,. \tag{27}$$

Then substituting $\lambda_0 + 1 = \log(Z)$ back to $p_i = \exp\left[-\left(\lambda_0 + 1 - \log(q_i) - \frac{1}{\tau}F_i(x_i)\right)\right]$, we obtain (22):

$$p_i = \frac{q_i \exp[F_i(x_i)/\tau]}{\sum_{j=1}^{N} q_j \exp[F_j(x_i)/\tau]} \,. \tag{28}$$

$\square$

## E    Enhancing Robustness in FedEBA+ through Local Self-Regularization

In this section, we introduce Local Self-Regularization (LSR) for FedEBA+ as a robustness solver.

**Remark E.1** (Robustness of EBA). *Typical aggregation methods focusing on fairness or heterogeneity often suffer significant performance degradation in scenarios with noisy labels (Pillutla et al., 2019; Yang et al., 2022; Xu et al., 2022). We demonstrate that our aggregation method maintains robustness to noisy labels by extending the local loss $F_i(x)$ to a* robust *loss $F_i^r(x)$. The aggregation then becomes:*

$$p_i = \frac{\exp\left(F_i^r(x)/\tau\right)}{\sum_j \exp\left(F_j^r(x)/\tau\right)};$$
$$F_i^r(x) = \mathbb{E}_{\xi_i}\left[F_i^{cls}(x; \xi_i) + \gamma F_i^{reg}(x; Augment(\xi_i))\right] \,, \tag{29}$$

*where $F_i^{cls}(x; \xi_i)$ represents the cross-entropy loss, and $F_i^{reg}(x; Augment(\xi_i))$ denotes the self-distillation loss with augmented data. The robust loss mitigates model output discrepancies between original and mildly augmented instances, addressing noisy label scenarios and enhancing robustness.*

The method is primarily based on the work of (Jiang et al., 2022). For the sake of completeness in this paper, we restate the LSR algorithm here. The LSR algorithm effectively regulates the local training process by implicitly preventing the model from memorizing noisy labels. Additionally, it explicitly narrows the model output discrepancy between original and augmented instances through self-distillation.

---

**Algorithm 2** Local Self-Regularization

1: **for** client $i$ in parallel **do**
2:     **Input:** client $i$, global model $x_t$, parameter $\gamma$, $\lambda \sim Beta(1,1)$.
3:     **Output:** local trained model $x_i^{t+1}$.
4:     **Initialize:** $x_i^{t,0} \leftarrow x_t$.
5:     **for** $k = 0, \cdots, K-1$ **do**
6:         $p_1, p_2 = Softmax(F_i(x_i^{t,k}; \xi_i)), Softmax(F(x_i^{t,k}; Augment(\xi_i)))$;
7:         $p = \lambda p_1 + (1-\lambda)p_2$;
8:         $p_{s,c} = \frac{p_c^{1/T_s}}{\sum_j p_j^{1/T_s}}$, where $c$ denotes the $c$-th class, and $T_s$ is the sharpening temperature;
9:         $F^{cls} = CorssEntropy(p_s, y)$;
10:        $F^{reg} = SelfDistillation(F(x_i^{t,k}; \xi_i), F(x_i^{t,k}; Augment(\xi_i)))$;
11:        $F_i^r = F^{cls} + \gamma F^{reg}$;
12:        Update $x_{t+1}^i$ with $F_i^r$;
13:     **end for**
14: **end for**

---

For the regression loss, self-distillation is performed on the network. We use the two output logits $\xi_i$ and $Augment(\xi_i)$ to conduct instance-level self-distillation. First, apply a softmax function with a distillation parameter $T_d$ to the output as:

$$q_{1,i}, q_{2,i} = \frac{\exp([F(x_i^{t,k}; \xi_i)]_c/T_d)}{\sum_j \exp([F(x_i^{t,k}; \xi_i)]_j/T_d)}, \frac{\exp([F(x_i^{t,k}; Augment(\xi_i))]_c/T_d)}{\sum_j \exp([F(x_i^{t,k}; Augment(\xi_i))]_j/T_d)}, \tag{30}$$

where $c$ and $j$ denote the output logits for the $c$-th and $j$-th class, respectively. The self-distillation loss term is formulated as:

$$F^{reg} = \frac{1}{2}(\text{KL}(q_1 \| U) + \frac{1}{2}(\text{KL}(q_2 \| U)), \tag{31}$$

where KL means Kullback-Leibler divergence and $U = \frac{1}{2}(q_1 + q_2)$.

In this way, we can express the *robust EBA* method by:

$$p_i = \frac{\exp(F_i^r(x)/\tau)}{\sum_j \exp(F_j^r(x)/\tau)}, \quad F_i^r(x) = \mathbb{E}_{\xi_i}\left[F_i^{cls}(x; \xi_i) + \gamma F_i^{reg}(x; Augment(\xi_i))\right]. \tag{32}$$

We experimentally demonstrate the robustness of EBA in Table 3.

### E.1 Toy example of extremal case

In this subsection, we examine an extreme case as an illustrative example. Consider two clients: client 1 with noisy data and client 2 with separable data. Assume the test accuracy on client 1 is consistently zero or the loss is always high, denoted as $H_1$.

After local updates on each client, the model adjusts its parameters to minimize the noise. However, in the absence of an underlying pattern, the weights do not capture any meaningful relationship between features and labels. Consequently, the loss can be assumed to be $H_1$, and the model parameter as $x_1^t = x_i^{t+1}$ without loss of generality, as the model has no convergence point.

In contrast, assume client 2's model is $y = \frac{1}{2}x^2$, and starting from $x_2^t = 2$, it converges to $x_2^{t+1} = 0$. Thus, for FedEBA+, the updated model is $\tilde{x} = 0 + x_1^t \cdot e^{\frac{H_1}{H_1+0}}$. For FedAvg, the updated model is $\hat{x} = \frac{1}{2}x_1^t$. Since $|e \cdot x_1| \geq |\frac{1}{2}x_1|$, we have $y(\tilde{x}) \leq y(\hat{x})$. Consequently, we can assert that the disparity between client 1 and client 2 using EBA+ is smaller than with FedAvg.

---

**Algorithm 3** Prac-FedEBA+

---

1: **Input:** Number of clients $m$, global learning rate $\eta$, local learning rate $\eta_L$, number of local epoch $K$, total training rounds $T$.
2: **Output:** Final model parameter $x_T$.
3: **Initialize:** model $x_0$.
4: **for** round $t = 1, \ldots, T$ **do**
5:     Server selects a set of clients $|S_t|$ and broadcast model $x_t$.
6:     **for** each worker $i \in S_t$, in parallel **do**
7:         **for** $k = 0, \cdots, K - 1$ **do**
8:             $x_{t,k+1}^i = x_{t,k}^i - \eta_L \nabla F_i(x_{t,k}^i; \xi_i)$;
9:         **end for**
10:        $\Delta_t^i = x_{t,K}^i - x_{t,0}^i = -\eta_L \sum_{k=0}^{K-1} \nabla F_i(x_{t,k}^i; \xi_i)$;
11:     **end for**
12:     Server receive model updates $\Delta_t^i$ and clients' loss $\mathbf{L} = [F_1(x_t), \ldots, F_{|S_t|}(x_t)]$;
13:     Approximate fair gradient: $\tilde{g}^t = \sum_{i \in S_t} \frac{\exp[F_i(x_t)/\tau]}{\sum_{i \in S_t} \exp[F_i(x_t)/\tau]} \frac{1}{K} \sum_{k=0}^{K-1} \nabla F_i(x_{t,k}^i; \xi_i)$;
14:     Align model: $\hat{\Delta}_i^t = (1 - \alpha)\Delta_i^t - \alpha \eta_L K \tilde{g}^t$;
15:     Aggregation: $\Delta_t = \sum_{i \in S_t} p_i \hat{\Delta}_t^i$, where $p_i = \frac{\exp[F_i(x_{t,K}^i)/\tau)]}{\sum_{i \in S_t} \exp[F_i(x_{t,K}^i)/\tau]}$ ;
16:     Server update: $x_{t+1} = x_t + \eta \Delta_t$;
17: **end for**

---

Hence, we assert that even in the extreme case, FedEBA+ effectively reduces performance variance through the entropy-based aggregation method.

## F   Practical Algorithm with effective communication.

To achieve the same communication costs to FedAvg, we introduce a practical adaptation of FedEBA+ termed Prac-FedEBA+. Specifically, Prac-FedEBA+ leverages the last round's gradient to approximate current round information, reducing the need for extensive communication between the server and clients, as outlined in Algorithm 3.

## G   Analysis Comparison with existing works

*Table 4.* Convergence rate comparison of FedEBA+ with existing works.

| Algorithm | Convergence Upper Bound | Rate Order |
|---|---|---|
| FedAvg (Yang et al., 2021) | $\frac{1}{c}\left(\frac{f^0-f^*}{\sqrt{nKT}} + \frac{\sigma_L^2 + 3K\sigma_G^2}{2\sqrt{nKT}} + \frac{5(\sigma_L^2 + 6K\sigma_G^2)^2}{2KT} + \frac{15(\sigma_L^2 + 6K\sigma_G^2)}{2\sqrt{nKT^3}}\right)$ | $\mathcal{O}(\frac{1}{\sqrt{nKT}} + \frac{1}{T} + \frac{1}{\sqrt{nKT^3}})$ |
| FedIS (Chen et al., 2020) | $\frac{1}{c}\left(\frac{(f^0-f^*)B^2}{\sqrt{nKT}} + \frac{2F\sigma_L^2 + 2F(1-n/m)K\sigma_G^2}{2\sqrt{nKT}} + \frac{B^2F}{T} + \frac{F^{2/3}\sigma_G}{T^{2/3}}\right)$ | $\mathcal{O}(\frac{1}{\sqrt{nKT}} + \frac{1}{T} + \frac{1}{\sqrt{T^3}})$ |
| FedNova (Wang et al., 2020) | $\frac{1}{c}\left(\frac{(f^0-f^*)}{\sqrt{nKT}} + \frac{A\sigma_L^2 + \overline{\tau}/\tau_{eff}}{2\sqrt{nKT}} + \frac{mC\sigma_G^2}{\overline{\tau}T}\right)$ | $\mathcal{O}(\frac{1}{\sqrt{nKT}} + \frac{1}{T})$ |
| FedEBA+ | $\frac{1}{c}\left(\frac{f^0 - f^*}{\sqrt{nKT}} + \frac{(1-\alpha)^2 \sum_{i=1}^m w_i^2 \sqrt{m}\sigma_L^2 + \alpha^2 K^{-1/2}\sqrt{m}\rho^2}{2\sqrt{nKT}} \right.$ $\left. + \frac{5(1-\alpha)^2(\sigma_L^2 + 6K\sigma_G^2) + 15(1-\alpha)^2\alpha^2 K\rho^2}{2KT}\right)$ | $\mathcal{O}(\frac{\sqrt{K/n}}{\sqrt{nKT}} + \frac{1}{T})$ |

In this paper, the fairness and global model performance are analyzed via variance and convergence, respectively. The comprehensive analysis significantly improves upon existing research.

- For the variance analysis, all existing fairness works are typically evaluated by comparing them with FedAvg. However, our analysis expands beyond linear models to include the strongly convex setting.

- For the convergence analysis, beyond the strongly convex and convex settings, we demonstrate that our algorithms converge in nonconvex settings with a convergence rate no worse than the state-of-the-art FedAvg algorithm, as shown in the Table 4.

To explicitly demonstrate the importance of the paper's theoretical merit, we provide the following table to illustrate its contributions compared with other fairness works:

*Table 5.* Analysis Comparison of Different Fairness Algorithms

| Algorithm | Variance analysis | Convergence analysis |
|---|---|---|
| q-FFL | ✓ | ✗ |
| FedMGDA+ | ✗ | ✓ Strongly convex |
| TERM | ✓ Linear model | ✓ Strongly convex |
| AFL | ✗ | ✓ Convex |
| PropFair | ✗ | ✓ Nonconvex |
| lp-proj | ✓ Linear model | ✓ Nonconvex |
| FedEBA+ | ✓ Linear model & Strongly convex | ✓ Nonconvex |

The above comparison reveals that, among existing work, only FedEBA+ and lp-proj offer simultaneous variance and convergence analysis. In contrast to lp-proj:

- FedEBA+ expands fairness analysis from generalized linear regression models to strongly convex models.

- Moreover, lp-proj is a personalized FL algorithm, markedly distinct from ours, as this paper focuses on achieving a fair global model. Consequently, the convergence analysis and fairness analysis are distinct. Only FedEBA+ aims to improve the global model's performance and variance simultaneously, employing variance and convergence analyses, respectively.

## H Assumptions for Convergence Analysis

To facilitate the convergence analysis, we adopt the following commonly used assumptions in FL.

**Assumption 1** (L-Smooth). *There exists a constant $L > 0$, such that $\|\nabla F_i(x) - \nabla F_i(y)\| \leq L\|x - y\|, \forall x, y \in \mathbb{R}^d$, and $i = 1, 2, \ldots, m$.*

**Assumption 2** (Unbiased Local Gradient Estimator and Local Variance). *Let $\xi_t^i$ be a random local data sample in the round $t$ at client $i$: $\mathbb{E}\left[\nabla F_i(x_t, \xi_t^i)\right] = \nabla F_i(x_t), \forall i \in [m]$. There exists a constant bound $\sigma_L > 0$, satisfying $\mathbb{E}\|\nabla F_i(x_t, \xi_t^i) - \nabla F_i(x_t)\|^2 \leq \sigma_L^2$.*

**Assumption 3** (Bound Gradient Dissimilarity). *For any set of weights $\{w_i \geq 0\}_{i=1}^m$ with $\sum_{i=1}^m w_i = 1$, there exist constants $\sigma_G^2 \geq 0$ and $A \geq 0$ such that $\sum_{i=1}^m w_i \|\nabla F_i(x)\|^2 \leq (A^2 + 1) \|\sum_{i=1}^m w_i \nabla F_i(x)\|^2 + \sigma_G^2$.*

These assumptions are commonly used in both non-convex optimization and FL literature, see e.g. (Karimireddy et al., 2020b; Yang et al., 2021; Wang et al., 2020). For Assumption 3, if all local loss functions are identical, then $A = 0$ and $\sigma_G = 0$.

## I Convergence Analysis of FedEBA+

In this section, we give the proof of Theorem 5.1.

Before going to the details of our convergence analysis, we first state the key lemmas used in our proof, which helps us to obtain the advanced convergence result.

**Lemma I.1.** *To make this paper self-contained, we restate the Lemma 3 in (Wang et al., 2020):*

*For any model parameter $\boldsymbol{x}$, the difference between the gradients of $f_{avg}(\boldsymbol{x})$ and $f(\boldsymbol{x})$ can be bounded as follows:*

$$\|\nabla f_{avg}(\boldsymbol{x}) - \nabla f(\boldsymbol{x})\|^2 \leq \chi^2_{\boldsymbol{w}\|\boldsymbol{p}} \left[A^2 \|\nabla f(\boldsymbol{x})\|^2 + \chi^2_{\boldsymbol{w}\|\boldsymbol{p}}\right], \tag{33}$$

*where $\chi^2_{\boldsymbol{w}\|\boldsymbol{p}}$ denotes the chi-square distance between $\boldsymbol{w}$ and $\boldsymbol{p}$, i.e., $\chi^2_{\boldsymbol{w}\|\boldsymbol{p}} = \sum_{i=1}^m (w_i - p_i)^2 / p_i$. $f(x)$ is the global objective with $f(x) = \sum_{i=1}^m w_i f_i(x)$ where $\boldsymbol{w}$ is usually the data ratio of clients, i.e., $\boldsymbol{w} = [\frac{n_i}{N}, \cdots, \frac{n_i}{N}]$. $f(x) = \sum_{i=1}^m p_i f_i(x)$ is the objective function of FedEBA+ with the reweight aggregation probability $\boldsymbol{p}$.*

We first bound the $\chi_{w\|p}^2$ by our dynamically $\tau$ clamping.

The stability of the weighted loss function (e.g., $\chi^2$ divergence) depends on ensuring that the assigned probability weight $p_i$ for any sample $i$ does not vanish. Specifically, we require $p_i$ to be greater than a predefined numerical safety threshold $\epsilon_p$ (e.g., $10^{-6}$).

The probability $p_i$ is defined using a Softmax operation on the per-sample loss $F_i$, modulated by the temperature parameter $\tau$:

$$p_i = \frac{e^{F_i/\tau}}{\sum_{j=1}^{m} e^{F_j/\tau}}, \tag{34}$$

where $m$ is the selected clients for the current communication round.

The constraint is tightest for the sample exhibiting the minimum probability, $p_{min}$, which corresponds to the minimum loss,

$$p_{min} = \frac{e^{F_{min}/\tau}}{\sum_{j=1}^{m} e^{F_j/\tau}}. \tag{35}$$

To find a strict lower bound for $p_{min}$, we must use the maximum possible value for the denominator. Let $F_{max} = \max_j F_j$. The denominator is bounded by:

$$e^{F_{max}/\tau} \leq \sum_{j=1}^{m} e^{F_j/\tau} \leq m \cdot e^{F_{max}/\tau} \tag{36}$$

Substituting the conservative upper bound for the denominator into the stability requirement $p_{min} > \epsilon$:

$$\frac{e^{F_{min}/\tau}}{m \cdot e^{F_{max}/\tau}} > \epsilon \tag{37}$$

We now rearrange the inequality to isolate $\tau$.

Combine the exponential terms:

$$e^{(F_{min} - F_{max})/\tau} > m\epsilon$$

Take the natural logarithm (ln) of both sides:

$$\frac{F_{min} - F_{max}}{\tau} > \ln(m\epsilon)$$

Define the Loss Range $\Delta F = F_{max} - F_{min}$. Since $\Delta F \geq 0$, the numerator is $-\Delta F$. Also, for practical values of $m$ and $\epsilon$ (i.e., $m\epsilon \ll 1$), the right-hand side, $\ln(m\epsilon)$, is negative.

$$\frac{-\Delta F}{\tau} > \ln(m\epsilon)$$

Multiply both sides by $\tau$ (assuming $\tau > 0$):

$$-\Delta F > \tau \ln(m\epsilon)$$

Divide by $\ln(m\epsilon)$. Since $\ln(m\epsilon)$ is negative, we must reverse the inequality sign:

$$\tau > \frac{-\Delta F}{\ln(m\epsilon)}$$

Simplify the expression using $\ln(m\epsilon) = -\ln(1/m\epsilon)$:

$$\tau > \frac{F_{max} - F_{min}}{\ln\left(\frac{1}{m\epsilon}\right)}$$

The temperature parameter $\tau$ must be dynamically set to satisfy the derived lower bound, ensuring that the minimum probability $p_{min}$ is strictly greater than $\epsilon$ and, consequently, guaranteeing the numerical stability of the inverse probability weighting scheme.

$$\tau_{min} = \frac{F_{max} - F_{min}}{\ln\left(\frac{1}{m\epsilon}\right)}. \tag{38}$$

*Proof.*

$$\nabla f_{avg}(x) - \nabla f(\boldsymbol{x}) = \sum_{i=1}^{m} (w_i - p_i) \nabla f_i^{avg}(\boldsymbol{x})$$

$$= \sum_{i=1}^{m} (w_i - p_i) \left(\nabla f_i^{avg}(\boldsymbol{x}) - \nabla f(\boldsymbol{x})\right) \tag{39}$$

$$= \sum_{i=1}^{m} \frac{w_i - p_i}{\sqrt{p_i}} \cdot \sqrt{p_i} \left(\nabla f_i^{avg}(\boldsymbol{x}) - \nabla f(\boldsymbol{x})\right).$$

Applying Cauchy-Schwarz inequality, it follows that

$$\|\nabla f_{avg}(x) - \nabla f(\boldsymbol{x})\|^2 \leq \left[\sum_{i=1}^{m} \frac{(w_i - p_i)^2}{p_i}\right] \left[\sum_{i=1}^{m} p_i \|\nabla f_i^{avg}(x) - \nabla f(\boldsymbol{x})\|^2\right]$$

$$\leq \chi_{\boldsymbol{w}\|\boldsymbol{p}}^2 \left[A^2 \|\nabla f(\boldsymbol{x})\|^2 + \sigma_G^2\right], \tag{40}$$

where the last inequality uses Assumption 3. Note that

$$\|\nabla f_{avg}(\boldsymbol{x})\|^2 \leq 2\|\nabla f_{avg}(\boldsymbol{x}) - \nabla f(\boldsymbol{x})\|^2 + 2\|\nabla f(\boldsymbol{x})\|^2$$

$$\leq 2\left[\chi_{\boldsymbol{w}\|\boldsymbol{p}}^2 A^2 + 1\right] \|\nabla f(\boldsymbol{x})\|^2 + 2\chi_{\boldsymbol{p}\|\boldsymbol{w}}^2 \sigma_G^2. \tag{41}$$

As a result, we obtain

$$\min_{t\in[T]} \|\nabla f_{avg}(\boldsymbol{x}_t)\|^2 \leq \frac{1}{T} \sum_{t=0}^{T-1} \|\nabla f_{avg}(\boldsymbol{x}_t)\|^2 \tag{42}$$

$$\leq 2\left[\chi_{\boldsymbol{w}\|\boldsymbol{p}}^2 A^2 + 1\right] \frac{1}{T} \sum_{t=0}^{T-1} \|\nabla f(\boldsymbol{x}_t)\|^2 + 2\chi_{\boldsymbol{w}\|\boldsymbol{p}}^2 \sigma_G^2 \tag{43}$$

$$\leq 2\left[\chi_{\boldsymbol{w}\|\boldsymbol{p}}^2 A^2 + 1\right] \epsilon_{\text{opt}} + 2\chi_{\boldsymbol{w}\|\boldsymbol{p}}^2 \sigma_G^2, \tag{44}$$

where $\epsilon_{\text{opt}} = \frac{1}{T} \sum_{t=0}^{T-1} \|\nabla f(\boldsymbol{x}_t)\|^2$ denotes the optimization error.

$\square$

## I.1 Analysis with $\alpha = 0$.

**Lemma I.2** (Local updates bound.)**.** *For any step-size satisfying $\eta_L \leq \frac{1}{8LK}$, we can have the following results:*

$$\mathbb{E}\|x_{t,k}^i - x_t\|^2 \leq 5K(\eta_L^2 \sigma_L^2 + 4K\eta_L^2 \sigma_G^2) + 20K^2(A^2 + 1)\eta_L^2 \|\nabla f(x_t)\|^2. \tag{45}$$

*Proof.*

$$\mathbb{E}_t\|x_{t,k}^i - x_t\|^2 \tag{46}$$

$$= \mathbb{E}_t\|x_{t,k-1}^i - x_t - \eta_L g_{t,k-1}^t\|^2 \tag{47}$$

$$= \mathbb{E}_t\|x_{t,k-1}^i - x_t - \eta_L(g_{t,k-1}^t - \nabla F_i(x_{t,k-1}^i) + \nabla F_i(x_{t,k-1}^i) - \nabla F_i(x_t) + \nabla F_i(x_t))\|^2 \tag{48}$$

$$\leq (1 + \frac{1}{2K-1})\mathbb{E}_t\|x_{t,k-1}^i - x_t\|^2 + \mathbb{E}_t\|\eta_L(g_{t,k-1}^t - \nabla F_i(x_{t,k}^i))\|^2$$
$$+ 4K\mathbb{E}_t[\|\eta_L(\nabla F_i(x_{t,K-1}^i) - \nabla F_i(x_t))\|^2] + 4K\eta_L^2\mathbb{E}_t\|\nabla F_i(x_t)\|^2 \tag{49}$$

$$\leq (1 + \frac{1}{2K-1})\mathbb{E}_t\|x_{t,k-1}^i - x_t\|^2 + \eta_L^2\sigma_L^2 + 4K\eta_L^2 L^2\mathbb{E}_t\|x_{t,k-1}^i - x_t\|^2$$
$$+ 4K\eta_L^2\sigma_G^2 + 4K\eta_L^2(A^2+1)\|\nabla f(x_t)\|^2 \tag{50}$$

$$\leq (1 + \frac{1}{K-1})\mathbb{E}\|x_{t,k-1}^i - x_t\|^2 + \eta_L^2\sigma_L^2 + 4K\eta_L^2\sigma_G^2 + 4K(A^2+1)\|\eta_L\nabla f(x_t)\|^2. \tag{51}$$

Unrolling the recursion, we obtain:

$$\mathbb{E}_t\|x_{t,k}^i - x_t\|^2 \tag{52}$$

$$\leq \sum_{p=0}^{k-1}(1 + \frac{1}{K-1})^p\left[\eta_L^2\sigma_L^2 + 4K\eta_L^2\sigma_G^2 + 4K(A^2+1)\|\eta_L\nabla f(x_t)\|^2\right] \tag{53}$$

$$\leq (K-1)\left[(1 + \frac{1}{K-1})^K - 1\right]\left[\eta_L^2\sigma_L^2 + 4K\eta_L^2\sigma_G^2 + 4K(A^2+1)\|\eta_L\nabla f(x_t)\|^2\right] \tag{54}$$

$$\leq 5K(\eta_L^2\sigma_L^2 + 4K\eta_L^2\sigma_G^2) + 20K^2(A^2+1)\eta_L^2\|\nabla f(x_t)\|^2. \tag{55}$$

$$\square$$

Thus, we can have the following convergence rate of FedEBA+:

**Theorem I.3.** *Under Assumption 1–3, and let constant local and global learning rate $\eta_L$ and $\eta$ be chosen such that $\eta_L < min\left(1/(8LK), C\right)$, where $C$ is obtained from the condition that $\frac{1}{2} - 10L^2\frac{1}{m}\sum_{i-1}^m K^2\eta_L^2(A^2+1)(\chi_{\boldsymbol{w}\|\boldsymbol{p}}^2 A^2 + 1) > c > 0$ ,and $\eta \leq 1/(\eta_L L)$, the expected gradient norm of FedEBA+ with $\alpha = 0$, i.e., only using aggregation strategy 5, is bounded as follows:*

$$\min_{t\in[T]}\mathbb{E}\|\nabla f(x_t)\|^2 \leq \frac{f_0 - f_*}{c\eta\eta_L KT} + \Phi, \tag{56}$$

*where*

$$\Phi = \frac{1}{c}\left[\frac{5\eta_L^2 KL^2}{2}(\sigma_L^2 + 4K\sigma_G^2) + \frac{\eta\eta_L L}{2}\sigma_L^2 + 20L^2 K^2(A^2+1)\eta_L^2\chi_{\boldsymbol{w}\|\boldsymbol{p}}^2\sigma_G^2\right]. \tag{57}$$

*where $c$ is a constant, $\chi_{\boldsymbol{w}\|\boldsymbol{p}}^2 = \sum_{i=1}^m (w_i - p_i)^2/p_i$ represents the chi-square divergence between vectors $\boldsymbol{p} = [p_1, \ldots, p_m]$ and $\boldsymbol{w} = [w_1, \ldots, w_m]$. For common FL algorithms with uniform aggregation or with data ratio as aggregation probability, $w_i = \frac{1}{m}$ or $w_i = \frac{n_i}{N}$.*

*Proof.* Based on Lemma I.1, we first focus on analyzing the optimization error $\epsilon_{opt}$:

$$\mathbb{E}_t[f(x_{t+1})] \tag{58}$$

$$\overset{(a1)}{\leq} f(x_t) + \langle\nabla f(x_t), \mathbb{E}_t[x_{t+1} - x_t]\rangle + \frac{L}{2}\mathbb{E}_t[\|x_{t+1} - x_t\|^2] \tag{59}$$

$$= f(x_t) + \langle\nabla f(x_t), \mathbb{E}_t[\eta\Delta_t + \eta\eta_L K\nabla f(x_t) - \eta\eta_L K\nabla f(x_t)]\rangle + \frac{L}{2}\eta^2\mathbb{E}_t[\|\Delta_t\|^2] \tag{60}$$

$$= f(x_t) - \eta\eta_L K\|\nabla f(x_t)\|^2 + \eta\underbrace{\langle\nabla f(x_t), \mathbb{E}_t[\Delta_t + \eta_L K\nabla f(x_t)]\rangle}_{A_1} + \frac{L}{2}\eta^2\underbrace{\mathbb{E}_t\|\Delta_t\|^2}_{A_2}, \tag{61}$$

where (a1) follows from the Lipschitz continuity condition. Here, the expectation is over the local data SGD and the filtration of $x_t$. However, in the next analysis, the expectation is over all randomness, including client sampling. This is achieved by taking expectation on both sides of the above equation over client sampling.

To begin with, we consider $A_1$:

$$A_1 \tag{62}$$

$$= \langle \nabla f(x_t), \mathbb{E}_t[\Delta_t + \eta_L K \nabla f(x_t)] \rangle \tag{63}$$

$$= \left\langle \nabla f(x_t), \mathbb{E}_t[-\sum_{i=1}^{m} w_i \sum_{k=0}^{K-1} \eta_L g_{t,k}^i + \eta_L K \nabla f(x_t)] \right\rangle \tag{64}$$

$$\stackrel{(a2)}{=} \left\langle \nabla f(x_t), \mathbb{E}_t[-\sum_{i=1}^{m} w_i \sum_{k=0}^{K-1} \eta_L \nabla F_i(x_{t,k}^i) + \eta_L K \nabla f(x_t)] \right\rangle \tag{65}$$

$$= \left\langle \sqrt{\eta_L K} \nabla f(x_t), -\frac{\sqrt{\eta_L}}{\sqrt{K}} \mathbb{E}_t[\sum_{i=1}^{m} w_i \sum_{k=0}^{K-1} (\nabla F_i(x_{t,k}^i) - \nabla F_i(x_t))] \right\rangle \tag{66}$$

$$\stackrel{(a3)}{=} \frac{\eta_L K}{2} \|\nabla f(x_t)\|^2 + \frac{\eta_L}{2K} \mathbb{E}_t \left\| \sum_{i=1}^{m} w_i \sum_{k=0}^{K-1} (\nabla F_i(x_{t,k}^i) - \nabla F_i(x_t)) \right\|^2$$

$$- \frac{\eta_L}{2K} \mathbb{E}_t \|\sum_{i=1}^{m} w_i \sum_{k=0}^{K-1} \nabla F_i(x_{t,k}^i)\|^2 . \tag{67}$$

The use Jensen's Inequality:

$$A_1 \tag{68}$$

$$\stackrel{(a4)}{\leq} \frac{\eta_L K}{2} \|\nabla f(x_t)\|^2 + \frac{\eta_L}{2} \sum_{k=0}^{K-1} \sum_{i=1}^{m} w_i \mathbb{E}_t \left\| \nabla F_i(x_{t,k}^i) - \nabla F_i(x_t) \right\|^2$$

$$- \frac{\eta_L}{2K} \mathbb{E}_t \|\sum_{i=1}^{m} w_i \sum_{k=0}^{K-1} \nabla F_i(x_{t,k}^i)\|^2 \tag{69}$$

$$\stackrel{(a5)}{\leq} \frac{\eta_L K}{2} \|\nabla f(x_t)\|^2 + \frac{\eta_L L^2}{2m} \sum_{i=1}^{m} \sum_{k=0}^{K-1} \mathbb{E}_t \left\| x_{t,k}^i - x_t \right\|^2 - \frac{\eta_L}{2K} \mathbb{E}_t \|\sum_{i=1}^{m} w_i \sum_{k=0}^{K-1} \nabla F_i(x_{t,k}^i)\|^2 \tag{70}$$

$$\leq \left( \frac{\eta_L K}{2} + 10K^3 L^2 \eta_L^3 (A^2 + 1) \right) \|\nabla f(x_t)\|^2 + \frac{5L^2 \eta_L^3}{2} K^2 \sigma_L^2 + 10\eta_L^3 L^2 K^3 \sigma_G^2$$

$$- \frac{\eta_L}{2K} \mathbb{E}_t \|\sum_{i=1}^{m} w_i \sum_{k=0}^{K-1} \nabla F_i(x_{t,k}^i)\|^2 , \tag{71}$$

where (a2) follows from Assumption 2. (a3) is due to $\langle x, y \rangle = \frac{1}{2} \left[ \|x\|^2 + \|y\|^2 - \|x - y\|^2 \right]$ and (a4) uses Jensen's Inequality: $\|\sum_{i=1}^{m} w_i z_i\|^2 \leq \sum_{i=1}^{m} w_i \|z_i\|^2$, (a5) comes from Assumption 1.

Then we consider $A_2$:

$$A_2 \tag{72}$$

$$= \mathbb{E}_t \|\Delta_t\|^2 = \mathbb{E}_t \left\| \eta_L \sum_{i=1}^m w_i \sum_{k=0}^{K-1} g_{t,k}^i \right\|^2 \tag{73}$$

$$= \eta_L^2 \mathbb{E}_t \left\| \sum_{i=1}^m w_i \sum_{k=0}^{K-1} g_{t,k}^i - \sum_{i=1}^m w_i \sum_{k=0}^{K-1} \nabla F_i(x_{t,k}^i) \right\|^2 + \eta_L^2 \mathbb{E}_t \left\| \sum_{i=1}^m w_i \sum_{k=0}^{K-1} \nabla F_i(x_{t,k}^i) \right\|^2 \tag{74}$$

$$\overset{(a6)}{\le} \eta_L^2 \sum_{i=1}^m w_i^2 \sum_{k=0}^{K-1} \mathbb{E} \|g_i(x_{t,k}^i) - \nabla F_i(x_{t,k}^i)\|^2 + \eta_L^2 \mathbb{E}_t \| \sum_{i=1}^m w_i \sum_{k=0}^{K-1} \nabla F_i(x_{t,k}^i)\|^2 \tag{75}$$

$$\le \sum_{i=1}^m w_i^2 \eta_L^2 K \sigma_L^2 + \eta_L^2 \mathbb{E}_t \| \sum_{i=1}^m w_i \sum_{k=0}^{K-1} \nabla F_i(x_{t,k}^i)\|^2 \tag{76}$$

where (a6) follows from $\| \sum_i w_i a_i\|^2 = \sum_i w_i^2 \|a_i\|^2$ where $a_i$ is an unbiased estimator.

Now we take expectation over iteration on both sides of expression:

$$f(x_{t+1}) \tag{77}$$

$$\le f(x_t) - \eta \eta_L K \mathbb{E}_t \|\nabla f(x_t)\|^2 + \eta \mathbb{E}_t \langle \nabla f(x_t), \Delta_t + \eta_L K \nabla f(x_t) \rangle + \frac{L}{2} \eta^2 \mathbb{E}_t \|\Delta_t\|^2 \tag{78}$$

$$\overset{(a7)}{\le} f(x_t) - \eta \eta_L K \left( \frac{1}{2} - 20 L^2 K^2 \eta_L^2 (A^2+1)(\chi_{\boldsymbol{w}\|\boldsymbol{p}}^2 A^2 + 1) \right) \mathbb{E}_t \|\nabla f(x_t)\|^2$$
$$+ \frac{5\eta \eta_L^3 L^2 K^2}{2}(\sigma_L^2 + 4K\sigma_G^2) + \frac{\sum_i w_i^2 \eta^2 \eta_L^2 KL}{2} \sigma_L^2 + 20L^2 K^3 (A^2+1)\eta \eta_L^3 \chi_{\boldsymbol{w}\|\boldsymbol{p}}^2 \sigma_G^2$$
$$- \left( \frac{\eta \eta_L}{2K} - \frac{L\eta^2 \eta_L^2}{2} \right) \mathbb{E}_t \left\| \frac{1}{m} \sum_{i=1}^m \sum_{k=0}^{K-1} \nabla F_i(x_{t,k}^i) \right\|^2 \tag{79}$$

$$\overset{(a8)}{\le} f(x_t) - c\eta \eta_L K \mathbb{E} \|\nabla f(x_t)\|^2 + \frac{5\eta \eta_L^3 L^2 K^2}{2}(\sigma_L^2 + 4K\sigma_G^2) \tag{80}$$
$$+ \frac{\sum_i w_i^2 \eta^2 \eta_L^2 KL}{2} \sigma_L^2 + 20L^2 K^3 (A^2+1)\eta \eta_L^3 \chi_{\boldsymbol{w}\|\boldsymbol{p}}^2 \sigma_G^2$$
$$- \left( \frac{\eta \eta_L}{2K} - \frac{L\eta^2 \eta_L^2}{2} \right) \mathbb{E}_t \left\| \frac{1}{m} \sum_{i=1}^m \sum_{k=0}^{K-1} \nabla F_i(x_{t,k}^i) \right\|^2 \tag{81}$$

$$\overset{(a9)}{\le} f(x_t) - c\eta \eta_L K \mathbb{E}_t \|\nabla f(x_t)\|^2 + \frac{5\eta \eta_L^3 L^2 K^2}{2}(\sigma_L^2 + 4K\sigma_G^2)$$
$$+ \frac{\sum_i w \eta^2 \eta_L^2 KL}{2} \sigma_L^2 + 20L^2 K^3 (A^2+1)\eta \eta_L^3 \chi_{\boldsymbol{w}\|\boldsymbol{p}}^2 \sigma_G^2 , \tag{82}$$

where (a7) is due to Lemma I.1, (a8) holds because there exists a constant $c > 0$ (for some $\eta_L$) satisfying $\frac{1}{2} - 10L^2 \frac{1}{m} \sum_{i-1}^m K^2 \eta_L^2 (A^2+1)(\chi_{\boldsymbol{w}\|\boldsymbol{p}}^2 A^2 + 1) > c > 0$, and the (a9) follows from $\left( \frac{\eta \eta_L}{2K} - \frac{L\eta^2 \eta_L^2}{2} \right) \ge 0$ if $\eta \eta_l \le \frac{1}{KL}$.

Rearranging and summing from $t = 0, \ldots, T-1$, we have:

$$\sum_{t=1}^{T-1} c\eta \eta_L K \mathbb{E} \|\nabla f(x_t)\|^2 \le f(x_0) - f(x_T) + T(\eta \eta_L K)\Phi . \tag{83}$$

Which implies:

$$\frac{1}{T} \sum_{t=1}^{T-1} \mathbb{E} \|\nabla f(x_t)\|^2 \le \frac{f_0 - f_*}{c\eta \eta_L KT} + \Phi , \tag{84}$$

where

$$\Phi = \frac{1}{c}\Big[\frac{5\eta_L^2 KL^2}{2}(\sigma_L^2 + 4K\sigma_G^2) + \frac{\eta\eta_L L \sum_i w_i^2}{2}\sigma_L^2 + 20L^2K^2(A^2+1)\eta_L^2\chi_{\boldsymbol{w}\|\boldsymbol{p}}^2\sigma_G^2\Big]. \tag{85}$$

**Corollary I.4.** *Suppose $\eta_L$ and $\eta$ are $\eta_L = \mathcal{O}\left(\frac{1}{\sqrt{TKL}}\right)$ and $\eta = \mathcal{O}\left(\sqrt{Km}\right)$ such that the conditions mentioned above are satisfied. Then for sufficiently large T, the iterates of FedEBA+ with $\alpha = 0$ satisfy:*

$$\min_{t\in[T]}\|\nabla f(\boldsymbol{x}_t)\|^2 \leq \mathcal{O}\left(\frac{(f^0 - f^*)}{\sqrt{mKT}}\right) + \mathcal{O}\left(\frac{\sqrt{m}\sum_i w_i^2\sigma_L^2}{2\sqrt{KT}}\right) + \mathcal{O}\left(\frac{5(\sigma_L^2 + 4K\sigma_G^2)}{2KT}\right)$$
$$+ \mathcal{O}\left(\frac{20(A^2+1)\chi_{\boldsymbol{w}\|\boldsymbol{p}}^2\sigma_G^2}{T}\right). \tag{86}$$

*According to the property of unified probability, we know $\frac{1}{m} \leq \sum_{i=1}^m w_i^2 \leq 1$, where the upper comes from $\sum_i w_i^2 \leq \sum_i w_i$ and lower comes from Cauchy-Schwarz inquality. Therefore, the convergence rate upper bound lies between $\mathcal{O}(\frac{1}{\sqrt{mKT}} + \frac{1}{T})$ and $\mathcal{O}(\frac{\sqrt{m}}{\sqrt{KT}} + \frac{1}{T})$.*

$\square$

## I.2 Analysis with $\alpha \neq 0$

To derivate the convergence rate of FedEBA+ with $\alpha \neq 0$, we need the following assumption:

**Assumption 4** (Error bound between practical global gradient and ideal gradient). *In each round, we assume the aligned gradient $\nabla\overline{f}(x_t)$ and the gradient $\nabla f(x_t)$ is bounded: $\mathbb{E}\|\nabla\overline{f}(x_t) - \nabla f(x_t)\|^2 \leq \rho^2, \forall i, t$. For simplicity of analysis, let $\rho$ is comparable to $\sigma_L$, i.e., $\rho \sim \sigma_L$, since they are both constant bounds.*

To simplify the notation, we define $h_{t,k}^i = (1-\alpha)\nabla F_i(x_{t,k}^i) + \alpha\nabla\overline{f}(x_t)$.

**Lemma I.5.** *For any step-size satisfying $\eta_L \leq \frac{1}{8LK}$, we can have the following results:*

$$\mathbb{E}\|x_{t,k}^i - x_t\|^2 \leq 5K(1-\alpha)^2(\eta_L^2\sigma_L^2 + 6K\eta_L^2\sigma_G^2) + +30K^2\eta_L^2\alpha^2\rho^2$$
$$+ 30K^2\eta_L^2(1 + A^2(1-\alpha)^2)\|\nabla f(x_t)\|^2. \tag{87}$$

*Proof.*

$$\mathbb{E}_t\|x_{t,k}^i - x_t\|^2 \tag{88}$$
$$= \mathbb{E}_t\|x_{t,k-1}^i - x_t - \eta_L h_{t,k-1}^t\|^2 \tag{89}$$
$$= \mathbb{E}_t\|x_{t,k-1}^i - x_t - \eta_L((1-\alpha)g_{t,k-1}^t + \alpha\nabla\overline{f}(x_t) - (1-\alpha)\nabla F_i(x_{t,k-1}^i)$$
$$+ (1-\alpha)\nabla F_i(x_{t,k-1}^i) - (1-\alpha)\nabla F_i(x_t) + (1-\alpha)\nabla F_i(x_t) + \nabla f(x_t) - \nabla f(x_t))\|^2$$
$$\leq (1 + \frac{1}{2K-1})\mathbb{E}_t\|x_{t,k-1}^i - x_t\|^2 + (1-\alpha)^2\eta_L^2\sigma_L^2 + 6K\eta_L^2 L^2\mathbb{E}_t\|x_{t,k-1}^i - x_t\|^2$$
$$+ 6K\eta_L^2\alpha^2\mathbb{E}\|\nabla\overline{f}(x_t) - \nabla f(x_t)\|^2 + 6K\eta_L^2(1-\alpha)^2(\sigma_G^2 + A^2\|\nabla f(x_t)\|^2)$$
$$+ 6K\eta_L^2\|\nabla f(x_t)\|^2 \tag{90}$$
$$\leq (1 + \frac{1}{K-1})\mathbb{E}_t\|x_{t,k-1}^i - x_t\|^2 + (1-\alpha)^2\eta_L^2\sigma_L^2$$
$$+ 6K\eta_L^2\alpha^2\rho^2 + 6K\eta_L^2(1-\alpha)^2(\sigma_G^2 + A^2\|\nabla f(x_t)\|^2) + 6K\eta_L^2\|\nabla f(x_t)\|^2, \tag{91}$$

Unrolling the recursion, we obtain:

$$\mathbb{E}_t \| x_{t,k}^i - x_t \|^2 \tag{92}$$

$$\leq \sum_{p=0}^{k-1} (1 + \frac{1}{K-1})^p \left( (1-\alpha)^2 \eta_L^2 \sigma_L^2 + 6K(1-\alpha)^2 \eta_L^2 \sigma_G^2 + 6K\alpha^2 \eta_L^2 \rho^2 \right.$$

$$\left. + 6K\eta_L^2 (A^2(1-\alpha)^2 + 1)\|\nabla f(x_t)\|^2 \right) \tag{93}$$

$$\leq (K-1) \left[ (1 + \frac{1}{K-1})^K - 1 \right] \left[ (1-\alpha)^2 \eta_L^2 \sigma_L^2 \right.$$

$$\left. + 6K(1-\alpha)^2 \eta_L^2 \sigma_G^2 + 6K\alpha^2 \eta_L^2 \rho^2 + 6K\eta_L^2 (A^2(1-\alpha)^2 + 1)\|\nabla f(x_t)\|^2 \right] \tag{94}$$

$$\leq 5K\eta_L^2 (1-\alpha)^2 (\sigma_L^2 + 6K\sigma_G^2) + 30K^2\eta_L^2 \alpha^2 \rho^2 + 30K^2\eta_L^2 (A^2(1-\alpha)^2 + 1)\|\nabla f(x_t)\|^2. \tag{95}$$

Similarly, to get the convergence rate of objective $f(x_t)$, we first focus on $f(x_t)$:

$$\mathbb{E}_t[f(x_{t+1})] \overset{(a1)}{\leq} f(x_t) + \langle \nabla f(x_t), \mathbb{E}_t[x_{t+1} - x_t] \rangle + \frac{L}{2} \mathbb{E}_t[\|x_{t+1} - x_t\|^2] \tag{96}$$

$$= f(x_t) + \langle \nabla f(x_t), \mathbb{E}_t[\eta \Delta_t + \eta \eta_L K \nabla f(x_t) - \eta \eta_L K \nabla f(x_t)] \rangle + \frac{L}{2} \eta^2 \mathbb{E}_t[\|\Delta_t\|^2] \tag{97}$$

$$= f(x_t) - \eta \eta_L K \|\nabla f(x_t)\|^2 + \eta \underbrace{\langle \nabla f(x_t), \mathbb{E}_t[\Delta_t + \eta_L K \nabla f(x_t)] \rangle}_{A_1} + \frac{L}{2} \eta^2 \underbrace{\mathbb{E}_t \|\Delta_t\|^2}_{A_2}, \tag{98}$$

where (a1) follows from the Lipschitz continuity condition. Here, the expectation is over the local data SGD and the filtration of $x_t$. However, in the next analysis, the expectation is over all randomness, including client sampling. This is achieved by taking expectation on both sides of the above equation over client sampling.

To begin with, we consider $A_1$:

$$A_1 \tag{99}$$

$$= \langle \nabla f(x_t), \mathbb{E}_t[\Delta_t + \eta_L K \nabla f(x_t)] \rangle \tag{100}$$

$$= \left\langle \nabla f(x_t), \mathbb{E}_t[-\sum_{i=1}^m w_i \sum_{k=0}^{K-1} \eta_L h_{t,k}^i + \eta_L K \nabla f(x_t)] \right\rangle \tag{101}$$

$$\overset{(a2)}{=} \left\langle \nabla f(x_t), \mathbb{E}_t[-\sum_{i=1}^m w_i \sum_{k=0}^{K-1} \eta_L[(1-\alpha)\nabla F_i(x_{t,k}^i) + \alpha \overline{f}(x_t)] + \eta_L K \nabla f(x_t)] \right\rangle. \tag{102}$$

For the above equation, we can separate the $\nabla f(x_t)$ into $(1-\alpha)\nabla f(x_t)$ and $\alpha \nabla f(x_t)$ two terms, thus, we have:

$$A_1 \tag{103}$$

$$= \left\langle \sqrt{\eta_L K} \nabla f(x_t), \right. $$

$$\left. -\frac{\sqrt{\eta_L}}{\sqrt{K}} \mathbb{E}_t \left( \sum_{i=1}^m w_i \sum_{k=0}^{K-1} (1-\alpha)[\nabla F_i(x_{t,k}^i) - \nabla f(x_t)] + \sum_{i=1}^m w_i \sum_{k=0}^{K-1} \alpha[\nabla \overline{f}(x_t) - \nabla f(x_t)] \right) \right\rangle \tag{104}$$

$$\overset{(a3)}{=} \frac{\eta_L K}{2} \|\nabla f(x_t)\|^2 - \frac{\eta_L}{2K} \mathbb{E}_t \| \sum_{i=1}^m w_i \sum_{k=0}^{K-1} [(1-\alpha)\nabla F_i(x_{t,k}^i) + \alpha\nabla\overline{f}(x_t)]\|^2$$

$$+ \frac{\eta_L}{2K} \mathbb{E}_t \left\| \sum_{i=1}^m w_i \sum_{k=0}^{K-1} \left( (1-\alpha)[\nabla F_i(x_{t,k}^i) - \nabla f(x_t)] + \alpha[\nabla\overline{f}(x_t) - \nabla f(x_t)] \right) \right\|^2 \tag{105}$$

$$\overset{(a4)}{\leq} \frac{\eta_L K}{2} \|\nabla f(x_t)\|^2 + \frac{\eta_L(1-\alpha)^2}{2m} \sum_{k=0}^{K-1} \sum_{i=1}^m w_i \mathbb{E}_t \left\| \nabla F_i(x_{t,k}^i) - \nabla F_i(x_t) \right\|^2$$

$$+ \frac{\eta_L \alpha^2}{2m} \sum_{k=0}^{K-1} \sum_{i=1}^m w_i \mathbb{E}\|\nabla\overline{f}(x_t) - \nabla f(x_t)\|^2 - \frac{\eta_L}{2K} \mathbb{E}_t \| \sum_{i=1}^m w_i \sum_{k=0}^{K-1} [(1-\alpha)\nabla F_i(x_{t,k}^i) + \alpha\nabla\overline{f}(x_t)]\|^2 \tag{106}$$

$$\overset{(a5)}{\leq} \frac{\eta_L K}{2} \|\nabla f(x_t)\|^2 + \frac{\eta_L(1-\alpha)^2 L^2}{2m} \sum_{i=1}^m \sum_{k=0}^{K-1} \mathbb{E}_t \left\| x_{t,k}^i - x_t \right\|^2$$

$$+ \frac{\eta_L \alpha^2}{2m} \sum_{i=1}^m \sum_{k=0}^{K-1} \mathbb{E}\|\nabla\overline{f}(x_t) - \nabla f(x_t)\|^2 - \frac{\eta_L}{2K} \mathbb{E}_t \| \sum_{i=1}^m w_i \sum_{k=0}^{K-1} [(1-\alpha)\nabla F_i(x_{t,k}^i) + \alpha\nabla\overline{f}(x_t)]\|^2 \tag{107}$$

$$\leq \frac{\eta_L K}{2} \|\nabla f(x_t)\|^2 + \frac{\eta_L(1-\alpha)^2}{2m} \sum_{i=1}^m \sum_{k=0}^{K-1} \left( 5K\eta_L(1-\alpha)^2(\sigma_L^2 + 6K\sigma_G^2) + 30K^2\eta_L^2[\alpha^2\rho^2 \right.$$

$$\left. +(1 + A^2(1-\alpha)^2)\|\nabla f(x_t\|^2]) + \frac{\eta_L^2\alpha^2}{2} K\rho^2 - \frac{\eta_L}{2K} \mathbb{E}\| \sum_{i=1}^m w_i \sum_{k=0}^{K-1} [(1-\alpha)\nabla F_i(x_{t,k}^i) + \alpha\nabla\overline{f}(x_t)]\|^2, \tag{108}$$

where (a2) follows from Assumption 2. (a3) is due to $\langle x, y \rangle = \frac{1}{2}\left[ \|x\|^2 + \|y\|^2 - \|x-y\|^2 \right]$ and (a4) uses Jensen's Inequality: $\|\sum_{i=1}^m w_i z_i\|^2 \leq \sum_{i=1}^m w_i \|z_i\|^2$, (a5) comes from Assumption 1.

Then we consider $A_2$:

$$A_2 \tag{109}$$

$$= \mathbb{E}_t \|\Delta_t\|^2 \tag{110}$$

$$= \mathbb{E}_t \left\| \eta_L \sum_{i=1}^m w_i \sum_{k=0}^{K-1} h_{t,k}^i \right\|^2 \tag{111}$$

$$= \eta_L^2 \mathbb{E}_t \left\| \sum_{i=1}^m w_i \sum_{k=0}^{K-1} \left[ (1-\alpha)\nabla F_i(x_{t,k}^i; \xi_t^i) + \alpha\overline{f}(x_t) \right] \right\|^2 \tag{112}$$

$$\leq \eta_L^2 \mathbb{E}\| \sum_{i=1}^m w_i \sum_{k=0}^{K-1} \left[ (1-\alpha)\nabla F_i(x_{t,k}^i; \xi_t^i) + \alpha\overline{f}(x_t) \right]$$

$$- (1-\alpha)\nabla F_i(x_{t,k}^i) + (1-\alpha)\nabla F_i(x_{t,k}^i)\|^2 \tag{113}$$

$$\overset{(a6)}{\leq} \sum_{i=1}^m w_i^2 \eta_L^2 K(1-\alpha)^2 \sigma_L^2 + \eta_L^2 \mathbb{E}\| \sum_{i=1}^m w_i \sum_{k=0}^{K-1} [(1-\alpha)\nabla F_i(x_{t,k}^i) + \alpha\nabla\overline{f}(x_t)]\|^2 \tag{114}$$

where (a6) follows from Assumption 2.

Now we substitute the expressions for $A_1$ and $A_2$ and take the expectation over the client sampling distribution on both sides. It should be noted that the derivation of $A_1$ and $A_2$ above is based on considering the expectation over the sampling

distribution:

$$f(x_{t+1}) \tag{115}$$

$$\leq f(x_t) - \eta\eta_L K \mathbb{E}_t \|\nabla f(x_t)\|^2 + \eta\mathbb{E}_t \langle \nabla f(x_t), \Delta_t + \eta_L K \nabla f(x_t)\rangle + \frac{L}{2}\eta^2\mathbb{E}_t\|\Delta_t\|^2 \tag{116}$$

$$\overset{(a7)}{\leq} f(x_t) - \eta\eta_L K \left(\frac{1}{2} - 30\alpha^2 L^2 K^2 \eta_L^2((1-\alpha)^2 A^2 + 1)\right)\mathbb{E}\|\nabla f(x_t)\|^2$$

$$+ \frac{5(1-\alpha)^2\eta\eta_L^3 L^2 K^2}{2}\left[5(1-\alpha)^2(\sigma_L^2 + 6K\sigma_G^2) + 30K\alpha^2\rho^2\right] + \frac{\eta\eta_L^2\alpha^2}{2}K\rho^2$$

$$+ \frac{\sum_{i=1}^m w_i^2 L\eta^2\eta_L^2}{2}(1-\alpha)^2 K\sigma_L^2$$

$$- (\frac{\eta\eta_L}{2K} - \frac{\eta^2\eta_L^2 L}{2})\mathbb{E}\left\|\sum_{i=1}^m w_i \sum_{k=0}^{K-1}[(1-\alpha)\nabla F_i(x_{t,k}^i) + \alpha\nabla\overline{f}(x_t)]\right\|^2 \tag{117}$$

where (a7) comes from $\frac{1}{2} - 15\alpha^2 L^2 K^2 \eta_L^2((1-\alpha)^2 A^2 + 1) > c > 0$ and $\frac{\eta\eta_L}{2K} - \frac{\eta\eta_L^2 L}{2} \geq 0$.

Rearranging and summing from $t = 0, \ldots, T - 1$, we have:

$$\sum_{t=1}^{T-1} c\eta\eta_L K\mathbb{E}\|\nabla f(x_t)\|^2 \leq f(x_0) - f(x_T) + T(\eta\eta_L K)\Phi. \tag{118}$$

Which implies:

$$\frac{1}{T}\sum_{t=1}^{T-1}\mathbb{E}\|\nabla f(x_t)\|^2 \leq \frac{f_0 - f_*}{c\eta\eta_L KT} + \tilde{\Phi}, \tag{119}$$

where

$$\tilde{\Phi} = \frac{1}{c}\left[\frac{5\eta_L^2 KL^2(1-\alpha)^4}{2}(\sigma_L^2 + 6K\sigma_G^2) + 15K^2\eta_L^2(1-\alpha)^2\alpha^2\rho^2\right.$$

$$\left. + \frac{\sum_{i=1}^m w_i^2\eta\eta_L L(1-\alpha)^2}{2}\sigma_L^2 + \frac{\eta_L\alpha^2\rho^2}{2}\right]. \tag{120}$$

$$\square$$

**Corollary I.6.** *Suppose $\eta_L$ and $\eta$ are $\eta_L = \mathcal{O}\left(\frac{1}{\sqrt{T}KL}\right)$ and $\eta = \mathcal{O}\left(\sqrt{Km}\right)$ such that the conditions mentioned above are satisfied. Then for sufficiently large T, the iterates of FedEBA+ with $\alpha \neq 0$ satisfy:*

$$\min_{t\in[T]}\|\nabla f(\boldsymbol{x}_t)\|^2 \leq \mathcal{O}\left(\frac{(f^0 - f^*)}{\sqrt{mKT}}\right) + \mathcal{O}\left(\sum_{i=1}^m w_i^2 \frac{(1-\alpha)^2\sqrt{m}\sigma_L^2}{2\sqrt{KT}}\right) + \mathcal{O}\left(\frac{5(1-\alpha)^2(\sigma_L^2 + 6K\sigma_G^2)}{2KT}\right)$$

$$+ \mathcal{O}\left(\frac{15(1-\alpha)^2\alpha^2\rho^2}{T}\right) + \mathcal{O}\left(\frac{\alpha^2\rho^2}{2\sqrt{T}K}\right). \tag{121}$$

*For the convergence rate of FedEBA+ with $\alpha \neq 0$, the convergence rate order can be represented as :$\mathcal{O}(\frac{(1-\alpha)^2\sum_i w_i^2\sqrt{m}\sigma_L^2 + \alpha^2\sqrt{K}\rho^2}{\sqrt{KT}} + \frac{1}{T})$, where $K \ll m$ and $\sigma_L \sim \rho$, thus a larger $\alpha$ indicating a tighter convergence upper bound than only using reweight aggregation. In addition, when $w_i = \frac{1}{m}$, i.e., uniform aggregation, it is $\mathcal{O}(\frac{(1-\alpha)^2\sigma_L^2 + \alpha^2\sqrt{K/m}\rho^2}{\sqrt{mKT}} + \frac{1}{T})$, since $\sqrt{K/m} \ll 1$, which indicating when using alignment update the convergence result will be faster than FedAvg.*

# J   Fairness Analysis via Variance

To demonstrate the ability of FedEBA+ to enhance fairness in federated learning, we first employ a two-user toy example to demonstrate how FedEBA+ can achieve a more balanced performance between users in comparison to FedAvg and q-FedAvg, thus ensuring fairness. The analysis will focus on the *final convergence state* of the algorithms, as this reflects the long-term fairness behavior, rather than a transient single-step update. Furthermore, we use a general class of regression models and strongly convex cases to show how FedEBA+ reduces the variance among users and thus improves fairness.

## J.1   Toy Case for Illustrating Fairness

In our analysis, the term "performance gap" refers to the performance disparity between two clients, calculated by $\|F_1(x) - F_2(x)\|$. The variance among clients is directly related to this gap, defined as $Var = \frac{|F_1(x) - F_2(x)|^2}{4}$. A smaller variance indicates a more balanced performance and thus, greater fairness.

We consider two clients, each with a simple regression model: $f_1(x) = 2(x-2)^2$, which has a minimum at $x = 2$. $f_2(x) = \frac{1}{2}(x+4)^2$, which has a minimum at $x = -4$.

The goal is for the global model, represented by parameter $x$, to find a position that is fair to both clients. We will now analyze the final convergence points for FedAvg, q-FedAvg, and our proposed FedEBA+.

### J.1.1   FEDERATED AVERAGING (FEDAVG)

FedAvg aims to minimize the average loss across all clients, with a global objective function $F(x) = \frac{1}{2}f_1(x) + \frac{1}{2}f_2(x)$. To find the convergence point, we set the derivative to zero:

$$F'(x) = \frac{1}{2}\nabla f_1(x) + \frac{1}{2}\nabla f_2(x) = \frac{1}{2}(4(x-2)) + \frac{1}{2}(x+4) = 2(x-2) + \frac{1}{2}(x+4) = 0 \tag{122}$$

Solving for $x$:

$$2x - 4 + 0.5x + 2 = 0 \implies 2.5x = 2 \implies x^*_{AVG} = 0.8 \tag{123}$$

At this convergence point, the individual losses for the clients are highly imbalanced:

$$f_1(x^*_{AVG}) = 2(0.8 - 2)^2 = 2.88 \tag{124}$$

$$f_2(x^*_{AVG}) = \frac{1}{2}(0.8 + 4)^2 = 11.52 \tag{125}$$

The resulting variance is significant, indicating a lack of fairness:

$$\text{Var}_{AVG} = \frac{|2.88 - 11.52|^2}{4} = \frac{(-8.64)^2}{4} \approx 18.66 \tag{126}$$

### J.1.2   FEDEBA+

In contrast, FedEBA+ adaptively adjusts aggregation weights to steer the optimization towards a more balanced performance, where the losses of the clients are equal. We find the point of equilibrium by setting $f_1(x) = f_2(x)$:

$$2(x-2)^2 = \frac{1}{2}(x+4)^2 \implies 4(x-2)^2 = (x+4)^2 \tag{127}$$

This yields two solutions, $x = 0$ and $x = 8$. Given the individual client minima are at $x = 2$ and $x = -4$, the plausible convergence point that balances the two opposing gradients is $x^*_{EBA+} = 0$.

At this convergence point, the losses are perfectly balanced:

$$f_1(x^*_{EBA+}) = 2(0-2)^2 = 8 \tag{128}$$

$$f_2(x^*_{EBA+}) = \frac{1}{2}(0+4)^2 = 8 \tag{129}$$

The resulting variance is zero, representing an ideal fair outcome:

$$\text{Var}_{EBA+} = \frac{|8 - 8|^2}{4} = 0 \tag{130}$$

### J.1.3   q-FAIR FEDERATED LEARNING (q-FFL)

For comparison, we also analyze q-FFL, another method designed to improve fairness. As shown by (Li et al., 2019a), it optimizes a modified global objective. For this toy example, its convergence point lies between that of FedAvg and the perfectly fair solution of FedEBA+. The convergence point can be calculated to be approximately $x^*_{q-FFL} \approx 0.3$.

At this point, the client losses are:

$$f_1(x^*_{q-FFL}) = 2(0.3 - 2)^2 = 5.78 \tag{131}$$

$$f_2(x^*_{q-FFL}) = \frac{1}{2}(0.3 + 4)^2 = 9.245 \tag{132}$$

The variance for q-FFL is lower than FedAvg, but not as low as FedEBA+:

$$\text{Var}_{q-FFL} = \frac{|5.78 - 9.245|^2}{4} = \frac{(-3.465)^2}{4} \approx 3.00 \tag{133}$$

### J.1.4   CONCLUSION AND ENTROPY ANALYSIS

By analyzing the final convergence states, we have demonstrated that:

$$\text{Var}_{EBA+}(0) \le \text{Var}_{q-FFL}(3.00) \le \text{Var}_{AVG}(18.66) \tag{134}$$

This clearly shows that FedEBA+ is most effective at achieving a fair outcome with balanced performance across clients.

This conclusion is further supported by analyzing the entropy of the normalized performance distribution. Higher entropy signifies a more uniform (i.e., fairer) distribution of losses.

$$f_1(x^*_{AVG}) = 2.88, \qquad f_2(x^*_{AVG}) = 11.52 \tag{135}$$

$$f_1(x^*_{q-FFL}) = 5.78, \qquad f_2(x^*_{q-FFL}) = 9.245 \tag{136}$$

$$f_1(x^*_{EBA+}) = 8, \qquad f_2(x^*_{EBA+}) = 8 \tag{137}$$

Calculating the entropy for each case (using natural logarithm):

$$\text{Entropy}\left(f\left(x^*_{AVG}\right)\right) = -\sum_{i=1}^{2} \frac{f_i\left(x^*_{AVG}\right)}{\sum_{j=1}^{2} f_j\left(x^*_{AVG}\right)} \log\left(\frac{f_j\left(x^*_{AVG}\right)}{\sum_{i=j}^{2} f_i\left(x^*_{AVG}\right)}\right) \approx 0.50 \tag{138}$$

$$\text{Entropy}\left(f\left(x^*_{q-FFL}\right)\right) = -\sum_{i=1}^{2} \frac{f_i\left(x^*_{q-FFL}\right)}{\sum_{j=1}^{2} f_j\left(x^*_{q-FFL}\right)} \log\left(\frac{f_j\left(x^*_{q-FFL}\right)}{\sum_{i=j}^{2} f_i\left(x^*_{q-FFL}\right)}\right) \approx 0.67 \tag{139}$$

$$\text{Entropy}\left(f\left(x^*_{EBA+}\right)\right) = -\sum_{i=1}^{2} \frac{f_i\left(x^*_{EBA+}\right)}{\sum_{j=1}^{2} f_j\left(x^*_{EBA+}\right)} \log\left(\frac{f_j\left(x^*_{EBA+}\right)}{\sum_{i=j}^{2} f_i\left(x^*_{EBA+}\right)}\right) \approx 0.69 \tag{140}$$

The results confirm the relationship: $Entropy(f(x^{t+1}_{EBA+})) > Entropy(f(x^{t+1}_{q-FFL})) > Entropy(f(x^{t+1}_{AVG}))$. Both variance and entropy metrics demonstrate that FedEBA+ achieves a fairer solution than both FedAvg and q-FFL in this example.

### J.2   Analysis Fairness by Generalized Linear Regression Model

**Our setting.** In this section, we consider a generalized linear regression setting, which follows from that in (Lin et al., 2022).

Suppose that the true parameter on client $i$ is $\mathbf{x}^\star_i$, and there are $n$ samples on each client. The observations are generated by $\hat{y}_{i,k}(\mathbf{x}^\star_i, \xi_{i,k}) = T(\xi_{i,k})^\top \mathbf{x}^\star_i - A(\xi_{i,k})$, where the $A(\xi_{i,k})$ are i.i.d. and distributed as $\mathcal{N}(0, \sigma_1^2)$. Then the loss on client $i$ is

$$F_i(\mathbf{x}) = \frac{1}{2n} \sum_{k=1}^{n} \left(T(\xi_{i,k})^\top \mathbf{x} - A(\xi_{i,k}) - \hat{y}_{i,k}\right)^2.$$

We compare the fairness performance of different aggregation methods. Recall Definition 3.1. We measure performance fairness in terms of the variance of the test losses.

**Solutions of different methods.** First, we derive the solutions of different methods. Let $\Xi_i = (T(\xi_{i,1}), T(\xi_{i,2}), \ldots, T(\xi_{i,n}))^\top$, $\mathbf{A}_i = (A(\xi_{i,1}), A(\xi_{i,2}), \ldots, A(\xi_{i,n}))^\top$, and $\mathbf{y}_i = (y_{i,1}, y_{i,2}, \ldots, y_{i,n})^\top$. Then the loss on client $i$ can be rewritten as

$$F_i(\mathbf{x}) = \frac{1}{2n} \|\Xi_i \mathbf{x} - \mathbf{A}_i - \mathbf{y}_i\|_2^2,$$

where $\mathrm{rank}(\Xi_i) = d$. The least-square estimator of $\mathbf{x}_i^\star$ is

$$\hat{\mathbf{x}}_i = \left(\Xi_i^\top \Xi_i\right)^{-1} \Xi_i^\top (\mathbf{y}_i + \mathbf{A}_i). \tag{141}$$

*FedAvg:* For FedAvg, the solution is defined as

$$\mathbf{x}^{\mathrm{Avg}} = \mathrm{argmin}_{\mathbf{x} \in \mathbb{R}^d} \frac{1}{N} \sum_{i=1}^{N} F_i(\mathbf{x}).$$

One can check that

$$\mathbf{x}^{\mathrm{Avg}} = \left(\sum_{i=1}^{N} \Xi_i^\top \Xi_i\right)^{-1} \sum_{i=1}^{N} \Xi_i^\top (\mathbf{y}_i + \mathbf{A}_i)$$

$$= \left(\sum_{i=1}^{N} \Xi_i^\top \Xi_i\right)^{-1} \sum_{i=1}^{N} \Xi_i^\top \Xi_i \hat{\mathbf{x}}_i + \Lambda, \tag{142}$$

where

$$\Lambda = \left(\sum_{i=1}^{N} \Xi_i^\top \Xi_i\right)^{-1} \sum_{i=1}^{N} \Xi_i^\top \mathbf{A}_i,$$

and $\hat{\mathbf{x}}_i = \mathrm{argmin}_{\mathbf{x} \in \mathbb{R}^d} F_i(\mathbf{x})$ is the solution on client $i$.

*FedEBA+:* For our method FedEBA+, the solution of the global model is

$$\mathbf{x}^{\mathrm{EBA+}} = \mathrm{argmin}_{\mathbf{x} \in \mathbb{R}^d} \sum_{i=1}^{N} p_i F_i(\mathbf{x})$$

$$= \left(\sum_{i=1}^{N} p_i \Xi_i^\top \Xi_i\right)^{-1} \sum_{i=1}^{N} p_i \Xi_i^\top \Xi_i \hat{\mathbf{x}}_i + \hat{\Lambda}, \tag{143}$$

where $p_i \propto \exp(F_i(\mathbf{x}_i^\star)/\tau)$, and

$$\hat{\Lambda} = \left(\sum_{i=1}^{N} p_i \Xi_i^\top \Xi_i\right)^{-1} \sum_{i=1}^{N} p_i \Xi_i^\top \mathbf{A}_i.$$

Following the setting of (Lin et al., 2022), to make the calculations clean, we assume $\Xi_i^\top \Xi_i = n b_i \boldsymbol{I}_d$. Then the solutions of different methods can be simplified as

- FedAvg:

$$\mathbf{x}^{\mathrm{Avg}} = \frac{\sum_{i=1}^{N} b_i (\hat{\mathbf{x}}_i + \mathbf{A}_i)}{\sum_{i=1}^{N} b_i}.$$

- FedEBA+:

$$\mathbf{x}^{\mathrm{EBA+}} = \frac{\sum_{i=1}^{N} b_i p_i (\hat{\mathbf{x}}_i + \mathbf{A}_i)}{\sum_{i=1}^{N} b_i p_i}.$$

**Test Loss.** We compute the test losses of different methods. In this part, we assume $b_i = b$ to make calculations clean. This is reasonable since we often normalize the data.

Recall that the dataset on client $i$ is $(\boldsymbol{\Xi}_i, \mathbf{y}_i)$, where $\boldsymbol{\Xi}_i$ is fixed and $\mathbf{y}_i$ follows Gaussian distribution $\mathcal{N}(\boldsymbol{\Xi}_i \mathbf{x}_i^\star, \sigma_2^2 \boldsymbol{I}_n)$. Then the data heterogeneity across clients only lies in the heterogeneity of $\mathbf{x}_i^\star$. Besides, since the distribution of $\Lambda$ also follows Gaussian distribution $\mathcal{N}(0, \sigma_1^2 \boldsymbol{I}_n)$, $\mathbf{x}_i^\star + \mathbf{A}_i$ follows from $\mathcal{N}(\boldsymbol{\Xi}_i \mathbf{x}_i^\star, \sigma^2 \boldsymbol{I}_n)$, where $\sigma^2 = \sigma_1^2 + \sigma_2^2$. Then, we can obtain the distribution of the solutions of different methods.

Let $\overline{\mathbf{x}}^\star = \frac{1}{N} \sum_{i=1}^N \mathbf{x}_i^\star$. We have

- FedAvg:

$$\mathbf{x}^{\mathrm{Avg}} \sim \mathcal{N}\left(\overline{\mathbf{x}}^\star, \frac{\sigma^2}{bNn} \boldsymbol{I}_d\right).$$

- FedEBA+:

$$\mathbf{x}^{\mathrm{EBA+}} \sim \mathcal{N}\left(\tilde{\mathbf{x}}^\star, \sum_{i=1}^N p_i^2 \frac{\sigma^2}{bn} \boldsymbol{I}_d\right), \quad \tilde{\mathbf{x}}^\star = \sum_{i=1}^N p_i \mathbf{x}_i^\star.$$

Since $\boldsymbol{\Xi}_i$ is fixed, we assume the test data is $(\boldsymbol{\Xi}_i, \mathbf{y}_i')$, where $\mathbf{y}_i' = \boldsymbol{\Xi}_i \mathbf{x}_i^\star + \mathbf{z}_i'$ with $\mathbf{z}_i' \sim \mathcal{N}(\mathbf{0}_n, \sigma_z^2 \boldsymbol{I}_n)$ independent of $\mathbf{z}_i$. Then the test loss on client $i$ is defined as:

$$F_i^{\mathrm{te}}(\mathbf{x}) = \frac{1}{2n} \mathbb{E} \left\| \boldsymbol{\Xi}_i \mathbf{x} + \mathbf{A}_i - \mathbf{y}_i' \right\|_2^2 \tag{144}$$

$$= \frac{1}{2n} \mathbb{E} \left\| \boldsymbol{\Xi}_i \mathbf{x} + \mathbf{A}_i - (\boldsymbol{\Xi}_i \mathbf{x}_i^\star + \mathbf{z}_i') \right\|_2^2 \tag{145}$$

$$= \frac{\tilde{\sigma}^2}{2} + \frac{1}{2n} \mathbb{E} \left\| \boldsymbol{\Xi}_i (\mathbf{x} - \mathbf{x}_i^\star) \right\|_2^2 \tag{146}$$

$$= \frac{\tilde{\sigma}^2}{2} + \frac{b}{2} \mathbb{E} \left\| \mathbf{x} - \mathbf{x}_i^\star \right\|_2^2 \tag{147}$$

$$= \frac{\tilde{\sigma}^2}{2} + \frac{b}{2} \operatorname{tr}(\operatorname{var}(\mathbf{x})) + \frac{b}{2} \left\| \mathbb{E}\mathbf{x} - \mathbf{x}_i^\star \right\|_2^2. \tag{148}$$

where $\tilde{\sigma}$ is a Gaussian variance, which comes from the fact that both $\mathbf{A}_i$ and $\mathbf{z}_i'$ follow Gaussian distribution with mean $0$.

Therefore, for different methods, we can compute that

$$F_i^{\mathrm{te}}\left(\mathbf{x}^{\mathrm{Avg}}\right) = \frac{\tilde{\sigma}^2}{2} + \frac{\tilde{\sigma}^2 d}{2Nn} + \frac{b}{2} \left\| \overline{\mathbf{x}}^\star - \mathbf{x}_i^\star \right\|_2^2, \tag{149}$$

$$F_i^{\mathrm{te}}\left(\mathbf{x}^{\mathrm{EBA+}}\right) = \frac{\tilde{\sigma}^2}{2} + \sum_{k=1}^N p_k^2 \frac{\tilde{\sigma}^2 d}{2n} + \frac{b}{2} \left\| \tilde{\mathbf{x}}^\star - \mathbf{x}_i^\star \right\|_2^2. \tag{150}$$

Define $\operatorname{var}$ as the variance operator. Then we give the formal version of Theorem 5.4.

The variance of test losses on different clients of different aggregation methods are as follows:

$$V^{\mathrm{Avg}} = \operatorname{var}\left(F_i^{\mathrm{te}}\left(\mathbf{x}^{\mathrm{Avg}}\right)\right) = \frac{b^2}{4} \operatorname{var}\left(\left\| \overline{\mathbf{x}}^\star - \mathbf{x}_i^\star \right\|_2^2\right), \tag{151}$$

$$V^{\mathrm{EBA+}} = \operatorname{var}\left(F_i^{\mathrm{te}}\left(\mathbf{x}^{\mathrm{EBA+}}\right)\right) = \frac{b^2}{4} \operatorname{var}\left(\left\| \tilde{\mathbf{x}}^\star - \mathbf{x}_i^\star \right\|_2^2\right). \tag{152}$$

Based on a simple fact, assigning larger weights to smaller values and smaller weights to larger values gives a variance no larger than that of the uniform distribution. This means $V^{\mathrm{EBA+}} \leq V^{\mathrm{Avg}}$.

Formally, let $A_i = \|\tilde{\mathbf{x}}^\star - \mathbf{x}_i^\star\|^2$. From equation (150), we know that $F_i^{\mathrm{te}}(\mathbf{x}^{\mathrm{EBA+}}) \propto A_i$, and $p_i \propto F_i$. Thus, we know $p_i \propto A_i$.

Then, we consider the expression of $V^{\text{EBA+}} = \frac{b^2}{4} \text{var}(A_i)$. Assume $A_1 > A_2 > \cdots > A_N$, then the corresponding aggregation probability distribution is $p_1 > p_2 > \cdots > p_N$.

We show the analysis of variance with set size 2, while the analysis can be extended to larger client numbers. For FedEBA+, we have

$$\text{var}(A_i) = \sum_{i=1}^{2} p_i \left( A_i - \sum_{j=1}^{2} p_j A_j \right)^2 \tag{153}$$

$$= p_1 (A_1 - (p_1 A_1 + p_2 A_2))^2 + p_2 (A_2 - (p_1 A_1 + p_2 A_2))^2 \tag{154}$$

$$= p_1 p_2 (A_1 - A_2)^2, \tag{155}$$

where the last equality follows from $p_1 + p_2 = 1$.

According to our previous analysis, $p_1 > p_2$ while $A_1 > A_2$. By the Cauchy-Schwarz inequality, one can prove that $p_1 p_2 \leq \frac{1}{4}$, where $\frac{1}{4}$ comes from uniform aggregation. Therefore, we prove that $V^{\text{EBA+}} \leq V^{\text{Avg}}$.

### J.3 Fairness Analysis by Smooth and Strongly Convex Loss Functions

In this section, we define the test loss on client $i$ as $L_i(x)$, to distinguish it from the training loss $F_i(x)$.

To extend the analysis to a more general case, we first introduce the following assumptions:

**Assumption 5** (Bounded gradients and strongly convex loss functions)**.** *The test loss function $L_i(x)$ for each client has bounded gradient norm:*

$$\|\nabla L_i(x)\|_2 \leq L_{\text{smooth}}, \tag{156}$$

*and is $\mu$-strongly convex:*

$$L_i(y) \geq L_i(x) + \langle \nabla L_i(x), y - x \rangle + \frac{1}{2}\mu \|y - x\|^2. \tag{157}$$

The variance of FedAvg with $N$ clients' losses can be formulated as:

$$V_N^{\text{Avg}} = \frac{1}{N} \sum_{i=1}^{N} L_i^2(x) - \left( \frac{1}{N} \sum_{i=1}^{N} L_i(x) \right)^2. \tag{158}$$

For FedEBA+, the variance can be formulated with a similar form, only different in the client's loss $L_i(\tilde{x})$, abbreviated as $\tilde{L}_i$. Then, the variance of FedEBA+ with $N$ clients can be formulated as:

$$V_N^{\text{EBA+}} = \frac{1}{N} \sum_{i=1}^{N} \tilde{L}_i^2 - \left( \frac{1}{N} \sum_{i=1}^{N} \tilde{L}_i \right)^2. \tag{159}$$

When the client number is $N + 1$, abbreviating FedAvg's loss $L_i(x)$ as $L_i$, we have

$$V_{N+1}^{\text{Avg}} = \frac{1}{N+1} \sum_{i=1}^{N+1} L_i^2 - \left( \frac{1}{N+1} \sum_{i=1}^{N+1} L_i \right)^2 \tag{160}$$

$$= \frac{N}{N+1} V_N^{\text{Avg}} + \frac{\sum_{i=1}^{N} (L_i - L_{N+1})^2}{(N+1)^2}. \tag{161}$$

We start proving $V_N^{\text{Avg}} \geq V_N^{\text{EBA+}}$ for all $N$ by considering a special case with two clients. There are two clients, Client 1 and Client 2, each with local model $x_1, x_2$ and training loss $F_1(x_1)$ and $F_2(x_2)$.

In this analysis, we assume Client 2 to be the *outlier*, which means the client's optimal parameter and model parameter distribution is far away from Client 1. In particular, $\mu_2 >> L_{\text{smooth}}^1$.

The global model starts with $x = 0$, and after enough local training updates, the models $x_1, x_2$ will converge to their personal optima $x_1^*, x_2^*$. Without loss of generality, we let Client 1 have $F_1(x_1^*) = 0$ and Client 2 have $F_2(x_2^*) = a > 0$. Let $x_1^* < x_2^*$.

Based on the proposed aggregation $p_i \propto \exp(F_i(x)/\tau)$, we can derive the aggregated global model $\tilde{x}$ of FedEBA+ as:

$$\tilde{x} = p_1 x_1^* + p_2 x_2^* = \frac{x_1^* + e^a x_2^*}{e^a + 1}. \tag{162}$$

For FedAvg, the aggregated global model $\overline{x}$ is:

$$\overline{x} = \frac{x_1^* + x_2^*}{2}. \tag{163}$$

For FedEBA+, the test losses of Client 1 and Client 2 are $\tilde{L}_1 = L_1(\tilde{x})$ and $\tilde{L}_2 = L_2(\tilde{x})$, respectively. The corresponding variance is

$$V_2^{\text{EBA+}} = \frac{1}{4}(\tilde{L}_1 - \tilde{L}_2)^2.$$

For FedAvg, the test losses of Client 1 and Client 2 are $\overline{L}_1 = L_1(\overline{x})$ and $\overline{L}_2 = L_2(\overline{x})$, respectively. The corresponding variance is

$$V_2^{\text{Avg}} = \frac{1}{4}(\overline{L}_1 - \overline{L}_2)^2.$$

Since Client 2 is an outlier with $F_2(x_2^*) > 0$ and $x_1^* < x_2^*$, we can conclude that $F_2(x)$ is monotonically decreasing on $(x_1^*, x_2^*)$, and $F_1(x)$ is monotonically increasing on $(x_1^*, x_2^*)$. Besides, without loss of generality, since $\nabla F_1(x) \leq L_{\text{smooth}}^1 << \mu_2$, we can let $\mu = \frac{a}{x_2^* - x_1^*}$.

Thus, we have $\frac{a}{x_2^* - x_1^*} > \nabla F_1(x_2^*)$. According to the property of calculus, we can check that $F_2(x) - F_1(x) > 0$ is monotonically decreasing on $(x_1^*, x_2^*)$.

Since

$$x_2^* - \tilde{x} = \frac{x_2^* - x_1^*}{e^a + 1} \leq x_2^* - \overline{x} = \frac{x_2^* - x_1^*}{2}, \tag{164}$$

we have

$$(F_2(\tilde{x}) - F_1(\tilde{x}))^2 \leq (F_2(\overline{x}) - F_1(\overline{x}))^2.$$

So far, we have proved $V_2^{\text{EBA+}} \leq V_2^{\text{Avg}}$.

To extend the analysis to arbitrary $N$, we utilize mathematical induction.

Assume $V_N^{\text{EBA+}} \leq V_N^{\text{Avg}}$. We need to derive $V_{N+1}^{\text{EBA+}} \leq V_{N+1}^{\text{Avg}}$.

Consider a similar scenario as we analyze with two clients. We assume Client $N + 1$ to be an outlier, which means the client's optimal value and parameter distribution are far away from other clients. In particular, $\mu_{N+1} >> L_{\text{smooth}}^{\text{others}}$. Without loss of generality, let the optimal value $F_{N+1}(x_{N+1}^*)$ for Client $N + 1$ be $a$, and those of the others be zero.

Again, the global model starts with $x = 0$, and after enough local training updates, the models will converge to their personal optima $x_1^*, x_2^*, \ldots, x_{N+1}^*$, with $x_{N+1}^* > x_{\text{others}}^*$.

By (161), we have:

$$V_{N+1}^{\text{Avg}} = \frac{N}{N+1} V_N^{\text{Avg}} + \frac{\sum_{i=1}^{N}(\overline{L}_i - \overline{L}_{N+1})^2}{(N+1)^2}, \tag{165}$$

where $\overline{L}_i$ is the test loss of client $i$ after average aggregation, and

$$V_{N+1}^{\text{EBA+}} = \frac{N}{N+1} V_N^{\text{EBA+}} + \frac{\sum_{i=1}^{N}(\tilde{L}_i - \tilde{L}_{N+1})^2}{(N+1)^2}. \tag{166}$$

Since we know $V_N^{\text{EBA}+} \leq V_N^{\text{Avg}}$, as long as we have

$$\frac{\sum_{i=1}^{N}(\tilde{L}_i - \tilde{L}_{N+1})^2}{(N+1)^2} \leq \frac{\sum_{i=1}^{N}(\overline{L}_i - \overline{L}_{N+1})^2}{(N+1)^2},$$

we can finish the proof.

Consider an arbitrary client $i \in [N]$. Since we already know $F_{N+1}(x_{N+1}^*) = a > F_i(x_i^*) = 0$, the expression for $\tilde{x}$ is

$$\tilde{x} = \sum_{i=1}^{N+1} p_i x_i^* = \frac{1}{N + e^a}\sum_{i=1}^{N} x_i^* + \frac{e^a}{N + e^a} x_{N+1}^*. \tag{167}$$

For FedAvg,

$$\overline{x} = \frac{1}{N+1}\sum_{i=1}^{N+1} x_i^*. \tag{168}$$

Following the same analysis on Client $i$ and Client $N+1$, we can conclude that $F_{N+1}(x) - F_i(x) > 0$ is monotonically decreasing on $(x_i^*, x_{N+1}^*)$.

Since

$$x_{N+1}^* - \tilde{x} = \frac{N x_{N+1}^* - \sum_{i=1}^{N} x_i^*}{e^a + N} \tag{169}$$

$$\leq x_{N+1}^* - \overline{x} = \frac{N x_{N+1}^* - \sum_{i=1}^{N} x_i^*}{N+1}, \tag{170}$$

we have

$$(F_{N+1}(\tilde{x}) - F_i(\tilde{x}))^2 \leq (F_{N+1}(\overline{x}) - F_i(\overline{x}))^2, \quad \forall i \in [N].$$

Therefore, we have

$$\frac{\sum_{i=1}^{N}(\tilde{L}_i - \tilde{L}_{N+1})^2}{(N+1)^2} \leq \frac{\sum_{i=1}^{N}(\overline{L}_i - \overline{L}_{N+1})^2}{(N+1)^2}.$$

So far, we have proved $V_{N+1}^{\text{EBA}+} \leq V_{N+1}^{\text{Avg}}$.

According to mathematical induction, we prove $V_N^{\text{EBA}+} \leq V_N^{\text{Avg}}$ for arbitrary client number $N$ under the smooth and strongly convex setting.

# K  Pareto-optimality Analysis

In addition to variance, *Pareto-optimality* can serve as another metric to assess fairness, as suggested by several studies (Wei & Niethammer, 2022; Hu et al., 2022). This metric achieves equilibrium by reaching each client's optimal performance without hindering others (Guardieiro et al., 2023). We prove that FedEBA+ achieves Pareto optimality through the entropy-based aggregation strategy.

**Definition K.1** (Pareto optimality). *Suppose we have a group of $m$ clients in FL, and each client $i$ has a performance score $f_i$. Pareto optimality happens when we can't improve one client's performance without making someone else's worse: $\forall i \in [1, m], \exists j \in [1, m], j \neq i$ such that $f_i \leq f_i'$ and $f_j > f_j'$, where $f_i'$ and $f_j'$ represent the improved performance measures of participants $i$ and $j$, respectively.*

In the following proposition, we show that FedEBA+ satisfies Pareto optimality. '

**Proposition K.2** (Pareto optimality.). *The proposed maximum entropy model $\mathbb{H}(p_i)$ is proven to be monotonically increasing under the given constraints, ensuring that the aggregation strategy $\varphi(p) = \arg\max_{p \in \mathcal{P}} h(p(f))$ is Pareto optimal. Here, $p(f)$ is the aggregation weights $p = [p_1, p_2, \ldots, p_m]$ of the loss function $f = [f_1, f_2, \ldots, f_m]$, and $h(\cdot)$ represents the entropy function. The proof can be found in Appendix K.*

In this following, we demonstrate the Proposition K.2. In particular, we consider the degenerate setting of FedEBA+ where the parameter $\alpha = 0$. We first provide the following lemma that illustrates the correlation between Pareto optimality and monotonicity.

**Lemma K.3** (Property 1 in (Sampat & Zavala, 2019).)**.** *The allocation strategy* $\varphi(p) = \arg\max\limits_{p \in \mathcal{P}} h(p(f))$ *is Pareto optimal if $h$ is a strictly monotonically increasing function.*

In order for this paper to be self-contained, we restate the proof of Property 1 in (Sampat & Zavala, 2019) here:

**Proof Sketch:** We prove the result by contradiction. Consider that $p^* = \varphi(\mathcal{P})$ is not Pareto optimal; thus, there exists an alternative $p \in \mathcal{P}$ such that

$$\sum_i p_i f_i = \frac{\sum_i p_i \log p_i}{Z} \geq \sum_i p_i^* f_i = \frac{\sum_i p_i^* \log p_i^*}{Z} \,, \tag{171}$$

where $Z > 0$ is a constant. Since $h(p)$ is a strictly monotonically increasing function, we have $h(p) > h(p^*)$. This is a contradiction because $h^*$ maximizes $h(\cdot)$.

According to the above lemma, to show our algorithm achieves Pareto-optimal, we only need to show it is monotonically increasing.

Recall the objective of maximum entropy:

$$\mathbb{H}(p) = -\sum p(x) log(p(x)) \,, \tag{172}$$

subject to certain constraints on the probabilities $p(x)$.

To show that the proposed aggregation strategy is monotonically increasing, we need to prove that if the constraints on the probabilities $p(x)$ are relaxed, then the maximum entropy of the aggregation probability increases.

One way to do this is to use the properties of the logarithm function. The logarithm function is strictly monotonically increasing. This means that for any positive real numbers a and b, if $a \leq b$, then $\log(a) \leq \log(b)$.

Now, suppose that we have two sets of constraints on the probabilities $p(x)$, and that the second set of constraints is a relaxation of the first set. This means that the second set of constraints allows for a larger set of probability distributions than the first set of constraints.

If we maximize the entropy subject to the first set of constraints, we get some probability distribution $p(x)$. If we then maximize the entropy subject to the second set of constraints, we get some probability distribution $q(x)$ such that $p(x) \leq q(x)$ for all x.

Using the properties of the logarithm function and the definition of the entropy, we have:

$$H(p(x)) = -\sum (p(x) \log(p(x))) \tag{173}$$

$$\leq -\sum (p(x) \log(q(x))) \tag{174}$$

$$= -\sum ((p(x)/q(x)) q(x) \log(q(x))) \tag{175}$$

$$= H(q(x)) - \sum ((\frac{p(x)}{q(x)} q(x) \log(p(x)/q(x)))) \tag{176}$$

$$\leq H(q(x)) \,. \tag{177}$$

This means that the entropy $H(q(x))$ is greater or equal to $H(p(x))$ when the second set of constraints is a relaxation of the first set of constraints. As the entropy increases when the constraints are relaxed, the maximum entropy-based aggregation strategy is monotonically increasing.

Up to this point, we proved that our proposed aggregation strategy is monotonically increasing. Combined with the Lemma K.3, we can prove that equation (5) is Pareto optimal.

## L    Uniqueness of our Aggregation Strategy

In this section, we prove the proposed entropy-based aggregation strategy is unique.

Recall our optimization objective of constrained maximum entropy:

$$H(p(x)) = - \sum (p(x) \log(p(x))), \tag{178}$$

subject to certain contains, which is $\sum_i p_i = 1, p_i \geq 0, \sum_i p_i F_i = \tilde{f}$.

Based on equation 5, and writing the entropy in matrix form, we have:

$$H_{i,j}(p) = \begin{cases} p_i(\frac{F_i}{\tau} - \log \sum e^{F_i/\tau}) = -ap_i & \text{for } i = j \\ 0 & \text{otherwise} \end{cases}, \tag{179}$$

where $a$ is some positive constant.

For every non-zero vector $v$ we have that:

$$v^T H(p)v = \sum_{j \in \mathcal{N}} -ap_i v_j^2 < 0. \tag{180}$$

The Hessian is thus negative definite.

Furthermore, since the constraints are linear, both convex and concave, the constrained maximum entropy function is strictly concave and thus has a unique global maximum.

## M    Experiment Details

### M.1    Experimental Environment

For all experiments, we use NVIDIA GeForce RTX 3090 GPUs. Each simulation trail with 2000 communication rounds and 5 random seeds.

**Baselines**    We compared several advanced FL fairness algorithms with FedEBA+, including FedAvg (McMahan et al., 2017), FedSGD (McMahan et al., 2016), AFL (Mohri et al., 2019),q-FFL (Li et al., 2019a),FedMGDA+(Hu et al., 2022),PropFair (Zhang et al., 2023), TERM (Li et al., 2020a), FOCUS (Chu et al., 2023), Ditto (Li et al., 2021),FedFV (Wang et al., 2021), lp-proj (Lin et al., 2022) and AAggFF (Hahn et al., 2024). For AAggFF, we used the Normal CDF transformation while keeping other settings same as in the paper.

**Hyper-parameters**    As shown in Table 3, we tuned some hyper-parameters of baselines to ensure the performance in line with the previous studies and listed parameters used in FedEBA+. All experiments are running over 2000 rounds for the local epoch ($K = 10$) with local batch size $B = 50$ for MNIST and $B = 64$ for CIFAR datasets. The learning rate remains the same for different methods, that is $\eta = 0.1$ on MNIST, Fasion-MNIST, CIFAR-10, $\eta = 0.05$ on Tiny-ImageNet and $\eta = 0.01$ on CIFAR-100 with decay rate $d = 0.999$.

## N    Additional Experiment Results

### N.1    Fairness Evaluation of FedEBA+

In this section, we provide additional experimental results to illustrate that FedEBA+ is superior to other baselines.

While FedEBA+ explores the theoretical limits, it indeed introduces extra communication and computation costs. Prac-FedEBA+ (proposed in Proposition 4.3) is specifically designed for resource-constrained edge devices, which matches FedAvg and still maintains a decent performance.

As detailed in Table 9, FedEBA+ consistently outperforms FedAvg in terms of fairness improvement. Specifically, the improvement rates achieved by FedEBA+ over FedAvg are 33%, 33.4%, and 34.7% when the local step size $K$ is set to 1, 5, and 10, respectively. This increasing trend unequivocally demonstrates that higher $K$ values benefit FedEBA+ disproportionately more than FedAvg in reducing outcome variance (fairness).

*Table 6.* Hyperparameters of baselines.

| Algorithm | Hyper-parameters |
|---|---|
| q-FFL | $q \in \{0.001, 0.01, 0.1, 0.5, 10, 15\}$ |
| PropFair | $M \in \{0.2, 2.0, 5.0\}, \epsilon = 0.2$ |
| AFL | $\lambda \in \{0.1, 0.5, 0.7\}$ |
| TERM | $T \in \{0.1, 0.5, 0.8, 1, 5\}$ |
| FedMGDA+ | $\epsilon \in \{0, 0.03, 0.08, 0.1, 1.0\}$ |
| FedProx | $q = \{0.001, 0.001, 0.1, 0.5, 10.0, 15.0\}$ |
| Ditto | $\lambda = \{0.0, 0.5\}$ |
| FOCUS | $\beta = 0.5, cluster = 2$ |
| lp-proj | $local model dim = 60, \lambda = 15, p = 1.0$ |
| FedFV | $\alpha \in \{0.1, 0.2, 0.5\}, \tau \in \{0, 1, 10\}$ |
| FedEBA+ | $\tau \in \{0.1, 0.5, 1.0, 5.0, 10.0, 20.0\}, \alpha \in \{0.0, 0.1, 0.5, 0.9\}$ |

*Table 7.* **Performance of algorithms on FashionMNIST and CIFAR-10.** We report the accuracy of global model, variance fairness, worst 5%, and best 5% accuracy. The data is divided into 100 clients, with 10 clients sampled in each round. All experiments are running over 2000 rounds for a single local epoch ($K = 10$) with local batch size $= 50$, and learning rate $\eta = 0.1$. The reported results are averaged over 5 runs with different random seeds. We highlight the best and the second-best results by using **bold font** and blue text.

| Algorithm | FashionMNIST (MLP) | | | | CIFAR-10 (CNN) | | | |
|---|---|---|---|---|---|---|---|---|
| | Global Acc. | Var. | Worst 5% | Best 5% | Global Acc. | Var. | Worst 5% | Best 5% |
| FedAvg | $86.49 \pm 0.09$ | $62.44 \pm 4.55$ | $71.27 \pm 1.14$ | $95.84 \pm 0.35$ | $67.79 \pm 0.35$ | $103.83 \pm 10.46$ | $45.00 \pm 2.83$ | $85.13 \pm 0.82$ |
| FedSGD | $83.79 \pm 0.28$ | $81.72 \pm 0.26$ | $61.19 \pm 0.30$ | $96.60 \pm 0.20$ | $67.48 \pm 0.37$ | $95.79 \pm 4.03$ | $48.70 \pm 0.9$ | $84.20 \pm 0.40$ |
| q-FFL$|_{q=0.001}$ | $87.05 \pm 0.25$ | $66.67 \pm 1.39$ | $72.11 \pm 0.03$ | $95.09 \pm 0.71$ | $68.53 \pm 0.18$ | $97.42 \pm 0.79$ | $48.40 \pm 0.60$ | $84.70 \pm 1.31$ |
| q-FFL$|_{q=0.01}$ | $86.62 \pm 0.03$ | $58.11 \pm 3.21$ | $71.36 \pm 1.98$ | $95.29 \pm 0.27$ | $68.85 \pm 0.03$ | $95.17 \pm 1.85$ | $48.20 \pm 0.80$ | $84.10 \pm 0.10$ |
| q-FFL$|_{q=0.5}$ | $86.57 \pm 0.19$ | $54.91 \pm 2.82$ | $70.88 \pm 0.98$ | $95.06 \pm 0.17$ | $68.76 \pm 0.22$ | $97.81 \pm 2.18$ | $48.33 \pm 0.84$ | $84.51 \pm 1.33$ |
| q-FFL$|_{q=10.0}$ | $77.29 \pm 0.20$ | $47.20 \pm 0.82$ | $61.99 \pm 0.48$ | $92.25 \pm 0.57$ | $40.78 \pm 0.06$ | $85.93 \pm 1.48$ | $22.70 \pm 0.10$ | $56.40 \pm 0.21$ |
| q-FFL$|_{q=15.0}$ | $75.77 \pm 0.42$ | $46.58 \pm 0.75$ | $61.63 \pm 0.46$ | $89.60 \pm 0.42$ | $36.89 \pm 0.14$ | $79.65 \pm 5.17$ | $19.30 \pm 0.70$ | $51.30 \pm 0.09$ |
| FedMGDA+$|_{\epsilon=0.0}$ | $86.01 \pm 0.31$ | $58.87 \pm 3.23$ | $71.49 \pm 0.16$ | $95.45 \pm 0.43$ | $67.16 \pm 0.33$ | $97.33 \pm 1.68$ | $46.00 \pm 0.79$ | $83.30 \pm 0.10$ |
| FedMGDA+$|_{\epsilon=0.03}$ | $84.64 \pm 0.25$ | $57.89 \pm 6.21$ | $73.49 \pm 1.17$ | $93.22 \pm 0.20$ | $65.19 \pm 0.87$ | $89.78 \pm 5.87$ | $48.84 \pm 1.12$ | $81.94 \pm 0.67$ |
| FedMGDA+$|_{\epsilon=0.08}$ | $84.90 \pm 0.34$ | $61.55 \pm 5.87$ | $\mathbf{73.64 \pm 0.85}$ | $92.78 \pm 0.12$ | $65.06 \pm 0.69$ | $93.70 \pm 14.10$ | $48.23 \pm 0.82$ | $82.01 \pm 0.09$ |
| AFL$|_{\lambda=0.5}$ | $84.14 \pm 0.18$ | $90.76 \pm 6.13$ | $60.11 \pm 0.69$ | $96.00 \pm 0.09$ | $63.51 \pm 1.22$ | $87.49 \pm 0.58$ | $44.73 \pm 0.90$ | $82.10 \pm 0.62$ |
| AFL$|_{\lambda=0.1}$ | $83.21 \pm 0.31$ | $68.33 \pm 4.53$ | $68.04 \pm 0.35$ | $94.09 \pm 0.21$ | $65.60 \pm 0.14$ | $87.67 \pm 2.39$ | $46.01 \pm 0.40$ | $82.30 \pm 0.12$ |
| PropFair$|_{M=0.2,thres=0.2}$ | $85.51 \pm 0.28$ | $75.27 \pm 5.38$ | $63.60 \pm 0.53$ | $97.60 \pm 0.19$ | $65.79 \pm 0.53$ | $79.67 \pm 5.71$ | $49.88 \pm 0.93$ | $82.40 \pm 0.40$ |
| PropFair$|_{M=5.0,thres=0.2}$ | $84.59 \pm 1.01$ | $85.31 \pm 8.62$ | $61.40 \pm 0.55$ | $96.40 \pm 0.29$ | $66.91 \pm 1.43$ | $78.90 \pm 6.48$ | $50.16 \pm 0.56$ | $85.40 \pm 0.34$ |
| TERM$|_{T=0.1}$ | $84.31 \pm 0.38$ | $73.46 \pm 2.06$ | $68.23 \pm 0.10$ | $94.16 \pm 0.16$ | $65.41 \pm 0.37$ | $91.99 \pm 2.69$ | $49.08 \pm 0.66$ | $81.98 \pm 0.19$ |
| TERM$|_{T=0.5}$ | $82.19 \pm 1.41$ | $87.82 \pm 2.62$ | $62.11 \pm 0.71$ | $93.25 \pm 0.39$ | $61.04 \pm 1.96$ | $96.78 \pm 7.67$ | $42.45 \pm 1.73$ | $80.06 \pm 0.62$ |
| TERM$|_{T=0.8}$ | $81.33 \pm 1.21$ | $95.65 \pm 9.56$ | $56.41 \pm 0.56$ | $92.88 \pm 0.70$ | $59.21 \pm 1.45$ | $82.63 \pm 3.64$ | $41.33 \pm 0.68$ | $77.39 \pm 1.04$ |
| FedFV$|_{\alpha=0.1,\tau_{fv}=1}$ | $86.51 \pm 0.28$ | $49.73 \pm 2.26$ | $71.33 \pm 1.16$ | $95.89 \pm 0.23$ | $68.94 \pm 0.27$ | $90.84 \pm 2.67$ | $50.53 \pm 4.33$ | $86.00 \pm 1.23$ |
| FedFV$|_{\alpha=0.2,\tau_{fv}=0}$ | $86.42 \pm 0.38$ | $52.41 \pm 5.94$ | $71.22 \pm 1.35$ | $95.47 \pm 0.43$ | $68.89 \pm 0.15$ | $82.99 \pm 3.10$ | $50.08 \pm 0.40$ | $86.24 \pm 1.17$ |
| FedFV$|_{\alpha=0.5,\tau_{fv}=10}$ | $86.88 \pm 0.26$ | $47.63 \pm 1.79$ | $71.49 \pm 0.39$ | $95.62 \pm 0.29$ | $69.42 \pm 0.60$ | $78.10 \pm 3.62$ | $52.80 \pm 0.34$ | $85.76 \pm 0.80$ |
| FedFV$|_{\alpha=0.1,\tau_{fv}=10}$ | $86.98 \pm 0.45$ | $56.63 \pm 1.85$ | $66.40 \pm 0.57$ | $\mathbf{98.80 \pm 0.12}$ | $71.10 \pm 0.44$ | $86.50 \pm 7.36$ | $49.80 \pm 0.72$ | $\mathbf{88.42 \pm 0.25}$ |
| FedEBA+$|_{\alpha=0,\tau=0.1}$ | $86.70 \pm 0.11$ | $50.27 \pm 5.60$ | $71.13 \pm 0.69$ | $95.47 \pm 0.27$ | $69.38 \pm 0.52$ | $89.49 \pm 10.95$ | $50.40 \pm 1.72$ | $86.07 \pm 0.90$ |
| FedEBA+$|_{\alpha=0.5,\tau=0.1}$ | $87.21 \pm 0.06$ | $\mathbf{40.02 \pm 1.58}$ | $73.07 \pm 1.03$ | $95.81 \pm 0.14$ | $72.39 \pm 0.47$ | $70.60 \pm 3.19$ | $55.27 \pm 1.18$ | $86.27 \pm 1.16$ |
| FedEBA+$|_{\alpha=0.9,\tau=0.1}$ | $\mathbf{87.50 \pm 0.19}$ | $43.41 \pm 4.34$ | $72.07 \pm 1.47$ | $95.91 \pm 0.19$ | $\mathbf{72.75 \pm 0.25}$ | $\mathbf{68.71 \pm 4.39}$ | $\mathbf{55.80 \pm 1.28}$ | $86.93 \pm 0.52$ |

## N.2  Fairness Evaluation in Different Non-i.i.d. Cases

We adopt two kinds of data splitation strategies to change the degree of non-i.i.d., which are data devided by labels mentioned in the main text, and the data partitioning in deference to the Latent Dirichlet Allocation (LDA) with the Dirichlet parameter

*Table 8.* Comparison of the Algorithms' Communication Costs. $\mathcal{O}(d)$ denotes the size of the model parameters (and gradients), while $\mathcal{O}(1)$ refers to negligible loss values transferred in Prac-FedEBA+. Practical cost and time per round data are derived from experiments using a CNN model on the CIFAR-10 dataset.

| Algorithm | Uplink | Downlink | Total Bytes Transferred | Practical Cost | Practical Time per Round |
|---|---|---|---|---|---|
| FedAvg | $\mathcal{O}(d)$ | $\mathcal{O}(d)$ | $\mathcal{O}(2 \times \mathcal{O}(d))$ | 3.04MB | 1.08s |
| FedEBA+ | $\mathcal{O}(d) + \mathcal{O}(d)$ (Gradient) | $\mathcal{O}(d)$ | $\mathcal{O}(3 \times \mathcal{O}(d))$ | 4.56MB | 1.29s |
| Prac-FedEBA+ | $\mathcal{O}(d)$ | $\mathcal{O}(d)$ | $\mathcal{O}(2 \times \mathcal{O}(d) + \mathcal{O}(1))$ | 3.04MB | 1.08s |

*Table 9.* Performance comparison of the algorithms across different local update step sizes $K$.

| Step Size | Algorithm | Accuracy ↑ | Variance (fairness ↓) | Worst 5% ↑ | Best 5% ↑ |
|---|---|---|---|---|---|
| K=10 | FedAvg | $67.79 \pm 0.35$ | $103.83 \pm 10.46$ | $45.00 \pm 2.83$ | $85.13 \pm 0.82$ |
| | q-FFL | $68.76 \pm 0.22$ | $97.81 \pm 2.18$ | $48.33 \pm 0.84$ | $84.51 \pm 1.33$ |
| | FedEBA+ | $72.75 \pm 0.25$ | $68.71 \pm 4.39$ | $55.80 \pm 1.28$ | $86.93 \pm 0.52$ |
| | Prac-FedEBA+ | $69.83 \pm 0.34$ | $74.16 \pm 1.66$ | $52.40 \pm 0.50$ | $84.10 \pm 0.39$ |
| K=5 | FedAvg | $69.66 \pm 0.18$ | $89.46 \pm 7.12$ | $50.60 \pm 1.67$ | $90.00 \pm 2.02$ |
| | q-FFL | $71.35 \pm 0.41$ | $67.33 \pm 1.96$ | $54.40 \pm 1.24$ | $85.20 \pm 0.20$ |
| | FedEBA+ | $73.94 \pm 0.32$ | $59.05 \pm 2.42$ | $55.62 \pm 0.40$ | $86.00 \pm 0.29$ |
| | Prac-FedEBA+ | $72.20 \pm 0.23$ | $64.84 \pm 1.65$ | $53.83 \pm 0.61$ | $85.84 \pm 0.20$ |
| K=1 | FedAvg | $65.49 \pm 0.28$ | $83.53 \pm 3.23$ | $46.00 \pm 0.60$ | $82.40 \pm 0.38$ |
| | q-FFL | $65.72 \pm 0.18$ | $73.34 \pm 3.50$ | $47.40 \pm 0.55$ | $79.80 \pm 1.08$ |
| | FedEBA+ | $69.76 \pm 0.33$ | $55.94 \pm 4.63$ | $55.20 \pm 1.22$ | $84.20 \pm 0.75$ |
| | Prac-FedEBA+ | $67.21 \pm 0.24$ | $64.47 \pm 3.98$ | $53.25 \pm 0.86$ | $83.02 \pm 0.64$ |

. Based on FedAvg, we have experimented with various data segmentation strategies for FedEBA+ to verify the performance of FedEBA+ for scenarios with different kinds of data held by clients.

### N.3 Privacy Evaluation.

We also evaluate FedEBA+ under privacy preservation. Following (Abadi et al., 2016), we insert Gaussian noise into the intermediate regularization variable $\delta$ with noise standard deviation $\sigma_2 : \tilde{\sigma}_i \leftarrow \sigma_i + \frac{1}{L}\mathcal{N}(0, \sigma_2^2 C_0^2 I)$, where $L$ is the batch size, $\sigma_2$ is the noise parameter, $C_2$ is the clipping constant. The result is shown in Figure 4. With $\sigma_2 \leq 5$, the curves show only marginal reductions without significant performance degradation. However, higher values of $\sigma_2$ risk compromising performance. This suggests that our approach is compatible with a specific threshold of privacy preservation. In addition, Table 13 shows that compared to other fairness baselines, FedEBA+ maintains its fairness and performance advantage when using differential privacy.

### N.4 Ablation Study

**Remark N.1** (The annealing manner for $\tau$). *While we set $\tau$ as a constant in our algorithm, we demonstrate that utilizing an annealing schedule for $\tau$ can further enhance performance. The linear annealing schedule is defined below:*

$$\tau^T = \tau^0/(1 + \kappa(T-1)), \tag{181}$$

*where $T$ is the total communication rounds and hyperparameter $\kappa$ controls the decay rate. There are also concave schedule $\tau^k = \tau^0/(1+\kappa(T-1))^{\frac{1}{2}}$ and convex schedule $\tau^k = \tau^0/(1+\kappa(T-1))^3$. We experiment with different annealing strategies for $\tau$ in Figure 5.*

For the annealing schedule of $\tau$ mentioned above, Figure 5 shows that the annealing schedule has advantages in reducing the variance compared with constant $\tau$. Besides, the global accuracy is robust to the annealing strategy, and the annealing strategy is robust to the initial temperature $T_0$.

*Table 10.* Performance of algorithms+momentum on Fashion-MNIST to show that FedEBA+ is orthogonal to advance optimization methods like momentum  (Karimireddy et al., 2020a), allowing seamless integration. All experiments are running over 2000 rounds on the MLP model for a single local epoch ($K = 10$) with local batch size $= 50$, global momentum $= 0.9$ and learning rate $\eta = 0.1$. The reported results are averaged over 5 runs with different random seeds. We highlight the best and the second-best results by using bold font and blue text.

| Method | Global Acc. | Var. | Worst 5% | Best 5% |
|---|---|---|---|---|
| FedAvg | 86.68± 0.37 | 66.15± 3.23 | 72.18± 0.22 | 96.04±± 0.35 |
| AFL$\|_{\lambda=0.05}$ | 79.68± 0.91 | 55.00± 3.34 | 66.67± 0.12 | 94.00± 0.08 |
| AFL$\|_{\lambda=0.7}$ | 85.41± 0.30 | 63.42±± 1.55 | **73.83± 0.37** | 96.46± 0.12 |
| q-FFL$\|_{q=0.01}$ | 86.82± 0.20 | 64.11± 2.17 | 71.08± 0.16 | 96.29± 0.08 |
| q-FFL$\|_{q=15}$ | 79.59± 0.48 | 62.26± 2.88 | 66.33± 1.14 | 90.07± 0.98 |
| FedMGDA+$\|_{\epsilon=0.0}$ | 82.69± 0.52 | 65.26± 3.81 | 69.63± 1.20 | 92.67± 0.54 |
| PropFair$\|_{M=5,thres=0.2}$ | 85.67± 0.19 | 73.44± 2.44 | 64.59± 0.42 | 97.47± 0.11 |
| FedProx$\|_{\mu=0.1}$ | 86.76± 0.26 | 60.69± 3.07 | 72.67± 0.29 | 95.96± 0.14 |
| TERM$\|_{T=0.1}$ | 84.58± 0.28 | 76.44± 2.50 | 69.52± 0.36 | 94.04± 0.50 |
| FedFV$\|_{\alpha=0.1,\tau=10}$ | 87.46± 0.18 | 58.35± 1.89 | 67.71± 0.56 | **97.79± 0.18** |
| FedEBA+$\|_{\alpha=0.9,T=0.1}$ | **87.67± 0.28** | **46.67± 1.09** | 71.90± 0.70 | 96.26± 0.03 |

*Table 11.* Performance of algorithms+VARP on Fashion-MNIST to show that FedEBA+ is orthogonal to advance optimization methods like VARP (Jhunjhunwala et al., 2022), allowing seamless integration. All experiments are running over 2000 rounds on the MLP model for a single local epoch ($K = 10$) with local batch size $= 50$, global learning rate $= 1.0$ and client learning rate $= 0.1$. The reported results are averaged over 5 runs with different random seeds. We highlight the best and the second-best results by using bold font and blue text.

| Method | Global Acc. | Var. | Worst 5% | Best 5% |
|---|---|---|---|---|
| FedAvg (FedVARP) | 87.12± 0.08 | 59.96± 2.48 | 72.45± 0.26 | 96.09±± 0.27 |
| q-FFL$\|_{q=0.01}$ | 86.73± 0.31 | 62.89± 2.67 | 73.55± 0.11 | 95.54± 0.14 |
| q-FFL$\|_{q=15}$ | 78.98± 0.63 | 58.28± 1.95 | 67.12± 0.97 | 88.42± 0.67 |
| FedFV$\|_{\alpha=0.1,\tau=10}$ | 87.28± 0.10 | 57.90± 1.77 | 67.41± 0.30 | **97.66± 0.06** |
| FedEBA+$\|_{\alpha=0.9,T=0.1}$ | **87.45± 0.18** | **49.91± 2.38** | 71.44± 0.64 | 95.94± 0.09 |

*Table 12.* **Ablation study for Dirichlet parameter** $\alpha$**.** Performance comparison between FedAvg and FedEBA+ on CIFAR-100 using ResNet18 (devided by Dirichlet Distribution with $\alpha \in \{0.1, 0.5, 1.0\}$). We report the global model's accuracy, fairness of accuracy, worst 5% and best 5% accuracy. All experiments are running over 2000 rounds for a single local epoch ($K = 10$) with local batch size $= 64$, and learning rate $\eta = 0.01$. The reported results are averaged over 5 runs with different random seeds.

| Algorithm | Global Acc. | | | Var. | | | Worst 5% | | | Best 5% | | |
|---|---|---|---|---|---|---|---|---|---|---|---|---|
| | $\alpha = 0.1$ | $\alpha = 0.5$ | $\alpha = 1.0$ | $\alpha = 0.1$ | $\alpha = 0.5$ | $\alpha = 1.0$ | $\alpha = 0.1$ | $\alpha = 0.5$ | $\alpha = 1.0$ | $\alpha = 0.1$ | $\alpha = 0.5$ | $\alpha = 1.0$ |
| FedAvg | 30.94±0.04 | 54.69±0.25 | 64.91±0.02 | 17.24±0.08 | 7.92±0.03 | 5.18±0.06 | 0.20±0.00 | 38.79±0.24 | 54.36±0.11 | 65.90±1.48 | 70.10±0.25 | 75.43±0.39 |
| FedEBA+ | 33.39±0.22 | 58.55±0.41 | 65.98±0.04 | 16.92±0.04 | 7.71±0.08 | 4.44±0.10 | 0.95±0.15 | 41.63±0.16 | 58.20±0.17 | 68.51±0.21 | 74.03±0.07 | 74.96±0.16 |

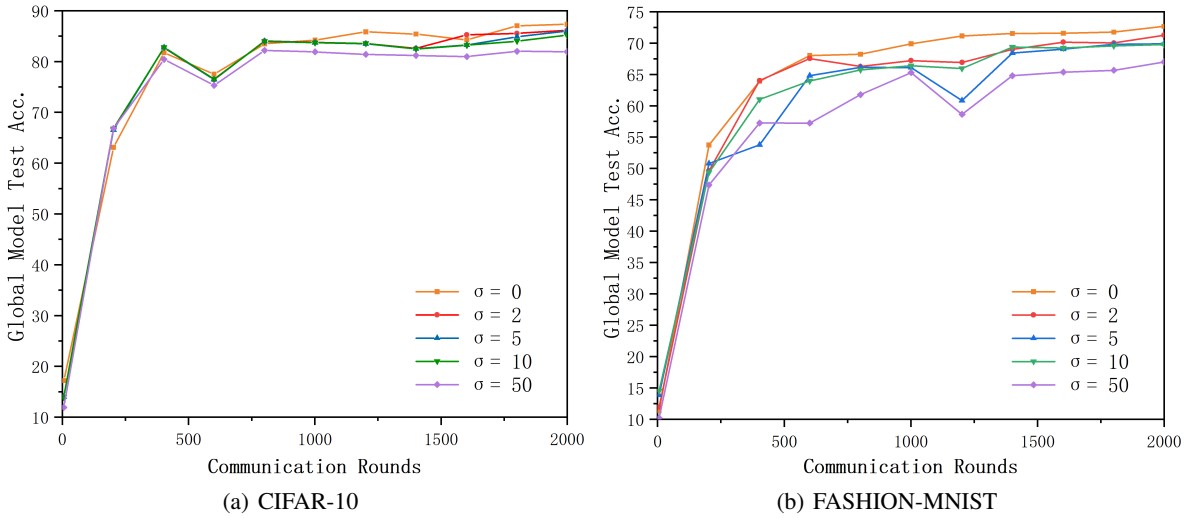

(a) CIFAR-10          (b) FASHION-MNIST

*Figure 4.* Privacy Evaluation of FedEBA+.

*Table 13.* Performance of fairness algorithms under different differential privacy noise $\sigma$.

| noise $\sigma_2$ | Fashion-MNIST | | | | CIFAR10 | | | |
|---|---|---|---|---|---|---|---|---|
| | Global Acc. | Var. | Worst 5% | Best 5% | Global Acc. | Var. | Worst 5% | Best 5% |
| | | | | FedEBA+ | | | | |
| 0 | 87.50±0.19 | 43.41±4.34 | 72.07±1.47 | 95.91±0.19 | 72.75±0.25 | 68.71±4.39 | 55.80±1.28 | 86.93±0.52 |
| 2 | 86.24±0.14 | 75.67±3.40 | 63.67±0.74 | 97.9±0.22 | 70.69±0.40 | 76.25±3.56 | 51.87±0.25 | 86.5±0.24 |
| 5 | 86.01±0.08 | 73.11±2.62 | 64.90±0.94 | 98.0±0.16 | 69.86±0.14 | 76.4±2.38 | 51.20±0.11 | 85.15±0.45 |
| 10 | 85.96±0.08 | 71.52±2.45 | 64.8±1.85 | 97.53±0.34 | 69.48±0.32 | 85.53±2.10 | 49.93±0.77 | 84.53±0.62 |
| 50 | 83.43±0.14 | 79.7±1.18 | 61.37±1.52 | 97.00±0.59 | 67.57±0.68 | 120.83±2.80 | 45.40±0.99 | 86.17±0.33 |
| | | | | FedAvg | | | | |
| 0 | 86.49±0.09 | 62.44±4.55 | 71.27±1.14 | 95.84±0.35 | 67.79±0.35 | 103.83±10.46 | 45.00±2.83 | 85.13±0.82 |
| 2 | 64.20±0.22 | 534.40±1.24 | 7.4±0.2 | 93.2±0 | 45.29±0.81 | 101.04±9.70 | 23.4±0.10 | 68.2±0.33 |
| 5 | 64.14±0.02 | 536.57±2.72 | 7.4±0 | 93.1±0.13 | 45.01±0.33 | 98.38±5.24 | 26.4±1.5 | 66.2±1.2 |
| 10 | 64.10±0.13 | 533.34±4.26 | 7.2±0 | 93.0±0 | 45.45±0.62 | 97.50±4.93 | 26.6±2.2 | 68.0±1.4 |
| 50 | 64.06±0.05 | 533.61±2.40 | 7.55±0.16 | 93.1±0.10 | 45.27±0.92 | 100.54±6.23 | 26.5±1.33 | 66.4±1.4 |
| | | | | qFedAvg | | | | |
| 0 | 86.57±0.19 | 54.91±2.82 | 70.88±0.98 | 95.06±0.17 | 68.76± 0.22 | 97.81±2.18 | 48.33±0.84 | 84.51±1.33 |
| 2 | 64.17±0.02 | 529.99±0.92 | 7.8±0 | 93.2±0 | 43.79±0.70 | 187.79±2.03 | 16.8±0 | 76.14±2.32 |
| 5 | 64.16±0.04 | 530.55±1.17 | 7.6±0 | 93.2±0 | 44.50±0.78 | 191.12±1.70 | 15.4±1.14 | 73.8±1.28 |
| 10 | 64.15±0.03 | 526.82±0.67 | 7.6±0 | 93.2±0 | 43.42±0.80 | 200.31±2.80 | 14.33±1.24 | 73.8±1.14 |
| 50 | 64.21±0.07 | 529.58±0.50 | 7.6±0 | 93.2±0 | 43.92±0.92 | 195.69±3.07 | 15.88±1.30 | 74.2±0.84 |
| | | | | FedMGDA+ | | | | |
| 0 | 84.64±0.25 | 57.89±6.21 | 73.49±1.17 | 93.22±0.20 | 65.19±0.87 | 89.78±5.87 | 48.84±1.12 | 81.94±0.67 |
| 2 | 79.34±0.06 | 112.12±1.49 | 56.67±0.25 | 95.13±0.09 | 43.84±0.22 | 183.39±3.17 | 14.60±1.2 | 70.40±0.4 |
| 5 | 77.13±0.15 | 136.19±1.20 | 51.8±0.40 | 95.00±0.22 | 41.39±0.63 | 96.67±2.88 | 23.2±0.6 | 62.00±0.2 |
| 10 | 71.02±0.01 | 248.45±2.18 | 36.7±0.7 | 93.2±0.13 | 36.75±0.45 | 107.94±4.10 | 16.2±0.34 | 57.00±4.0 |
| 50 | 57.04±0.03 | 754.46±0.81 | 0.2±0 | 93.9±0.1 | 23.08±0.05 | 203.65±3.6 | 0.40±0 | 56.4±0.43 |

For different fairness evaluation metrics, Table 15 demonstrates that in our setting, FedEBA+ exhibits competitive performance under FAA metrics. Instead, FOCUS exhibits a relatively large FAA. This discrepancy arises from the differing settings between ours and FOCUS's. In our scenario, only a subset of clients undergoes training, contrasting with FOCUS's full client participation, consequently leading to subpar clustering performance.

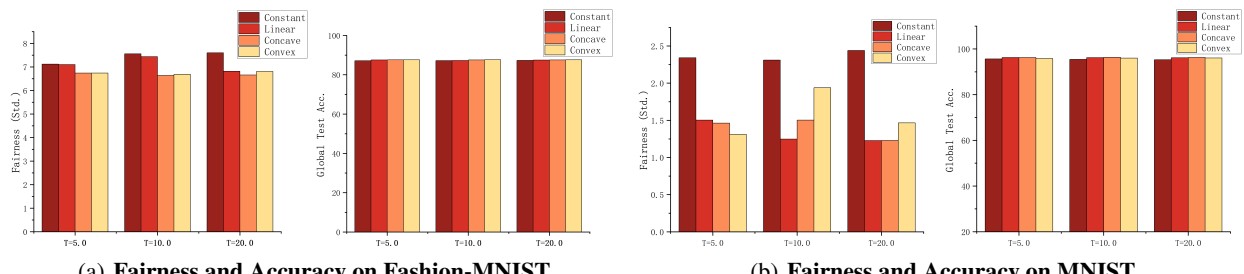

(a) **Fairness and Accuracy on Fashion-MNIST**  (b) **Fairness and Accuracy on MNIST**

*Figure 5.* Ablation study for Annealing schedule $\tau$

*Table 14.* **Ablation study for FedEBA+ on four datasets.** We test the effectiveness of FedEBA+ when decomposing each proposed step, i.e., entropy-based aggregation and alignment update, on different datasets. FedEBA differs from FedAvg only in the aggregation method, and FedEBA+ incorporates the alignment into FedEBA. FedAvg serves as the backbone, FedAvg+① is employed to demonstrate the individual effectiveness of our proposed aggregation step, FedAvg+② is utilized to showcase the individual effectiveness of our proposed alignment step, and FedAvg + ① + ② is used to show the effectiveness of our proposed algorithm, FedEBA+.

| Algorithm | CIFAR-10 (CNN) | | | | FashionMNIST (MLP) | | | |
|---|---|---|---|---|---|---|---|---|
| | Global Acc. ↑ | Var. ↓ | Worst 5% ↑ | Best 5% ↑ | Global Acc.↑ | Var. ↓ | Worst 5% ↑ | Best 5% ↑ |
| FedAvg | 67.79±0.35 | 103.83±10.46 | 45.00±2.83 | 85.13±0.82 | 86.49±0.09 | 62.44±4.55 | 71.27±1.14 | 95.84±0.35 |
| FedAvg+① | 69.38±0.52 | 89.49±10.95 | 50.40±1.72 | 86.07±0.90 | 86.70±0.11 | 50.27±5.60 | 71.13±0.69 | 95.47±0.27 |
| FedAvg+② | 72.04±0.51 | 75.73±4.27 | 53.45±1.25 | 87.33±0.23 | 87.42± 0.09 | 60.08±7.30 | 69.12±1.23 | 97.8±0.19 |
| FedAvg+①+② | 72.75±0.25 | 68.71±4.39 | 55.80±1.28 | 86.93±0.52 | 87.50±0.19 | 43.41±4.34 | 72.07±1.47 | 95.91±0.19 |

| Algorithm | CIFAR-100 (Resnet-18) | | | | Tiny-ImageNet (MobileNet-2) | | | |
|---|---|---|---|---|---|---|---|---|
| | Global Acc. ↑ | Var. ↓ | Worst 5% ↑ | Best 5% ↑ | Global Acc.↑ | Var. ↓ | Worst 5% ↑ | Best 5% ↑ |
| FedAvg | 30.94±0.04 | 17.24±0.08 | 0.20±0.00 | 65.90±1.48 | 61.99±0.17 | 19.62±1.12 | 53.60±0.06 | 71.18±0.13 |
| FedAvg+① | 31.23±0.25 | 14.14±0.09 | 0.99±0.30 | 66.54±0.40 | 63.34±0.25 | 15.29±1.36 | 54.17±0.04 | 70.98±0.10 |
| FedAvg+② | 31.78±0.25 | 17.02±0.08 | 0.41±0.03 | 65.94±0.20 | 63.46±0.04 | 14.52±0.21 | 54.36±0.03 | 71.13±0.03 |
| FedAvg+①+② | 31.98±0.30 | 13.75±0.16 | 1.12±0.05 | 67.94±0.54 | 63.75±0.09 | 13.89±0.72 | 55.64±0.18 | 70.93±0.22 |

*Table 15.* **Performance of Fair FL Algorithms under FAA:** We present results under the FAA metric, as utilized in (Chu et al., 2023), where FAA represents the discrepancy in excess loss across clients. The algorithms are tested on the FashionMNIST and CIFAR-10 datasets, with 10 out of 100 clients participating in each round. Specifically, for FOCUS, we adhere to the settings in (Chu et al., 2023) and set the cluster number to 2. The smaller the FAA, the better.

| | FedAvg | AFL | q-FFL | FedFV | FOCUS | FedEBA+ |
|---|---|---|---|---|---|---|
| FashionMNIST | 0.7262±0.010 | 0.4500±0.006 | 0.4624±0.008 | 0.3749±0.017 | 1.16±0.161 | 0.4048±0.011 |
| CIFAR-10 | 2.296±0.031 | 0.8104±0.009 | 0.8465±0.013 | 0.7733±0.017 | 2.6448±0.061 | 0.6846±0.035 |

*Table 16.* **Performance of Algorithms with Various Metrics.** We provide the results under cosine similarity and entropy metrics, as used in (Li et al., 2019a), the geometric angle corresponds to cosine similarity metric, and KL divergence between the normalized accuracy vector $\mathbf{a}$ and uniform distribution $\mathbf{u}$ that can be directly translated to the entropy of $\mathbf{a}$. We test the algorithms on the FashionMNIST dataset, with fine-tuned hyperparameters.

| **Algorithm** | Global Acc. | Var. | Angle (○) | KL $(a\|\|u)$ |
|---|---|---|---|---|
| FedAvg | 86.49 ± 0.09 | 62.44±4.55 | 8.70±1.71 | 0.0145±0.002 |
| q-FFL | 87.05± 0.25 | 66.67± 1.39 | 7.97±0.06 | 0.0127±0.001 |
| FedMGDA+ | 84.64±0.25 | 57.89±6.21 | 8.21±1.71 | 0.0132±0.0004 |
| AFL | 85.14±0.18 | 57.39±6.13 | 7.28±0.45 | 0.0124±0.0002 |
| PropFair | 85.51±0.28 | 75.27±5.38 | 8.61±2.29 | 0.0139±0.002 |
| TERM | 84.31±0.38 | 73.46±2.06 | 9.04±0.45 | 0.0137±0.004 |
| FedFV | 86.98±0.45 | 56.63±1.85 | 8.01±1.14 | 0.0111±0.0002 |
| FedEBA+ | 87.50±0.19 | 43.41±4.34 | 6.46±0.65 | 0.0063±0.0009 |

*Table 17.* **The impact of neural networks scalability of different widths on algorithms.** To test scalability, we set up experiments with CNNs that are narrower and wider than the main paper, and provided the running time required for each communication round. Specifically, the narrower CNN includes two convolutional layers (channel 3-32-32), and three linear layers (dimension 800-128-64-10). The wider CNN includes two convolutional layers (channel 3-128-128), and three linear layers (dimension 1600-384-192-10), with all other experimental settings being the same as the default.

| Algorithm | Narrower CNN | | | | Wider CNN | | | |
|---|---|---|---|---|---|---|---|---|
| | Global Acc. ↑ | Var. ↓ | Worst 5% ↑ | Best 5% ↑ | Global Acc.↑ | Var. ↓ | Worst 5% ↑ | Best 5% ↑ |
| FedAvg | 65.37±0.27 | 116.91±1.02 | 41.60±0.86 | 84.73±1.75 | 69.93±0.46 | 79.28±3.02 | 50.61±0.50 | 85.20±0.65 |
| q-FFL | 65.22±0.71 | 106.98±1.76 | 42.33±0.52 | 84.33±1.16 | 69.60±0.48 | 74.00±3.35 | 50.27±1.52 | 83.33±0.94 |
| FedEBA+ | **70.59**±0.61 | **58.95**±6.49 | **54.67**±2.65 | 84.13±0.52 | **74.14**±0.07 | **57.35**±5.74 | **56.47**±1.04 | **85.47**±0.25 |

*Table 18.* **The impact of neural networks scalability of different depths on algorithms.** To test scalability, we set up experiments with CNNs that are shallower and deeper than the main paper, and provided the running time required for each communication round. Specifically, the shallower CNN includes only one convolutional layer (channel 3-64), and three linear layers (dimension 64-384-192-10). The deeper CNN includes three convolutional layers (channel 3-64-128-128), and three linear layers (dimension 512-384-192-10), with all other experimental settings being the same as the default.

| Algorithm | Shallower CNN | | | | Deeper CNN | | | |
|---|---|---|---|---|---|---|---|---|
| | Global Acc. ↑ | Var. ↓ | Worst 5% ↑ | Best 5% ↑ | Global Acc.↑ | Var. ↓ | Worst 5% ↑ | Best 5% ↑ |
| FedAvg | 45.10±0.86 | 119.56±17.13 | 25.53±2.66 | 67.93±2.75 | 67.71±0.45 | 82.11±5.09 | 48.40±0.33 | 83.53±1.11 |
| q-FFL | 44.82±0.82 | 108.05±7.40 | 26.33±2.22 | 66.07±0.25 | 65.75±0.42 | 77.13±8.44 | 48.81±1.39 | 81.60±0.16 |
| FedEBA+ | **46.91**±1.28 | 113.30±20.19 | 25.80±2.90 | **68.60**±1.73 | **69.67**±0.42 | **69.95**±5.55 | **51.53**±1.62 | **83.80**±0.99 |

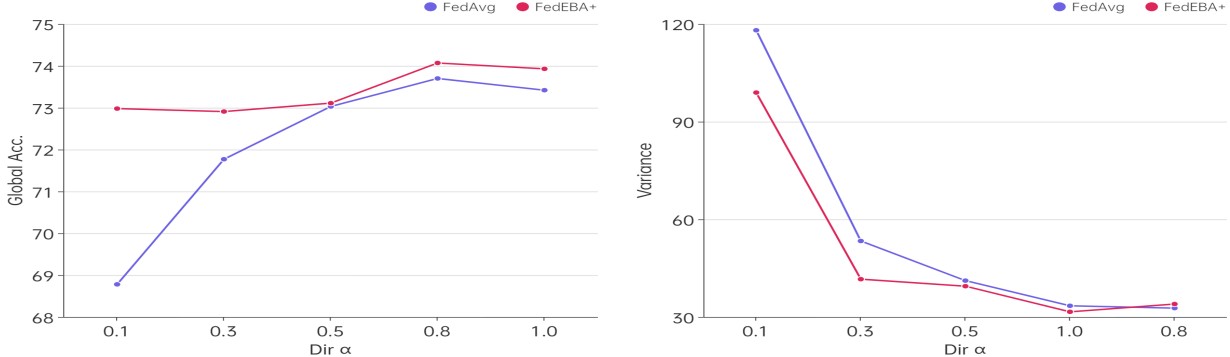

*Figure 6.* **Comparison of performance on CIFAR-10 under different degrees of Non-IID.** We performed different degrees of Non-IID partitioning on the CIFAR-10 dataset using Latent Dirichlet Allocation (LDA). Specifically, according to the degree of Non-IID from high to low, we set Dirichlet $\alpha \in \{0.1, 0.3, 0.5, 0.8, 1.0\}$. Combined with the different Non-IID partitions discussed in the main paper, this comprehensively demonstrates the performance of FedEBA+ under various scenarios.

*Table 19.* Comparison of Accuracy and Fairness on Reddit Dataset, with 20 out of 817 clients participating in each round. The batch size is set to 20. All other experimental settings strictly follow those in AAggFF (Hahn et al., 2024). For AAggFF, it uses the normal CDF, as it was identified as the best-performing curve in the AAggFF.

| Algorithm | Accuracy | Variance (Fairness) | Worst 5% | Best 5% |
|---|---|---|---|---|
| FedAvg | 13.98±1.78 | 41.99±2.45 | 3.98±0.40 | 24.92±1.15 |
| AAggFF | 13.94±0.26 | 38.89±2.26 | 5.93±0.86 | 25.74±1.12 |
| PropFair | 13.69±0.30 | 35.64±0.48 | 5.00±0.62 | 22.99±0.18 |
| qFedAvg | 13.86±0.78 | 34.69±1.33 | 4.25±0.34 | 23.39±0.71 |
| FedEBA+ | 13.90±0.35 | 34.61±2.12 | 4.21±0.13 | 26.82±0.44 |

