# OpenReview forum: "FedEBA+: Towards Fair and Effective Federated Learning via Entropy-Based Model"
_ICML.cc/2026/Conference — ICML 2026 regular_

### Official Review · Reviewer_HUzZ · 2026-03-09

**Soundness:** 3
**Presentation:** 3
**Significance:** 2
**Originality:** 2
**Overall Recommendation:** 4
**Confidence:** 3

**Summary:**

This paper explores a core and well-motivated problem in fair FL: due to data heterogeneity, existing fair FL algorithms often face the dilemma of compromising between global model performance and client performance fairness, i.e., the performance variance between different clients. To overcome this trade-off, the authors propose a novel optimization framework called FedEBA+, which creatively combines information theory principles with model gradient alignment mechanisms.

**Compliance With Llm Reviewing Policy:**

Affirmed.

**Final Justification:**

The authors addressed my concerns during the rebuttal phase. My final recommendation is 4: Weak Accept.

**Key Questions For Authors:**

(1) What is the essential difference in the final optimization behavior between the aggregation strategy of FedEBA+ and the exponential tilting used in TERM? Providing concrete scenarios where the two approaches produce significantly different outcomes would help improve my evaluation of this manuscript.

(2) Can the assumption $b_i = b$ in the variance analysis be relaxed? Do the conclusions still hold under non-uniform data distributions?

(3) Is the performance drop of the top 5% clients an inherent property of the method, or can it be mitigated through hyperparameter tuning?

(4) Could the authors provide clear guidelines to help users determine when Prac-FedEBA+ is sufficient, rather than requiring the full version with higher communication cost?

(5) In Table 2, why are Std and Var respectively used for CIFAR-100 and Tiny-ImageNet? Would the conclusions remain consistent if a unified metric were adopted?

**Limitations:**

yes

**Strengths And Weaknesses:**

Strengths :

(1) The paper clearly points out the common misconception that unconstrained maximum entropy degenerates into FedAvg, providing a convincing theoretical entry point for fair aggregation.

(2) The aggregation weights have a closed-form expression and introduce almost no additional computational overhead, making the method more practical than iterative fairness-oriented approaches.

(3) The paper provides convergence guarantees for non-convex settings. The experimental evaluation is also extensive, covering five datasets, four architectures, and more than ten baselines, along with multidimensional validations such as noise robustness and compatibility with differential privacy.

(4) Prac-FedEBA+ reduces the communication cost to a level comparable to FedAvg, demonstrating attention to practical deployment constraints.

Weaknesses:
(1) The final aggregation weight $p_i \propto \exp(F_i/\tau)$ essentially performs a softmax over the loss, which is functionally similar to approaches such as TERM and AAggFF. The paper should more candidly discuss this overlap.

(2) The variance analysis relies on simplifying assumptions such as $b_i=b$, as well as the construction of outlier clients in the inductive proof, which may deviate significantly from realistic deep learning scenarios.

(3) The claim “Beyond the Trade-off” is not fully supported by the experiments. On several datasets, the accuracy of high-performing clients decreases. Moreover, on CIFAR-100, the variance of AFL is lower than that of FedEBA+, but this observation is somewhat downplayed. The remaining trade-offs should be discussed more objectively.

(4) The best results are almost all obtained using the full version with a 50% increase in communication cost. The practical version shows a noticeable performance gap, and the paper lacks sufficient analysis explaining its appropriate usage scenarios.

---

> ### Author Rebuttal · Authors · 2026-03-31
>
> We sincerely thank you for the constructive review.
>
> Please refer to this anonymous link for the supplementary experimental results:
>
> https://anonymous.4open.science/r/ICML-Rebuttal-Tables/readme.md
>
> ### **For Weakness 1 and Question 1**
>
> We would like to highlight the differences and innovations compared to the existing reweighting algorithms:
>
> - **Idea of Aggregation**: FedEBA+ derives aggregation probabilities via constrained entropy maximization, offering new insights into client weighting.
> - **Algorithm Design**: Beyond aggregation, FedEBA+ introduces a step-wise alignment update (Sec 4.2–4.3) to push clients to update with the guidance of the ideal fair direction. The ablation experiments have demonstrated that step-wise alignment is of great significance. (Tables 1 & 2 in the link)
> - **Theoretical Analysis**: FedEBA+ is the first aggregation-based method to offer **fairness analysis** for linear and convex models, plus **non-convex convergence analysis** (w/o alignment).
> - **Experiments**: Empirical results confirm FedEBA+’s superiority.
>
> ### **For Weakness 2 and Question 2**
>
> $b_i$ represents the scale of the input features ($\Xi_i^T \Xi_i = n b_i I_d$). Assuming $b_i = b$ is a standard simplifying convention in theoretical FL analyses, practically.
>
> - **Yes, relax the assumption of $b_i$ would not change the conclusion.** The fundamental mechanism of our algorithm still holds: the maximum entropy weights assign higher probabilities to clients with larger losses (which are driven by larger $\|x - x_i\|^2$ gaps). By over-weighting and aligning toward these clients, the algorithm minimizes the maximum $\|x - x_i\|^2$ distance across the network, reducing the overall variance.
> - Furthermore, our variance analysis for the strongly convex case *does not* rely on $b_i=b$. It relies purely on Lipschitz smoothness and strong convexity, demonstrating that the variance reduction holds mathematically even under more general non-uniform data distributions.
>
> ### **For Weakness 3 and Question 3**
>
> "Beyond the Trade-off" refers specifically to the long-standing dilemma in fair FL, where improving **fairness (i.e., reducing the performance variance across all clients, defined in Sec 3, Definition 3.1)** often comes at the cost of sacrificing **global model performance (i.e., the average accuracy across all clients)**. Our work aims to break *this specific trade-off*.
>
> - Top 5% Drop: This is an inherent and often unavoidable property of any algorithm that enforces performance uniformity. So I’m afraid that parameter tuning may not help.
> - The trade-off: One may notice that AFL has a fairness advantage on CIFAR-100, but this is due to the collapse of AFL's global performance. Such a global model is meaningless. Inspired by your suggestion, we introduced the coefficient of variation $C_V = \frac{\text{Variance}}{\text{Accuracy}}$ to improve the measurement of the trade-off (Please refer to Table 10 in the link).
>
> Obviously, FedEBA+ has pushed the trade-off between performance and fairness to a new frontier. Many thanks for your insightful suggestion, and we will add $C_V$ as a quantitative indicator for the trade-off in the revision.
>
> ### **For Weakness 4 and Question 4**
>
> We are very pleased to discuss with you the practical choices:
>
> - **High-Drift Scenarios (When to use Full FedEBA+):** The full FedEBA+ is necessary in scenarios where local models are expected to drift significantly between communication rounds. This occurs when (1) the number of local epochs ($K$) is large; (2) the data heterogeneity is extreme.
> - **Moderate-Drift Scenarios (Prac-FedEBA+ is usually sufficient):** We would like to highlight that our main experiments were conducted under **highly challenging non-IID settings** (2 shards per client or Dirichlet with low $\alpha$). Even in these extreme cases, as shown in Tables 1 & 2 (in the main paper), **Prac-FedEBA+ still achieved the second-best fairness and accuracy on 3 of 4 tasks**, outperforming most other baselines while maintaining almost the same communication cost as FedAvg.
>
> Our clear guideline for users is therefore: Default to Prac-FedEBA+, and switch to the full FedEBA+ only when resources are abundant, and the highest possible performance is required, or if instability is observed in extremely high-drift scenarios
>
> ### **For Question 5**
>
> This was purely a typographical choice for table formatting. Sorry for any possible confusion in the reading.
>
> - **Of course, Standard Deviation and Variance are functionally equivalent for the purpose of ranking the uniformity of the performance distribution.** The conclusions are consistent. To avoid confusion, we will unify all tables to use Variance in the revision.
>
> We are, of course, happy to discuss any further questions you may have. We hope our detailed derivation and responses have thoroughly addressed your concerns. Thank you again for making our work much stronger.

---

> > ### Author Rebuttal · Reviewer_HUzZ · 2026-04-01
> >
> > I have no further questions and will maintain my score. Thank you for the clarification.

---

> > > ### Author Response · Authors · 2026-04-02
> > >
> > > We sincerely thank you for your positive feedback and for fully acknowledging the revisions presented in our rebuttal. We are pleased that the additional derivations, metrics, and experimental results proved helpful and addressed your concerns.
> > >
> > > We appreciate your time and insightful guidance throughout the review process.
> > >
> > > Best Regards,
> > >
> > > Authors of the submission #13444

---

### Official Review · Reviewer_ZyXc · 2026-03-09

**Soundness:** 3
**Presentation:** 3
**Significance:** 3
**Originality:** 3
**Overall Recommendation:** 5
**Confidence:** 4

**Summary:**

This paper proposes a new fairness algorithm for federated learning called FedEBA+, which enhances fairness by designing aggregation weights based on the principle of maximum entropy, and introduces a gradient alignment strategy to maintain the overall model performance. Theoretical proof of convergence has been provided, and experiments show that it outperforms existing methods on multiple datasets.

**Compliance With Llm Reviewing Policy:**

Affirmed.

**Final Justification:**

The authors have addressed my concerns during the rebuttal process. I am inclined to recommend acceptance of this manuscript.

**Key Questions For Authors:**

In practical applications, client devices often have heterogeneous model architectures. Does the gradient alignment step in FedEBA+ rely on homogeneous architectures? If the architectures are different, how can this method be generalized or modified to adapt to this heterogeneous federated learning scenario?

**Limitations:**

yes

**Strengths And Weaknesses:**

Strengths:

This paper breaks the trade-off between fairness and performance. Most existing methods, when aiming to enhance fairness, result in a decrease in global accuracy. However, FedEBA+ achieves a significant improvement in fairness while even being able to enhance or maintain the accuracy of the global model through the gradient alignment strategy.

A convergence proof in the non-convex setting is provided, and the fairness analysis is extended from simple linear models to strong convex and generalized regression scenarios.

This paper presents the practical version Prac-FedEBA+, whose communication cost is comparable to that of FedAvg.

Weaknesses:

Although the method is intended to help underperforming clients, in cases of extreme data distribution or the presence of malicious clients, over-weighting low-performance clients may introduce noise.

---

> ### Author Rebuttal · Authors · 2026-03-31
>
> We sincerely thank you for the professional and insightful feedback.
>
> Please refer to this anonymous link for the supplementary experimental results:
>
> https://anonymous.4open.science/r/ICML-Rebuttal-Tables/readme.md
>
> ### **For Weakness 1**
>
> Inspired by your suggestion, we evaluated FedEBA+:
> - **Against severe data corruption (Malicious Clients)**, where local noisy labels follow the symmetric flipping approach introduced in [1][2], with a
> noise ratio of ϵ set to 0.5. All the other settings, like the learning rate, are kept the same as the main paper settings. We’ve also conducted additional tests on the results obtained by training with the robust algorithm Local Self-Regularization (LSR) under the noise label condition. As shown in Tables 3 & 4 (please refer to the tables in the link), FedEBA+ **remained relatively robust and maintained superior fairness and accuracy compared to baselines in the extreme cases**.
> - **On extremely unbalanced** cross-device settings on the Reddit dataset (a standard large-scale cross-device benchmark from the LEAF suite).  As the results show (please refer to Table 8 in the link), FedEBA+ effectively negotiates the gradient conflicts. It prevents the global model from diverging or overfitting to any single participant, **yielding better fairness than baselines**, while keeping competitive performance.
>
> ### **For Question 1**
>
> - **Current State:** As currently formulated in the main text, the step-wise gradient alignment in FedEBA+ relies on homogeneous architectures because it directly aggregates and aligns parameter gradients ($\nabla F_i(x_t)$).
> - **Can it generalize (Model Heterogeneity)?:** Fortunately, the core principles of FedEBA+ can be generalized to heterogeneous architectures by shifting the alignment from the *parameter space* to the *output/logit space*, utilizing techniques akin to Federated Knowledge Distillation (e.g., FedMD [3], FedDF [4]). The modification would proceed as follows:
>     1. **EBA is architecture-agnostic:** Step 1 (Entropy-Based Aggregation) remains completely unchanged, as the weights $p_i$ are calculated using the scalar loss $F_i(x)$, which does not depend on the model architecture.
>     2. **Fair Logit Alignment:** Instead of sending parameter gradients, clients share their prediction logits on a small public reference dataset. The server calculates the "ideal fair global logit" by aggregating the clients' logits using our maximum-entropy weights $p_i$.
>     3. **Local Alignment via Distillation:** During local training, clients update their heterogeneous models using a combined objective: their standard local loss, plus a KL-divergence penalty that aligns their local predictions with the "ideal fair global logit" broadcast by the server.
>
> This adaptation allows the framework to enforce fairness through output alignment rather than weight alignment. We greatly appreciate this suggestion, as it highlights the broad applicability of our framework. We will add a discussion outlining this extension for heterogeneous architectures in our revision, and it is one of our future directions.
>
> We are very glad to discuss this insightful idea with you, and we hope our new results and discussion have thoroughly addressed your concerns. Thanks again for your time and effort.
>
> [1] Xuefeng Jiang, Sheng Sun, Yuwei Wang, and Min Liu. Towards federated learning against noisy labels via local self-regularization. In Proceedings of the 31st ACM International Conference on Information & Knowledge Management, pp. 862–873, 2022.
>
> [2] Xiuwen Fang and Mang Ye. Robust federated learning with noisy and heterogeneous clients. In Proceedings of the IEEE/CVF Conference on Computer Vision and Pattern Recognition, pp. 10072–10081, 2022.
>
> [3] Daliang Li and Junpu Wang. FedMD: Heterogeneous Federated Learning via Model Distillation. arXiv:1910.03581, 2019.
>
> [4] Tao Lin, Lingjing Kong, Sebastian U. Stich, and Martin Jaggi. Ensemble Distillation for Robust Model Fusion in Federated Learning. NeurIPS, 2020.

---

> > ### Author Rebuttal · Reviewer_ZyXc · 2026-04-02
> >
> > The author added new experiments and answered my questions point by point.

---

> > > ### Author Response · Authors · 2026-04-02
> > >
> > > We are delighted to have successfully addressed all your concerns. Once again, we sincerely appreciate your insightful questions and constructive suggestions, which have greatly improved our final manuscript.
> > >
> > > Best Regards,
> > >
> > > Authors of the submission #13444

---

### Official Review · Reviewer_9TAD · 2026-03-14

**Soundness:** 2
**Presentation:** 3
**Significance:** 3
**Originality:** 3
**Overall Recommendation:** 4
**Confidence:** 4

**Summary:**

This paper addresses performance fairness in federated learning (FL), where the central challenge is balancing client-level fairness and global model accuracy under heterogeneous (non‑i.i.d) data. The proposed solution, FedEBA+, integrates constrained maximum entropy aggregation with gradient and model alignment strategies. The authors provide solid theoretical analysis (convergence + fairness) and empirical validation.

**Compliance With Llm Reviewing Policy:**

Affirmed.

**Final Justification:**

Authors addressed my concerns and my misunderstanding. They also added additional experiments which strengthened the paper.

**Key Questions For Authors:**

Could the authors comment (or show) on how FedEBA+ would perform in non‑i.i.d and unbalanced settings, for example where a minority of clients holds rare classes, fewer samples, or experiences a distinct data or concept shift (e.g., rotated or corrupted inputs)?

Maybe I missed it but do the authors have practical guidance or heuristics for selecting τ and α jointly?

**Limitations:**

the possibility that enforcing uniform performance may disadvantage certain stakeholders or mask deeper group-level or demographic disparities could be an interesting point to include, especially since the paper does not discuss the potential applications for which this method is relevant. Performance fairness in FL could be more applicable to cross-device settings where a client has data from 1 person, and might not be suitable for cross-silo for instance.

**Strengths And Weaknesses:**

*Soundness*
The paper is technically sound, offering rigorous theoretical guarantees on both convergence and fairness, with some empirical validation across diverse datasets. however, I have some concerns with respect to the empirical experiments. Even if the distribution is non-i.i.d, the datasets size is the same. Performance fairness issues often rise when there is some unbalance and minority vs majority clients. Tests on such scenarios makes more sense than simply non-i.i.d (it should be non-i.i.d + unbalanced). This is because the tasks are not contradictory or opposing.   As a result the improvement also with the baselines is currently marginal, and proving the utility would require creating a minority that is different from the majority by either holding a specific subset of the classes, or having some other shift like concept shift through rotating the images for example.

*Originality and presentation*
 It is original in formulating fairness-aware federated aggregation through constrained maximum entropy with a closed-form solution, and in coupling this with alignment mechanisms to preserve global performance. The presentation is clear and well-structured, with precise algorithm descriptions and  thorough ablations. Overall, the work is significant, as it advances the theoretical understanding of fairness in federated learning while providing a practical, high-performing algorithm that consistently improves client-level fairness without sacrificing accuracy. The fairness–stability behavior of FedEBA+ is governed by two interacting hyperparameters, τ (entropy temperature) and α (alignment strength). While the authors demonstrate robustness across a reasonable range of values, the method still requires nontrivial tuning.

---

> ### Author Rebuttal · Authors · 2026-03-31
>
> We sincerely thank you for appreciating the originality of our constrained maximum entropy formulation, the rigorous theoretical guarantees, and the significance of our work.
>
> Please refer to this anonymous link for the supplementary experimental results:
>
> https://anonymous.4open.science/r/ICML-Rebuttal-Tables/readme.md
>
> ### **For Weakness 1 and Question 1**
>
> Thanks for your insightful suggestions.
>
> - **Unbalanced Sizes & Rare Classes:** We have indeed carefully considered these extreme scenarios in our original submission. In our main experiments (Section 6.1), we utilized the Dirichlet allocation ($\alpha=0.1$) for CIFAR-100 and Tiny-ImageNet. By definition, a low-alpha Dirichlet partition naturally creates severe imbalances not only in class distributions but also in **the number of samples per client**. Even under these contradictory, unbalanced conditions, FedEBA+ consistently outperformed baselines (Table 2 in the main paper).
> - **Corrupted Inputs:** Inspired by your suggestion, we add the evaluation of FedEBA+ against severe data corruption, where local noisy labels follow the symmetric flipping approach introduced in [1][2], with a noise ratio of ϵ set to 0.5. All the other settings, like the learning rate, are kept the same as the main paper settings. We conducted additional tests on the results obtained by training with the robust algorithm Local Self-Regularization (LSR) under the noise label condition.
>
> As shown in Tables 3 & 4 (please refer to the link), FedEBA+ remained highly robust and maintained superior fairness and accuracy compared to baselines in the extreme cases.
>
> ### **For Weakness 2 and Question 2**
>
> We conducted a detailed experimental analysis of the hyperparameters (please refer to Tables 5 & 6 in the link), and summarized the results as follows:
>
> - Increasing the value of $\alpha$ generally leads to an improvement in both fairness and global test accuracy. (we generally recommend a value of $\alpha \ge 0.5$.)
> - $\tau$ controls the "strictness" of the fairness intervention. As Table 6 (in the link) shows, **decreasing $\tau$ specifically forces the model to prioritize fairness** (variance drops to 70.60 at $\tau=0.1$), while global accuracy remains stable. However, setting $\tau$ too close to 0 (e.g., 0.05) makes the aggregation overly sharp, which can induce slight instability. **We recommend setting $\tau \in [0.1, 1.0]$.**
>
> Importantly, the algorithm's performance remains **relatively robust and stable** across the tested range of parameter adjustments. To further eliminate the need for manual tuning of $\tau$ altogether, practitioners can use the dynamic annealing schedule. By decaying $\tau$ over the communication rounds $T$, the algorithm automatically shifts from exploring a smoother landscape to enforcing strict fairness. We test three schedules based on an initial temperature $\tau_0$:
> *Linear:* $\tau_t = \tau_0 / (1 + T)$, *Concave:* $\tau_t = \tau_0 / (1 + T)^{1/2}$, *Convex:* $\tau_t = \tau_0 / (1 + T)^3$
>
> As shown in our extended results (please refer to Table 7 in the link), utilizing any annealing schedule effectively bypasses the need for hyperparameter tuning and yields superior fairness.
>
> ### **Discussion on Limitations and Applicability**
>
> As you suggested, we extend the evaluation to both the cross-device and cross-silo settings.
>
> - We evaluated FedEBA+ on the Reddit dataset (a standard large-scale cross-device benchmark from the LEAF suite).
> - We constructed a scenario on CIFAR-10 with exactly 10 clients, where all 10 clients participate in every round (full participation).
>
> As the results show (please refer to Tables 8 & 9 in the link), FedEBA+ effectively negotiates the gradient conflicts. It prevents the global model from diverging or overfitting to any single participant, **yielding better fairness than baselines in both scenarios while keeping competitive performance**.
>
> We are very grateful for your advice to make our work more solid. These results demonstrate the robust applicability of FedEBA+ across vastly different FL topologies. We will include all the discussions and new results in the revision. We hope our detailed results and responses have thoroughly addressed your concerns. Given these clarifications and the supplementary results to the paper, we believe the work merits a rating higher than its current one and respectfully ask that you consider this in your final evaluation.
>
> [1] Xuefeng Jiang, Sheng Sun, Yuwei Wang, and Min Liu. Towards federated learning against noisy labels via local self-regularization. In Proceedings of the 31st ACM International Conference on Information & Knowledge Management, pp. 862–873, 2022.
>
> [2] Xiuwen Fang and Mang Ye. Robust federated learning with noisy and heterogeneous clients. In Proceedings of the IEEE/CVF Conference on Computer Vision and Pattern Recognition, pp. 10072–10081, 2022.

---

> > ### Author Rebuttal · Reviewer_9TAD · 2026-04-03
> >
> > Authors added experiments that address my concerns

---

> > > ### Author Response · Authors · 2026-04-05
> > >
> > > We are glad to have addressed all your valuable comments. We sincerely thank you once again for your thoughtful questions and constructive suggestions, which have significantly enhanced the soundness of our manuscript.
> > >
> > > Best regards,
> > >
> > > Authors of submission #13444

---

### Official Review · Reviewer_qJxz · 2026-03-15

**Soundness:** 2
**Presentation:** 2
**Significance:** 2
**Originality:** 2
**Overall Recommendation:** 2
**Confidence:** 3

**Summary:**

This paper studies the tradeoff of global performance and client-level performance fairness in federated learning under heterogeneous data, using variance of per-client test loss as the main fairness notion. The core idea is to upweight higher-loss clients via a maximum-entropy-derived aggregation rule and to combine this with a gradient/model alignment mechanism intended to preserve global accuracy despite biased aggregation.The paper claims three main contributions: a constrained-entropy aggregation rule with closed-form weights, a step-wise alignment update that allegedly breaks the fairness/accuracy trade-off, and theory covering nonconvex convergence plus variance analysis beyond prior linear settings. Empirically, it reports improvements over several federated fairness baselines on FashionMNIST, CIFAR-10, CIFAR-100, and Tiny-ImageNet, with the full method outperforming FedAvg and ablated variants on variance and often on accuracy as well.

**Compliance With Llm Reviewing Policy:**

Affirmed.

**Key Questions For Authors:**

1. The update of $x^i_{t,K}$ of each local client seems missing in Algorithm 1. How is it updated?
2. What is $\tilde{f}(x)$ in Eq. (3)? It seems not computable to me.
3. Is there any theoretical justification on why the proposed method can achieve better tradeoff between global performance and local fairness?

**Limitations:**

Yes.

**Strengths And Weaknesses:**

Strengths:
- Broad baseline coverage across many fair FL methods, including q-FFL, AFL, FedMGDA+, Ditto, TERM, FOCUS, lp-proj, and AAggFF.
- The paper tries to provide more theory than many empirical FL fairness papers, including nonconvex convergence and variance analysis in stylized settings.
- The considered tradeoff is truely a dilemma in FL, and it is a hard problem in my opinion.

Weakensses:
- The mathematical definition of the problem is not well defined to me (or it's hard to follow from the presentation). Some inconsistencies in algorithm/theory presentation also reduce trust in the correctness of the theoretical results.
- Novelty is only moderate because the practical effect is still loss-based reweighting plus alignment, close to existing paradigms such as AAggFF and reweighting methods.

---

> ### Author Rebuttal · Authors · 2026-03-31
>
> We sincerely thank you for your time, effort, and valuable feedback. Our supplementary experimental results are available here:
>
> https://anonymous.4open.science/r/ICML-Rebuttal-Tables/readme.md
>
> ### **For weakness 1**
>
> The final objective of our method is:
>
> $\begin{aligned}
> \min_{x} \max_{p_i} L(x, p_i) := & \sum_{i=1}^N p_i F_i(x) - \beta \biggl[ \sum_{i=1}^N p_i \log p_i + \lambda_0 \left( \sum_{i=1}^N p_i - 1 \right) + \frac{1}{\tau} \left( \tilde{f}(x) - \sum_{i=1}^N p_i F_i(x) \right) \biggr].
> \end{aligned}$ (Eq. 6)
>
> To make it clearer, we view it as a combination of the standard FL objective and a fairness regularization term $\Phi$ (start from $-\beta$…).
>
> **Translating the Math into the Full Algorithm**
>
> Solving Eq. 6 naturally produces the two main steps of FedEBA+, which are aggregation (Sec 4.1) and step-wise alignment (Sec 4.2–4.3)
>
> - **Step 1: Aggregation (Sec. 4.1, Eq. 4):** Maximizing $\Phi$  yields our exact, closed-form exponential weights (Eq. 4).
>
>     Let's start from the Lagrangian $\Phi$ (Eq. 5)
>     Take the partial derivative with respect to $p_i$  and set it to zero, we have:
>
>     $\frac{\partial \Phi}{\partial p_i} = -\left[ \log p_i + 1 + \lambda_0 - \frac{1}{\tau} F_i(x_i) \right] = 0.$
>
>     That is,
>     $p_i = \exp\left( - \left(1 + \lambda_0  - \frac{1}{\tau} F_i(x_i) \right) \right)$
>
>     According to $\sum_{i=1}^N p_i = 1$, we have:
>     $\lambda_0 + 1 = \log \sum_{i=1}^N \exp\left( \frac{1}{\tau} F_i(x_i) \right)$
>
>     By substituting $\lambda_0 + 1$ into $p_i = \exp\left( - \left(1 + \lambda_0  - \frac{1}{\tau} F_i(x_i) \right) \right)$
>     , we obtain Eq. 4.
>
>     **We offer new insights into client reweighting (Algorithm 1 Line 14) from the perspective of maximizing entropy.**
>
> - **Step 2: Step-wise Alignment (Sec. 4.2 & 4.3):**
>
>     Compute the partial derivative of Eq. 6, and we have:
>     $\frac{\partial L\left(x,p_i\right)}{\partial x} = \left(1 + \frac{\beta}{\tau}\right) \sum_{i=1}^N p_i \nabla F_i(x) - \frac{\beta}{\tau} \nabla \tilde{f}(x)$
>
>     By defining the constant $\alpha = -\beta/\tau$, we have:
>     $\frac{\partial L\left(x,p_i\right)}{\partial x} = (1-\alpha)\sum_{i=1}^N p_i \nabla F_i(x) + \alpha \nabla \tilde{f}(x)$ (**Algorithm 1 Line 10)**.
>     **This definitely matches the result given in Eq. 7 in the main paper**. **And it is the core reason why our algorithm can outperform other reweighting-only baselines.**
>
> ### **For weakness 2**
>
> We compare FedEBA+ with AAggFF and the other reweighting methods below:
>
> - **Idea of Aggregation**: FedEBA+ derives aggregation probabilities via constrained entropy maximization, offering new insights into client weighting.
> - **Algorithm Design**: Beyond aggregation, FedEBA+ introduces a step-wise alignment update (Sec 4.2–4.3) to push clients to update with the guidance of the ideal fair direction. The ablation experiments have demonstrated that step-wise alignment is of great significance. (Tables 1 & 2 in the link)
> - **Theoretical Analysis**: FedEBA+ is the first aggregation-based method to offer **fairness analysis** for linear and convex models, plus **non-convex convergence analysis** (w/o alignment).
> - **Experiments**: Empirical results confirm FedEBA+’s superiority.
>
> ### **For Question 1**
>
> In Algorithm 1, due to the page limit, we compactly merged the entire K-step local update process into a single accumulated equation at Line 12:
>
> $\Delta^i_t = x^i_{t,K} - x^i_{t,0} = -\eta_L \sum_{k=0}^{K-1} h^i_{t,k}$
>
> The step-by-step updating equation should be:
>
> **$x^i_{t, k+1} \leftarrow x^i_{t,k} - \eta_L h^i_{t,k}$**
>
> Sorry for causing any inconvenience in reading, and we will explicitly state it in the revised version.
> ### **For Question 2**
> The Principled Approximation of the Ideal Fair Gradient $\nabla \tilde{f}(x)$ : In practical FL, only a subset of clients, $S_t$, take part in one certain communication round. Instead of computing the full global $\tilde{f}(x)$, we apply the entropy optimization to the losses of the clients in $S_t$. The resulting weighted gradient of this sample, $\tilde{g}^{b,t}$, is a **proper approximation of the ideal fair gradient** that can be constructed from the available data in round $t$.
> ### **For Question 3**
> Only $S_t$ being involved per round introduces difficulties in theoretical analysis. We’ve proved **the alignment mechanism is the key to improving the trade-off empirically.** (Tables 1 & 2 in the link)**.** It demonstrated that FedEBA+ can push performance and fairness to a more advanced level.
>
> Thank you once again for your thoughtful and constructive feedback. We have carefully integrated all of the discussions above and our new experimental results into the revised manuscript. With these clarifications and additional empirical evidence that address your concerns, we believe our work now better demonstrates its contributions and merits a higher rating than initially assigned. We respectfully ask you to reconsider your score in light of these updates.

---

> > ### Author Rebuttal · Reviewer_qJxz · 2026-04-03
> >
> > Thank you for the detailed rebuttal. In particular, Eq. (6) is helpful in clarifying the derivation of the proposed algorithm, and I appreciate its inclusion. I strongly encourage the authors to incorporate this full definition of the objective function into the main paper, as it significantly improves the transparency of the methodology. With this addition, the mathematical formulation is now clear to me.
> >
> > However, I remain unconvinced by the claim that the proposed method operates “beyond the trade-off between accuracy and fairness.” It is well established that federated learning (FL) involves an inherent trade-off between these two objectives, and a substantial body of prior work has explored strategies to navigate this tension. Based on the title, abstract, and TL;DR, I expected the paper to provide theoretical guarantees demonstrating that the proposed formulation and algorithm achieve an improved trade-off relative to existing approaches. At present, I do not see sufficient theoretical justification explaining why Eq. (6) leads to solutions with a better accuracy–fairness trade-off compared to prior methods.
> >
> > Overall, while the revisions have improved the clarity of the work, I do not yet find that the paper meets the standard expected for publication at ICML. Therefore, I will maintain my current score.

---

> > > ### Author Response · Authors · 2026-04-04
> > >
> > > We are delighted to hear that Eq. (6) has been helpful in clarifying the derivation and significantly improving the transparency of the methodology. Thank you for acknowledging our originally designed Eq. (6); as you suggested, we have incorporated the full definition of the objective function and the detailed derivation into the main paper.
> > >
> > > Please refer to the link for the supplementary experimental results:
> > >
> > > https://anonymous.4open.science/r/ICML-Rebuttal-Tables/readme.md
> > >
> > > ### **The "beyond the trade-off'' claim**
> > > We acknowledge that the phrase "beyond the trade-off'' is misleading and apologize for the confusion. Our actual goal is to improve fairness while maintaining accuracy, as stated in the main paper and TL;DR. Accordingly, and in line with prior studies, our theoretical analysis focuses on two metrics, i.e., fairness (via variance) and accuracy (via convergence) separately, rather than providing a strict theoretical guarantee on the accuracy–fairness trade-off. We agree that a theoretical characterization of the optimal accuracy–fairness trade-off is an important open problem, and is worth exploring in future work.
> > >
> > > Motivated by your comments, we have revised the full paper accordingly to better reflect this more precise positioning, specifically:
> > >
> > > Title: Beyond the Trade-off: Unifying Fairness and Performance in Federated Learning **$\rightarrow$ Unifying Fairness and Accuracy in Federated Learning via Entropy-Based Aggregation and Step-wise Alignment**
> > >
> > > Line 10: Federated Learning (FL) often suffers from a trade-off between global model performance and client-level fairness **$\rightarrow$** Federated Learning (FL) often suffers from **sacrificing global model accuracy when improving client-level fairness.**
> > >
> > > Line 16: Existing fair FL algorithms face a trade-off  $\rightarrow$ Existing fair FL algorithms face a **bottleneck**
> > >
> > > Line 20: In this paper, we propose a novel framework that bridges this trade-off   $\rightarrow$ In this paper, we propose a novel framework that **effectively improves fairness while preserving global accuracy**
> > >
> > > Line 19 (the right-side column): but often encounter a trade-off in global model utility   $\rightarrow$ but often **come at the cost of global model accuracy**
> > >
> > > Line 378: many baselines face an accuracy-variance trade-off   $\rightarrow$ many baselines **improve fairness (i.e., reduce variance) at the expense of global model accuracy**
> > >
> > > Line 442: a framework that effectively breaks the long-standing trade-off between global performance and client-level fairness   $\rightarrow$ This paper introduces FedEBA+, a framework that **improves client-level fairness while maintaining strong global accuracy**
> > > ### **New empirical results on accuracy–fairness trade-off**
> > > Inspired by the interesting discussion on the trade-off and your comments, we introduce a new metric for the accuracy–fairness trade-off, denoted by $C_V = \frac{\text{Variance}}{\text{Accuracy}}$ and provide new empirical results.
> > >
> > > Table 10: Accuracy–Fairness Trade-off $C_V$ on CIFAR-100 and Tiny-ImageNet.
> > >
> > > | Algorithm |CIFAR-100 $C_V$ ↓|Tiny-ImageNet $C_V$ ↓|
> > > |---|---|---|
> > > |FedAvg|9.61|0.32|
> > > |q-FFL|8.47|0.25|
> > > |AFL|6.15|0.27|
> > > |FedFV|9.81|0.25|
> > > |FedMGDA+|8.80|0.28|
> > > |TERM|10.20|0.32|
> > > |AAggFF|9.21|0.26|
> > > |Prac-FedEBA+|7.26|0.24|
> > > |FedEBA+|**5.91**|**0.22**|
> > >
> > > Table 11: Accuracy–Fairness Trade-off $C_V$ on FashionMNIST and CIFAR-10.
> > >
> > > |Algorithm|FashionMNIST $C_V$↓|CIFAR-10 $C_V$↓|
> > > |---|---|---|
> > > |FedAvg|0.72|1.53|
> > > |q-FFL|0.63|1.42|
> > > |FedMGDA+|0.68|1.38|
> > > |Ditto|0.64|1.43|
> > > |AFL|1.08|1.34|
> > > |PropFair|0.88|1.21|
> > > |TERM|0.87|1.41|
> > > |FOCUS|0.71|7.64|
> > > |lp-proj|0.66|1.14|
> > > |Rank-Core-Fed|0.68|1.30|
> > > |AAggFF|0.67|1.28|
> > > |Prac-FedEBA+|0.54|1.06|
> > > |FedEBA+|**0.50**|**0.94**|
> > >
> > > As shown in Tables 10 and 11, in terms of the accuracy–fairness trade-off metric $C_V$, FedEBA+ outperforms all baselines across four datasets.
> > >
> > > ### **Theoretical justification.**
> > >
> > > We, in line with prior studies, theoretically analyze fairness (variance) and accuracy (convergence) separately, as summarized in Table 12. We extend the fairness (variance) analysis from linear models, as typically considered in prior works, to the general strongly convex setting; and provide the convergence analysis in the challenging non-convex setting.
> > >
> > > Table 12: Theoretical analysis comparison
> > >
> > > |Algorithm|Fairness analysis (Variance) |Convergence analysis|
> > > |---|---|---|
> > > |q-FFL|Linear|None|
> > > |FedMGDA+|None|Strongly convex|
> > > |TERM|Linear|Strongly convex|
> > > |AFL|None|Convex|
> > > | PropFair | None | Non-convex |
> > > | lp-proj | Linear | Non-convex |
> > > | AAggFF | None | None |
> > > | **FedEBA+ (Ours)** | **Linear & Strongly convex** | **Non-convex** |
> > >
> > > We sincerely thank you for the valuable suggestions and inspiration, which largely strengthened our work. Considering our novel objective modeling, algorithm design, more complete theoretical analysis, and stronger empirical results, we believe that our contributions are substantial and respectfully ask that you consider this in your final evaluation.

---

### Decision · Program_Chairs · 2026-04-30

**Decision:**

Accept (regular)

**Comment:**

This paper studies the fairness-performance tradeoff in FL in the presence of data heterogeneity, global model performance vs. variance in client performance. It proposes a new fair algorithm FedEBA+ that enhances fairness by designing aggregation weights based on the principle of maximum entropy, and introduces a gradient alignment strategy to maintain the overall model performance.

All reviewers commented on the theoretical development of the paper with most of them appreciating its rigor. The overall reviews are somewhat divergent (2, 4, 4, 5), and the AC was not able to get the reviewers to engage in discussion to reach a consensus. The negative review concerns the practical impact of the proposed method, while pointing out that the paper title is perhaps overstating the result, as technically its method has not moved past the fundamental fairness-accuracy tradeoff.

The AC is leaning more positive on the paper, taking into consideration all reviews.